# Robustifying Algorithms of Learning Latent Trees with Vector Variables

**Fengzhuo Zhang**
Department of Electrical and Computer Engineering
National University of Singapore
fzzhang@u.nus.edu

**Vincent Y. F. Tan**
Department of Electrical and Computer Engineering
Department of Mathematics
National University of Singapore
vtan@nus.edu.sg

## Abstract

We consider learning the structures of Gaussian latent tree models with vector observations when a subset of them are arbitrarily corrupted. First, we present the sample complexities of Recursive Grouping (RG) and Chow-Liu Recursive Grouping (CLRG) without the assumption that the effective depth is bounded in the number of observed nodes, significantly generalizing the results in Choi et al. (2011). We show that Chow-Liu initialization in CLRG greatly reduces the sample complexity of RG from being exponential in the diameter of the tree to only logarithmic in the diameter for the hidden Markov model (HMM). Second, we robustify RG, CLRG, Neighbor Joining (NJ) and Spectral NJ (SNJ) by using the truncated inner product. These robustified algorithms can tolerate a number of corruptions up to the square root of the number of clean samples. Finally, we derive the first known instance-dependent impossibility result for structure learning of latent trees. The optimalities of the robust version of CLRG and NJ are verified by comparing their sample complexities and the impossibility result.

## 1 Introduction

Latent graphical models provide a succinct representation of the dependencies among observed and latent variables. Each node in the graphical model represents a random variable or a random vector, and the dependencies among these variables are captured by the edges among nodes. Graphical models are widely used in domains from biology [1], computer vision [2] and social networks [3].

This paper focuses on the structure learning problem of latent tree-structured Gaussian graphical models (GGM) in which the node observations are random *vectors* and a subset of the observations can be *arbitrarily corrupted*. This classical problem, in which the variables are *clean scalar* random variables, has been studied extensively in the past decades. The first information distance-based method, NJ, was proposed in [1] to learn the structure of phylogenetic trees. This method makes use of additive information distances to deduce the existence of hidden nodes and introduce edges between hidden and observed nodes. RG, proposed in [4], generalizes the information distance-based methods to make it applicable for the latent graphical models with general structures. Different from these information distance-based methods, quartet-based methods [5] utilize the relative geometry of every four nodes to estimate the structure of the whole graph. Although experimental comparisons of these algorithms were conducted in some works [4, 6, 7], since there is no instance-dependent

35th Conference on Neural Information Processing Systems (NeurIPS 2021).

impossibility result of the sample complexity of structure learning problem of latent tree graphical models, no thorough theoretical comparisons have been made, and the optimal dependencies on the diameter of graphs and the maximal distance between nodes $\rho_{\max}$ have not been found.

The success of the previously-mentioned algorithms relies on the assumption that the observations are i.i.d. samples from the generating distribution. The structure learning of latent graphical models in presence of (random or adversarial) noise remains a relatively unexplored problem. The presence of the noise in the samples violates the i.i.d. assumption. Consequently, classical algorithms may suffer from severe performance degradation in the noisy setting. There are some works studying the problem of structure learning of graphical models with noisy samples, where all the nodes in the graphical models are observed and not hidden. Several assumptions on the additive noise are made in these works, which limit the use of these proposed algorithms. For example, the covariance matrix of the noise is specified in [8], and the independence and/or distribution of the noise is assumed in [9–11, 7]. In contrast, we consider the structure learning of latent tree graphical models with *arbitrary* corruptions, where assumptions on the distribution and independence of the noise across nodes are not required [12]. Furthermore, the corruptions are allowed to be presented at *any position* in the data matrix; they do not appear solely as outliers. In this work, we derive bounds on the maximum number of corruptions that can be tolerated for a variety of algorithms, and yet structure learning can succeed with high probability.

Firstly, we derive the sample complexities of RG and CLRG where each node represents a random *vector*; this differs from previous works where each node is *scalar* random variable (e.g., [4, 13]). We explore the dependence of the sample complexities on the parameters of the graph. Compared with [4, Theorem 12], the derived sample complexities are applicable to a wider class of latent trees and capture the dependencies on more parameters of the underlying graphical models, such as $\rho_{\max}$, the maximum distance between any two nodes, and $\delta_{\min}$, the minimum over all determinants of the covariance matrices of the vector variables. Our sample complexity analysis clearly demonstrates and precisely quantifies the effectiveness of the Chow-Liu [14] initialization step in CLRG; this has been only verified experimentally [4]. For the particular case of the HMM, we show that the Chow-Liu initialization step reduces the sample complexity of RG which is $O\left(\left(\frac{9}{2}\right)^{\mathrm{Diam}(\mathbb{T})}\right)$ to $O\left(\log \mathrm{Diam}(\mathbb{T})\right)$, where $\mathrm{Diam}(\mathbb{T})$ is the tree diameter.

Secondly, we robustify RG, CLRG, NJ and SNJ by using the truncated inner product [15] to estimate the information distances in the presence of arbitrary corruptions. We derive their sample complexities and show that they can tolerate $n_1 = O\left(\frac{\sqrt{n_2}}{\log n_2}\right)$ corruptions, where $n_2$ is the number of clean samples.

Finally, we derive the first known instance-dependent impossibility result for learning latent trees. The dependencies on the number of observed nodes $|\mathcal{V}_{\mathrm{obs}}|$ and the maximum distance $\rho_{\max}$ are delineated. The comparison of the sample complexities of the structure learning algorithms and the impossibility result demonstrates the optimality of Robust Chow-Liu Recursive Grouping (RCLRG) and Robust Neighbor Joining (RNJ) in $\mathrm{Diam}(\mathbb{T})$ for some archetypal latent tree structures.

**Notation** We use san-serif letters $x$, boldface letters $\mathbf{x}$, and bold uppercase letters $\mathbf{X}$ to denote variables, vectors and matrices, respectively. The notations $[\mathbf{x}]_i$, $[\mathbf{X}]_{ij}$, $[\mathbf{X}]_{:,j}$ and $\mathrm{diag}(\mathbf{X})$ are respectively the $i^{\mathrm{th}}$ entry of vector $\mathbf{x}$, the $(i, j)^{\mathrm{th}}$ entry of $\mathbf{X}$, the $j^{\mathrm{th}}$ column of $\mathbf{X}$, and the diagonal entries of matrix $\mathbf{X}$. The notation $x^{(k)}$ represents the $k^{\mathrm{th}}$ sample of $x$. $\|\mathbf{x}\|_0$ is the $l_0$ norm of the vector $\mathbf{x}$, i.e., the number of non-zero terms in $\mathbf{x}$. The set $\{1, \ldots, n\}$ is denoted as $[n]$. For a tree $\mathbb{T} = (\mathcal{V}, \mathcal{E})$, the internal (non-leaf) nodes, the maximal degree and the diameter of $\mathbb{T}$ are denoted as $\mathrm{Int}(\mathbb{T})$, $\mathrm{Deg}(\mathbb{T})$, and $\mathrm{Diam}(\mathbb{T})$, respectively. We denote the closed neighborhood and the degree of $x_i$ as $\mathrm{nbd}[x_i; \mathbb{T}]$ and $\deg(i)$, respectively. The length of the (unique) path connecting $x_i$ and $x_j$ is denoted as $\mathrm{d}_{\mathbb{T}}(x_i, x_j)$.

## 2 Preliminaries and problem statement

A GGM [16, 17] is a multivariate Gaussian distribution that factorizes according to an undirected graph $\mathbb{G} = (\mathcal{V}, \mathcal{E})$. More precisely, a $l_{\mathrm{sum}}$-dimensional random vector $\mathbf{x} = [\mathbf{x}_1^\top, \ldots, \mathbf{x}_p^\top]^\top$, where $\mathbf{x}_i \in \mathbb{R}^{l_i}$ and $l_{\mathrm{sum}} = \sum_{i=1}^p l_i$, follows a Gaussian distribution $\mathcal{N}(\mathbf{0}, \mathbf{\Sigma})$, and it is said to be *Markov* on a graph $\mathbb{G} = (\mathcal{V}, \mathcal{E})$ with vertex set $\mathcal{V} = \{x_1, \ldots, x_p\}$ and edge set $\mathcal{E} \subseteq \binom{\mathcal{V}}{2}$ and $(x_i, x_j) \in \mathcal{E}$ if

and only if the $(i, j)^{\text{th}}$ block $\boldsymbol{\Theta}_{ij}$ of the precision $\boldsymbol{\Theta} = \boldsymbol{\Sigma}^{-1}$ is not the zero matrix $\mathbf{0}$. We focus on tree-structured graphical models, which factorize according to acyclic and connected (tree) graphs.

A special class of graphical models is the set of *latent* graphical models $\mathbb{G} = (\mathcal{V}, \mathcal{E})$. The vertex set $\mathcal{V}$ is decomposed as $\mathcal{V} = \mathcal{V}_{\text{hid}} \cup \mathcal{V}_{\text{obs}}$. We only have access to $n$ i.i.d. samples drawn from the observed set of nodes $\mathcal{V}_{\text{obs}}$. The two goals of any structure learning algorithm are to learn the identities of the hidden nodes $\mathcal{V}_{\text{hid}}$ and how they are connected to the observed nodes.

## 2.1 System model for arbitrary corruptions

We consider tree-structured GGMs $\mathbb{T} = (\mathcal{V}, \mathcal{E})$ with observed nodes $\mathcal{V}_{\text{obs}} = \{x_1, \cdots, x_o\}$ and hidden nodes $\mathcal{V}_{\text{hid}} = \{x_{o+1}, \cdots, x_{o+h}\}$, where $\mathcal{V} = \mathcal{V}_{\text{hid}} \cup \mathcal{V}_{\text{obs}}$ and $\mathcal{E} \subseteq \binom{\mathcal{V}}{2}$. Each node $x_i$ represents a random *vector* $\mathbf{x}_i \in \mathbb{R}^{l_i}$. The concatenation of these random vectors is a multivariate Gaussian random vector with zero mean and covariance matrix $\boldsymbol{\Sigma}$ with size $l_{\text{sum}} \times l_{\text{sum}}$.

We have $n$ i.i.d. samples $\tilde{\mathbf{X}}_j = [\tilde{\mathbf{x}}_1^{(j)\top}, \cdots, \tilde{\mathbf{x}}_o^{(j)\top}]^\top \in \mathbb{R}^{l_{\text{sum}}}, j = 1, \ldots, n$ drawn from the observed nodes $\mathcal{V}_{\text{obs}} = \{x_1, \cdots, x_o\}$. However, the *observed* data matrix $\tilde{\mathbf{X}}_1^n = [\tilde{\mathbf{X}}_1, \cdots, \tilde{\mathbf{X}}_n]^\top \in \mathbb{R}^{n \times l_{\text{sum}}}$ may contain some corrupted elements. We allow an level-$(n_1/2)$ arbitrary corruption in the data matrix. This is made precise in the following definition.

**Definition 1** (Level-$m$ arbitrary corruption). *For the data matrix $\tilde{\mathbf{X}}_1^n \in \mathbb{R}^{n \times k}$ formed by $n$ clean samples of $k$ random variables (or a random vector of dimension $k$), an* level-$m$ *arbitrary corruption transforms $\tilde{\mathbf{X}}_1^n$ into $\mathbf{X}_1^n \in \mathbb{R}^{n \times k}$ such that*

$$\|[\tilde{\mathbf{X}}_1^n]_{:,i} - [\mathbf{X}_1^n]_{:,i}\|_0 \leq m \quad \text{for all} \quad i = 1, \ldots, k. \tag{1}$$

Definition 1 implies that there are at most $n_1/2$ corrupted terms in each column of $\mathbf{X}_1^n$; the remaining $n - n_1/2$ samples in this column are clean. In particular, the corrupted samples in different columns need not to be in the same rows. If the corruptions in different columns lie in the same rows, as shown in (the left of) Fig. 3, all the samples in the corresponding rows are corrupted; these are called *outliers*. Obviously, outliers form a special case of our corruption model. Since each variable has at most $n_1/2$ corrupted samples, the sample-wise inner product between two variables has at least $n_2 = n - n_1$ clean samples. There is no constraint on the statistical dependence or patterns of the corruptions. Unlike fixing the covariance matrix of the noise [8] or keeping the noise independent [9], we allow *arbitrary* corruptions on the samples, which means that the noise can have unbounded amplitude, can be dependent, and even can be generated from another graphical model (as we will see in the experimental results in Section 3.6).

## 2.2 Structural and distributional assumptions

To construct the correct latent tree from samples of observed nodes, it is imperative to constrain the class of latent trees to guarantee that the information from the distribution of observed nodes $p(\mathbf{x}_1, \ldots, \mathbf{x}_o)$ is sufficient to construct the tree. The distribution $p(\mathbf{x}_1, \ldots, \mathbf{x}_{o+h})$ of the observed and hidden nodes is said to have a *redundant* hidden node $x_j$ if the distribution of the observed nodes $p(\mathbf{x}_1, \ldots, \mathbf{x}_o)$ remains the same after we marginalize over $x_j$. To ensure that a latent tree can be constructed with no ambiguity, we need to guarantee that the true distribution does not have any redundant hidden node(s), which is achieved by following two conditions [18]: (C1) Each hidden node has at least three neighbors; the set of such latent trees is denoted as $\mathcal{T}_{\geq 3}$; (C2) Any two variables connected by an edge are neither perfectly dependent nor independent.

**Assumption 1.** *The dimensions of all the random vectors are all equal to $l_{\text{max}}$.*

In fact, we only require the random vectors of the internal (non-leaf) nodes to have the same length. However, for ease of notation, we assume that the dimensions of all the random vectors are $l_{\text{max}}$.

**Assumption 2.** *For every $x_i, x_j \in \mathcal{V}$, the covariance matrix $\boldsymbol{\Sigma}_{ij} = \mathbb{E}[\mathbf{x}_i \mathbf{x}_j^\top]$ has full rank, and the smallest singular value of $\boldsymbol{\Sigma}_{ij}$ is lower bounded by $\gamma_{\text{min}}$, i.e.,*

$$\sigma_{l_{\text{max}}}(\boldsymbol{\Sigma}_{ij}) \geq \gamma_{\text{min}} \quad \text{for all} \quad x_i, x_j \in \mathcal{V}, \tag{2}$$

*where $\sigma_i(\boldsymbol{\Sigma})$ is the $i^{\text{th}}$ largest singular value of $\boldsymbol{\Sigma}$.*

This assumption is a strengthening of Condition (C2) when each node represents a random vector.

**Assumption 3.** *The determinant of the covariance matrix of any node* $\boldsymbol{\Sigma}_{ii} = \mathbb{E}\big[\mathbf{x}_i \mathbf{x}_i^\top\big]$ *is lower bounded by* $\delta_{\min}$, *and the diagonal terms of the covariance matrix are upper bounded by* $\sigma_{\max}^2$, *i.e.,*

$$\min_{x_i \in \mathcal{V}} \det(\boldsymbol{\Sigma}_{ii}) \geq \delta_{\min} \quad and \quad \max_{x_i \in \mathcal{V}} \operatorname{diag}(\boldsymbol{\Sigma}_{ii}) \leq \sigma_{\max}^2. \tag{3}$$

Assumption 3 is natural; otherwise, $\boldsymbol{\Sigma}_{ii}$ may be arbitrarily close to a singular matrix.

**Assumption 4.** *The degree of each node is upper bounded by* $d_{\max}$, *i.e.,* $\operatorname{Deg}(\mathbb{T}) \leq d_{\max}$.

### 2.3 Information distance

We define the *information distance* for Gaussian random vectors and prove that it is additive for trees.

**Definition 2.** *The information distance between nodes* $x_i$ *and* $x_j$ *is*

$$\mathrm{d}(x_i, x_j) = -\log \frac{\prod_{k=1}^{l_{\max}} \sigma_k(\boldsymbol{\Sigma}_{ij})}{\sqrt{\det(\boldsymbol{\Sigma}_{ii}) \det(\boldsymbol{\Sigma}_{jj})}}. \tag{4}$$

Condition (C2) can be equivalently restated as constraints on the information distance.

**Assumption 5.** *There exist two constants* $0 < \rho_{\min} \leq \rho_{\max} < \infty$ *such that.*

$$\rho_{\min} \leq \mathrm{d}(x_i, x_j) \leq \rho_{\max} \quad for\ all \quad x_i, x_j \in \mathcal{V}. \tag{5}$$

Assumptions 2 and 5 both describe the properties of the correlation between random vectors from different perspectives. In fact, we can relate the constraints in these two assumptions as follows:

$$\gamma_{\min} e^{\rho_{\max}/l_{\max}} \geq \delta_{\min}^{1/l_{\max}}. \tag{6}$$

**Proposition 1.** *If Assumptions 1 and 2 hold,* $\mathrm{d}(\cdot, \cdot)$ *defined in Definition 2 is additive on the tree-structured GGM* $\mathbb{T} = (\mathcal{V}, \mathcal{E})$. *In other words,* $\mathrm{d}(x_i, x_k) = \mathrm{d}(x_i, x_j) + \mathrm{d}(x_j, x_k)$ *holds for any two nodes* $x_i, x_k \in \mathcal{V}$ *and any node* $x_j$ *on the path connecting* $x_i$ *and* $x_k$ *in* $\mathbb{T}$.

This additivity property is used extensively in the following algorithms. It was first stated and proved in Huang et al. [19]. We provide an alternative proof in Appendix G.

## 3 Robustifying latent tree structure learning algorithms

### 3.1 Robust estimation of information distances

Before delving into the details of robustifying latent tree structure learning algorithms, we first introduce the truncated inner product [15], which estimates the correlation against arbitrary corruption effectively and serves as a basis for the robust latent tree structure learning algorithms. Given $\mathbf{a}, \mathbf{b} \in \mathbb{R}^n$ and an integer $n_1$, we compute $q_i = a_i b_i$ for $i = 1, 2, \ldots, n$ and sort $\{|q_i|\}$. Let $\Upsilon$ be the index set of the $n - n_1$ smallest $|q_i|$'s. The truncated inner product is $\langle \mathbf{a}, \mathbf{b} \rangle_{n_1} = \sum_{i \in \Upsilon} q_i$. Note that the implementation of the truncated inner product requires the knowledge of corruption level $n_1$.

To estimate the information distance defined in Definition 2, we implement the truncated inner product to estimate each term of $\boldsymbol{\Sigma}_{ij}$, i.e., $[\hat{\boldsymbol{\Sigma}}_{ij}]_{st} = \frac{1}{n-n_1} \langle [\mathbf{X}_1^n]_{:,(i-1)l_{\max}+s}, [\mathbf{X}_1^n]_{:,(j-1)l_{\max}+t} \rangle_{n_1}$. Then the information distance is computed based on this estimate of $\boldsymbol{\Sigma}_{ij}$ as

$$\hat{\mathrm{d}}(x_i, x_j) = -\log \prod_{k=1}^{l_{\max}} \sigma_k(\hat{\boldsymbol{\Sigma}}_{ij}) + \frac{1}{2} \log \det(\hat{\boldsymbol{\Sigma}}_{ii}) + \frac{1}{2} \log \det(\hat{\boldsymbol{\Sigma}}_{jj}). \tag{7}$$

The truncated inner product guarantees that $\hat{\boldsymbol{\Sigma}}_{ij}$ converges in probability to $\boldsymbol{\Sigma}_{ij}$, which further ensures the convergence of the singular values and the determinant of $\boldsymbol{\Sigma}_{ij}$ to their nominal values.

**Proposition 2.** *If Assumptions 1 and 2 hold, the estimate of the information distance between* $x_i$ *and* $x_j$ *based on the truncated inner product* $\hat{\mathrm{d}}(x_i, x_j)$ *satisfies*

$$\mathbb{P}\Big(\big|\hat{\mathrm{d}}(x_i, x_j) - \mathrm{d}(x_i, x_j)\big| > \frac{2l_{\max}^2}{\gamma_{\min}}(t_1 + t_2)\Big) \leq 2l_{\max}^2 e^{-\frac{3n_2}{16\kappa n_1} t_1} + l_{\max}^2 e^{-c\frac{n_2}{\kappa^2} t_2^2}, \tag{8}$$

*where* $t_2 < \kappa = \max\{\sigma_{\max}^2, \rho_{\min}\}$, *and* $c$ *is an absolute constant.*

The first and second parts of (8) originate from the corrupted and clean samples respectively.

## 3.2 Robust Recursive Grouping algorithm

The RG algorithm was proposed in [4] to learn latent tree models with additive information distances. We extend the RG to be applicable to GGMs with vector observations and robustify it to learn the tree structure against arbitrary corruptions. We call this robustified algorithm Robust Recursive Grouping (RRG). RRG makes use of the additivity of information distance to identify the relationship between nodes. For any three nodes $x_i$, $x_j$ and $x_k$, the difference between the information distances $\mathrm{d}(x_i, x_k)$ and $\mathrm{d}(x_j, x_k)$ is denoted as $\Phi_{ijk} = \mathrm{d}(x_i, x_k) - \mathrm{d}(x_j, x_k)$.

**Lemma 3.** *[4] For information distances $\mathrm{d}(x_i, x_j)$ for all nodes $x_i, x_j \in \mathcal{V}$ in a tree $\mathbb{T} \in \mathcal{T}_{\geq 3}$, $\Phi_{ijk}$ has following two properties: (1) $\Phi_{ijk} = \mathrm{d}(x_i, x_j)$ for all $x_k \in \mathcal{V}\backslash\{x_i, x_j\}$ if and only if $x_i$ is a leaf node and $x_j$ is the parent of $x_j$ and (2) $-\mathrm{d}(x_i, x_j) < \Phi_{ijk'} = \Phi_{ijk} < \mathrm{d}(x_i, x_j)$ for all $x_k, x_{k'} \in \mathcal{V}\backslash\{x_i, x_j\}$ if and only if $x_i$ and $x_j$ are leaves and share the same parent.*

RRG initializes the active set $\Gamma^1$ to be the set of all observed nodes. In the $i^{\text{th}}$ iteration, as shown in Algorithm 1, RRG adopts Lemma 3 to identify relationships among nodes in active set $\Gamma^i$, and it removes the nodes identified as siblings and children from $\Gamma^i$ and adds newly introduced hidden nodes to form the active set $\Gamma^{i+1}$ in the $(i+1)^{\text{st}}$ iteration. The procedure of estimating the distances between the newly-introduced hidden node $x_{\text{new}}$ and other nodes is as follows. For the node $x_i$ which is the child of $x_{\text{new}}$, i.e., $x_i \in \mathcal{C}(x_{\text{new}})$, the information distance is estimated as

$$\hat{\mathrm{d}}(x_i, x_{\text{new}}) = \frac{1}{2(|\mathcal{C}(x_{\text{new}})| - 1)}\left(\sum_{j \in \mathcal{C}(x_{\text{new}})} \hat{\mathrm{d}}(x_i, x_j) + \frac{1}{|\mathcal{K}_{ij}|}\sum_{k \in \mathcal{K}_{ij}} \hat{\Phi}_{ijk}\right), \qquad (9)$$

where $\mathcal{K}_{ij} = \{x_k \in \mathcal{V}\backslash\{x_i, x_j\} : \max\{\hat{\mathrm{d}}(x_i, x_k), \hat{\mathrm{d}}(x_j, x_k)\} < \tau\}$ for some threshold $\tau > 0$. For $x_i \notin \mathcal{C}(x_{\text{new}})$, the distance is estimated as

$$\hat{\mathrm{d}}(x_i, x_{\text{new}}) = \begin{cases} \sum_{x_k \in \mathcal{C}(x_{\text{new}})} \frac{\hat{\mathrm{d}}(x_k, x_i) - \hat{\mathrm{d}}(x_k, x_{\text{new}})}{|\mathcal{C}(x_{\text{new}})|}. & \text{if } x_i \in \mathcal{V}_{\text{obs}} \\ \sum_{(x_k, x_j) \in \mathcal{C}(x_{\text{new}}) \times \mathcal{C}(i)} \frac{\hat{\mathrm{d}}(x_k, x_j) - \hat{\mathrm{d}}(x_k, x_{\text{new}}) - \hat{\mathrm{d}}(x_j, y_i)}{|\mathcal{C}(x_{\text{new}})||\mathcal{C}(i)|} & \text{otherwise} \end{cases}. \qquad (10)$$

The set $\mathcal{K}_{ij}$ is designed to ensure that the nodes involved in the calculation of information distances are not too far, since estimating long distances accurately requires a large number of samples. The maximal cardinality of $\mathcal{K}_{ij}$ over all nodes $x_i, x_j \in \mathcal{V}$ can be found, and we denote this as $N_\tau$, i.e., $|\mathcal{K}_{ij}| \leq N_\tau$.

The observed nodes are placed in the $0^{\text{th}}$ layer. The hidden nodes introduced in $i^{\text{th}}$ iteration are placed in $i^{\text{th}}$ layer. The nodes in the $i^{\text{th}}$ layer are in the active set $\Gamma^{i+1}$ in the $(i+1)^{\text{st}}$ iteration, but nodes in $\Gamma^{i+1}$ can be nodes created in the $j^{\text{th}}$ iteration, where $j < i$. For example, in Fig. 1, nodes $x_{12}$, $x_{14}$ and $x_{15}$ are created in the $1^{\text{st}}$ iteration, and they are in $\Gamma^2$. Nodes $x_1$, $x_2$ and $x_5$ are also in $\Gamma^2$, which are observed nodes. Eqns. (9) and (10) imply that the estimation error in the $0^{\text{th}}$ layer will propagate to the nodes in higher layers, and it is necessary to derive concentration results for the information distance related to the nodes in higher layers. To avoid repeating complicated expressions in the various concentration bounds to follow, we define the function

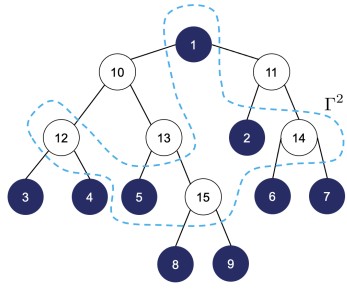

Figure 1: An illustration of the active set. The shaded nodes are the observed nodes and the rest are hidden nodes.

$$f(x) \triangleq 2l_{\max}^2 e^{-\frac{3n_2}{32\lambda\kappa n_1}x} + l_{\max}^2 e^{-c\frac{n_2}{4\lambda^2\kappa^2}x^2} =: ae^{-wx} + be^{-ux^2},$$

where $\lambda = 2l_{\max}^2 e^{\rho_{\max}/l_{\max}}/\delta_{\min}^{1/l_{\max}}$, $w = \frac{3n_2}{32\lambda\kappa n_1}$, $u = c\frac{n_2}{4\lambda^2\kappa^2}$, $a = 2l_{\max}^2$ and $b = l_{\max}^2$. To assess the proximity of the estimates $\hat{\mathrm{d}}(x_i, x_{\text{new}})$ in (9) and (10) to their nominal versions, we define

$$h^{(l)}(x) \triangleq s^l f(m^l x) = s^l\left(ae^{-wm^l x} + be^{-um^{2l}x^2}\right) \quad \text{for all} \quad l \in \mathbb{N} \cup \{0\}. \qquad (11)$$

where $s = d_{\max}^2 + 2d_{\max}^3(1 + 2N_\tau)$ and $m = 2/9$. The following proposition yields *recursive estimates* for the errors of the distances at various layers of the learned latent tree.

**Proposition 4.** *With Assumptions 1–5, if we implement the truncated inner product to estimate the information distance among observed nodes and adopt* (9) *and* (10) *to estimate the information*

*distances related to newly introduced hidden nodes, then the information distance related to the hidden nodes $x_{\mathrm{new}}$ created in the $l^{\mathrm{th}}$ layer $\hat{\mathrm{d}}(x_i, x_{\mathrm{new}})$ satisfies*

$$\mathbb{P}\Big(\big|\hat{\mathrm{d}}(x_i, x_{\mathrm{new}}) - \mathrm{d}(x_i, x_{\mathrm{new}})\big| > \varepsilon\Big) < h^{(l)}(\varepsilon) \quad \textit{for all} \quad x_i \in \Gamma^{l+1} \quad \textit{and} \quad l \in \mathbb{N} \cup \{0\}. \quad (12)$$

We note that Proposition 4 demonstrates that the coefficient of exponential terms in (12) grow exponentially with increasing layers (i.e., $m^l$ and $m^{2l}$ in (11)), which requires a commensurately large number of samples to control the tail probabilities.

**Theorem 1.** *Under Assumptions 1–5, RRG learns the correct latent tree with probability $1 - \eta$ if*

$$n_2 = \tilde{\Omega}\Big(\frac{l_{\mathrm{max}}^4 e^{2\rho_{\mathrm{max}}/l_{\mathrm{max}}} \kappa^2}{\delta_{\mathrm{min}}^{2/l_{\mathrm{max}}} \rho_{\mathrm{min}}^2} \Big(\frac{9}{2}\Big)^{2L_{\mathrm{R}}} \log \frac{|\mathcal{V}_{\mathrm{obs}}|^3}{\eta}\Big) \quad \textit{and} \quad n_1 = O\Big(\frac{\sqrt{n_2}}{\log n_2}\Big), \quad (13)$$

*where $L_{\mathrm{R}}$ is the number of iterations of RRG needed to construct the tree.*

Theorem 1 indicates that the number of clean samples $n_2$ required by RRG to learn the correct structure grows exponentially with the number of iterations $L_{\mathrm{R}}$. Specifically, for the full $m$-tree illustrated in Fig. 5, $n_2$ is exponential in the depth of the tree with high probability for structure learning to succeed. The sample complexity of RRG depends on $e^{2\rho_{\mathrm{max}}/l_{\mathrm{max}}}$, and the exponential relationship with $\rho_{\mathrm{max}}$ will be shown to be unavoidable in view of our impossibility result in Theorem 5. Huang et al. [19, Lemma 7.2 ] also derived a sample complexity result for learning latent trees but the algorithm is based on [5] instead of RG. RRG is able to tolerate $n_1 = O(\sqrt{n_2}/\log n_2)$ corruptions. This tolerance level originates from the properties of the truncated inner product; similar tolerances will also be seen for the sample complexities of subsequent algorithms. We expect this is also the case for [19], which is based on [5], though we have not shown this formally. In addition, the sample complexity is applicable to a wide class of graphical models that satisfies the Assumptions 1 to 5, while the sample complexity result [4, Theorem 11], which hides the dependencies on the parameters, only holds for a limited class of graphical models whose effective depths (the maximal length of paths between hidden nodes and their closest observed nodes) are bounded in $|\mathcal{V}_{\mathrm{obs}}|$.

### 3.3 Robust Neighbor Joining and Spectral Neighbor Joining algorithms

The NJ algorithm [1] also makes use of additive distances to identify the existence of hidden nodes. To robustify the NJ algorithm, we adopt robust estimates of information distances as the additive distances in the so-called RNJ algorithm. We first recap a result by Atteson [20].

**Proposition 5.** *If all the nodes have exactly two children, NJ will output the correct latent tree if*

$$\max_{x_i, x_j \in \mathcal{V}_{\mathrm{obs}}} \big|\hat{\mathrm{d}}(x_i, x_j) - \mathrm{d}(x_i, x_j)\big| \leq \rho_{\mathrm{min}}/2. \quad (14)$$

Unlike RG, NJ does not identify the parent relationship among nodes, so it is only applicable to binary trees in which each node has at most two children.

**Theorem 2.** *If Assumptions 1–5 hold and all the nodes have exactly two children, RNJ constructs the correct latent tree with probability at least $1 - \eta$ if*

$$n_2 = \Omega\Big(\frac{l_{\mathrm{max}}^4 e^{2\rho_{\mathrm{max}}/l_{\mathrm{max}}} \kappa^2}{\delta_{\mathrm{min}}^{2/l_{\mathrm{max}}} \rho_{\mathrm{min}}^2} \log \frac{|\mathcal{V}_{\mathrm{obs}}|^2}{\eta}\Big) \quad \textit{and} \quad n_1 = O\Big(\frac{\sqrt{n_2}}{\log n_2}\Big). \quad (15)$$

Theorem 2 indicates that the sample complexity of RNJ grows as $\log |\mathcal{V}_{\mathrm{obs}}|$, which is much better than RRG. Similarly to RRG, the sample complexity has an exponential dependence on $\rho_{\mathrm{max}}$.

In recent years, several variants of NJ algorithm have been proposed. The additivity of information distances results in certain properties of the rank of the matrix $\mathbf{R} \in \mathbb{R}^{|\mathcal{V}_{\mathrm{obs}}| \times |\mathcal{V}_{\mathrm{obs}}|}$, where $\mathbf{R}(i,j) = \exp(-\mathrm{d}(x_i, x_j))$ for all $x_i, x_j \in \mathcal{V}_{\mathrm{obs}}$. Jaffe *et al.* [6] proposed SNJ which utilizes the rank of $\mathbf{R}$ to deduce the sibling relationships among nodes. We robustify the SNJ algorithm by implementing the robust estimation of information distances, as shown in Algorithm 2.

Although SNJ was designed for discrete random variables, the additivity of the information distance proved in Proposition 1 guarantees the consistency of Robust Spectral NJ (RSNJ) for GGMs with vector variables. A sufficient condition for RSNJ to learn the correct tree can be generalized from [6].

**Proposition 6.** *If Assumptions 1–5 hold and all the nodes have exactly two children, a sufficient condition for RSNJ to recover the correct tree from $\hat{\mathbf{R}}$ is*

$$\|\hat{\mathbf{R}} - \mathbf{R}\|_2 \leq g(|\mathcal{V}_{\text{obs}}|, \rho_{\min}, \rho_{\max}), \tag{16}$$

*where*

$$g(x, \rho_{\min}, \rho_{\max}) = \begin{cases} \frac{1}{2}(2e^{-\rho_{\max}})^{\log_2(x/2)}e^{-\rho_{\max}}(1 - e^{-2\rho_{\min}}), & e^{-2\rho_{\max}} \leq 0.5 \\ e^{-3\rho_{\max}}(1 - e^{-2\rho_{\min}}), & e^{-2\rho_{\max}} > 0.5 \end{cases}.$$

Similar with RNJ, RSNJ also does not identify the parent relationship between nodes, so it only applies to binary trees. To state the next result succinctly, we assume that $\rho_{\max} \geq \frac{1}{2}\log 2$; this is the regime of interest because we consider large trees which implies that $\rho_{\max}$ is typically large.

**Theorem 3.** *If Assumptions 1–5 hold, $\rho_{\max} \geq \frac{1}{2}\log 2$, and all the nodes have exactly two children, RSNJ reconstructs the correct latent tree with probability at least $1 - \eta$ if*

$$n_2 = \Omega\Big(\frac{l_{\max}^4 e^{2\rho_{\max}(1/l_{\max} + \log_2(|\mathcal{V}_{\text{obs}}|/2) + 1)}\kappa^2}{\delta_{\min}^{2/l_{\max}}e^{2\rho_{\min}}}\log\frac{|\mathcal{V}_{\text{obs}}|^2}{\eta}\Big) \quad and \quad n_1 = O\Big(\frac{\sqrt{n_2}}{\log n_2}\Big). \tag{17}$$

Theorem 3 indicates that the sample complexity of RSNJ grows as $\text{poly}(|\mathcal{V}_{\text{obs}}|)$. Specifically, in the binary tree case, the sample complexity grows exponentially with the depth of the tree. Also, the dependence of sample complexity on $\rho_{\max}$ is exponential, i.e., $O\big(e^{2(1/l_{\max} + \log_2(|\mathcal{V}_{\text{obs}}|/2) + 1)\rho_{\max}}\big)$, but the coefficient of $\rho_{\max}$ is larger than those of RRG and RNJ, which are $O\big(e^{2\rho_{\max}/l_{\max}}\big)$. Compared to the sample complexity of SNJ in [6], the sample complexity of RSNJ has the same dependence on the number of observed nodes $|\mathcal{V}_{\text{obs}}|$, which means that the robustification of SNJ using the truncated inner product is able to tolerate $O\big(\frac{\sqrt{n_2}}{\log n_2}\big)$ corruptions.

### 3.4 Robust Chow-Liu Recursive Grouping

In this section, we show that the exponential dependence on $L_R$ in Theorem 1 can be provably mitigated with an accurate initialization of the structure. Different from RRG, RCLRG takes Chow-Liu algorithm as the initialization stage, as shown in Algorithm 3. The Chow-Liu algorithm [14] learns the maximum likelihood estimate of the tree structure by finding the maximum weight spanning tree of the graph whose edge weights are the mutual information quantities between these variables. In the estimation of the hidden tree structure, instead of taking the mutual information as the weights, we find the minimum spanning tree (MST) of the graph whose weights are information distances, i.e.,

$$\text{MST}(\mathcal{V}_{\text{obs}}; \mathbf{D}) := \underset{\mathbb{T} \in \mathcal{T}_{\mathcal{V}_{\text{obs}}}}{\arg\min} \sum_{(x_i, x_j) \in \mathbb{T}} d(x_i, x_j), \tag{18}$$

where $\mathcal{T}_{\mathcal{V}_{\text{obs}}}$ is the set of all the trees with node set $\mathcal{V}_{\text{obs}}$. To describe the process of finding the MST, we recall the definition of the *surrogate node* from [4].

**Definition 3.** *Given the latent tree $\mathbb{T} = (\mathcal{V}, \mathcal{E})$ and any node $x_i \in \mathcal{V}$, the surrogate node [4] of $x_i$ is $\text{Sg}(x_i; \mathbb{T}, \mathcal{V}_{\text{obs}}) = \arg\min_{x_j \in \mathcal{V}_{\text{obs}}} d(x_i, x_j)$.*

We introduce a new notion of distance that quantifies the sample complexity of RCLRG.

**Definition 4.** *Given the latent tree $\mathbb{T} = (\mathcal{V}, \mathcal{E})$ and any node $x_i \in \mathcal{V}$, the* contrastive distance *of $x_i$ with respect to $\mathcal{V}_{\text{obs}}$ is defined as*

$$d_{\text{ct}}(x_i; \mathbb{T}, \mathcal{V}_{\text{obs}}) = \min_{x_j \in \mathcal{V}_{\text{obs}} \setminus \{\text{Sg}(x_i; \mathbb{T}, \mathcal{V}_{\text{obs}})\}} d(x_i, x_j) - \min_{x_j \in \mathcal{V}_{\text{obs}}} d(x_i, x_j). \tag{19}$$

Definitions 3 and 4 imply that the surrogate node $\text{Sg}(x_i; \mathbb{T}, \mathcal{V}_{\text{obs}})$ of any observed node $x_i$ is itself $x_i$, and its contrastive distance is the information distance between the closest observed node and itself. It is shown that the Chow-Liu tree $\text{MST}(\mathcal{V}_{\text{obs}}; \mathbf{D})$ is equal to the tree where all the hidden nodes are contracted to their surrogate nodes [4], so it will be difficult to identify the surrogate node of some node if its contrastive distance is small. Under this scenario, more accurate estimates of the information distances are required to construct the correct Chow-Liu tree $\text{MST}(\mathcal{V}_{\text{obs}}; \mathbf{D})$.

**Proposition 7.** *The Chow-Liu tree* $\mathrm{MST}(\mathcal{V}_{\mathrm{obs}}; \hat{\mathbf{D}})$ *is constructed correctly if*

$$\left|\hat{\mathrm{d}}(x_i, x_j) - \mathrm{d}(x_i, x_j)\right| < \Delta_{\mathrm{MST}}/2 \quad \textit{for all} \quad x_i, x_j \in \mathcal{V}_{\mathrm{obs}}, \tag{20}$$

*where* $\Delta_{\mathrm{MST}} := \min_{x_j \in \mathrm{Int}(\mathbb{T})} \mathrm{d}_{\mathrm{ct}}(x_j; \mathbb{T}, \mathcal{V}_{\mathrm{obs}})$.

Hence, the contrastive distance describes the difficulty of learning the correct Chow-Liu tree.

**Theorem 4.** *With Assumptions 1–5, RCLRG constructs the correct latent tree with probability at least* $1 - \eta$ *if*

$$n_2 = \tilde{\Omega}\left(\max\left\{\frac{1}{\rho_{\min}^2}\left(\frac{9}{2}\right)^{2L_{\mathrm{C}}}, \frac{1}{\Delta_{\mathrm{MST}}^2}\right\} \frac{l_{\max}^4 e^{2\rho_{\max}/l_{\max}} \kappa^2}{\delta_{\min}^{2/l_{\max}}} \log \frac{|\mathcal{V}_{\mathrm{obs}}|^3}{\eta}\right) \quad \textit{and} \quad n_1 = O\left(\frac{\sqrt{n_2}}{\log n_2}\right), \tag{21}$$

*where* $L_{\mathrm{C}}$ *is the maximum number of iterations of RRG (over each internal node of the constructed Chow-Liu tree) in RCLRG needed to construct the tree.*

If we implement RCLRG with *true* information distances, $L_{\mathrm{C}} \leq \lceil\frac{1}{2}\mathrm{Deg}(\mathrm{MST}(\mathcal{V}_{\mathrm{obs}}; \hat{\mathbf{D}})) - 1\rceil$. Theorem 4 indicates that the sample complexity of RCLRG grows exponentially in $L_{\mathrm{C}} \ll L_{\mathrm{R}}$. Compared with [4, Theorem 12], the sample complexity of RCLRG in Theorem 4 is applicable to a wide class of graphical models that satisfy Assumptions 1 to 5, while the [4, Theorem 12] requires the assumption that the effective depths of latent trees are *bounded* in $|\mathcal{V}_{\mathrm{obs}}|$, which is rather restrictive.

### 3.5 Comparison of robust latent tree learning algorithms

Since the sample complexities of RRG, RCLRG, RSNJ and RNJ depend on different parameters and different structures of the underlying graphs, it is instructive to compare the sample complexities of these algorithms on some representative tree structures. These trees are illustrated in Fig. 5. RSNJ and RNJ are not able to identify the parent relationship among nodes, so they are only applicable to trees whose maximal degrees are no larger that 3, including the double-binary tree and the HMM. In particular, RNJ and RSNJ are not applicable to the full $m$-tree (for $m \geq 3$) and the double star. Derivations and more detailed discussions of the sample complexities are deferred to Appendix K.

| $n_2$ ╲ Algorithm  Tree | RRG | RCLRG | RSNJ | RNJ |
|---|---|---|---|---|
| Double-binary tree | $O\big(\psi(\frac{9}{2})^{\mathrm{Diam}(\mathbb{T})}\big)$ | $O\big(\psi(\frac{9}{2})^{\frac{1}{2}\mathrm{Diam}(\mathbb{T})}\big)$ | $O\big(e^{2t\rho_{\max}}\mathrm{Diam}(\mathbb{T})\big)$ | $O\big(\psi\mathrm{Diam}(\mathbb{T})\big)$ |
| HMM | $O\big(\psi(\frac{9}{2})^{\mathrm{Diam}(\mathbb{T})}\big)$ | $O\big(\psi\log\mathrm{Diam}(\mathbb{T})\big)$ | $O\big(e^{2t\rho_{\max}}\log\mathrm{Diam}(\mathbb{T})\big)$ | $O\big(\psi\log\mathrm{Diam}(\mathbb{T})\big)$ |
| Full $m$-tree | $O\big(\psi(\frac{9}{2})^{\mathrm{Diam}(\mathbb{T})}\big)$ | $O\big(\psi\mathrm{Diam}(\mathbb{T})\big)$ | N.A. | N.A. |
| Double star | $O\big(\psi\log d_{\max}\big)$ | $O\big(\psi\log d_{\max}\big)$ | N.A. | N.A. |

Table 1: The sample complexities of RRG, RCLRG, RSNJ and RNJ on the double-binary tree, the HMM, the full $m$-tree and the double star. We set $\psi := e^{2\rho_{\max}/l_{\max}}$ and $t = O(l_{\max}^{-1} + \log|\mathcal{V}_{\mathrm{obs}}|)$.

### 3.6 Experimental results

We present simulation results to demonstrate the efficacy of the robustified algorithms. Samples are generated from a HMM with $l_{\max} = 3$ and $\mathrm{Diam}(\mathbb{T}) = 80$. The Robinson-Foulds distance [21] between the true and estimated trees is adopted to measure the performances of the algorithms. For the implementations of CLRG and RG, we use the code from [4]. Other settings and more extensive experiments are given in Appendix L.

We consider three corruption patterns here. (i) *Uniform corruptions* are independent additive noises in $[-2A, 2A]$; (ii) *Constant magnitude corruptions* are also independent additive noises but taking values in $\{-A, +A\}$ with probability 0.5. These two types of noises are distributed randomly in $\mathbf{X}_1^n$; (iii) *HMM corruptions* are generated by a HMM which has the same structure as the original HMM but has different parameters. They replace the entries in $\mathbf{X}_1^n$ with samples generated by the variables in the same positions. In our simulations, $A$ is set to 60, and the number of corruptions $n_1$ is 100.

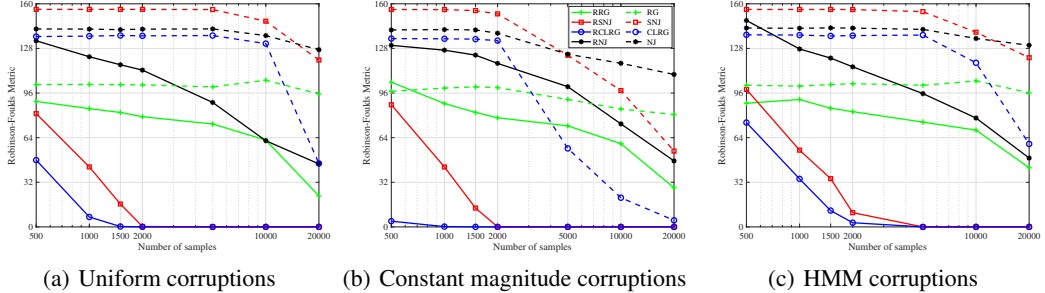

| (a) Uniform corruptions | (b) Constant magnitude corruptions | (c) HMM corruptions |

Figure 2: Robinson-Foulds distances of robustified and original algorithms averaged over 100 trials

Fig. 2 (error bars are in Appendix L.1) demonstrates the superiority of RCLRG in learning HMMs compared to other algorithms. The robustified algorithms also result in smaller estimation errors (Robinson-Foulds distances) compared to their unrobustified counterparts in presence of corruptions.

## 4 Impossibility result

**Definition 5.** *Given a triple* $(|\mathcal{V}_{\mathrm{obs}}|, \rho_{\max}, l_{\max})$, *the set* $\mathcal{T}(|\mathcal{V}_{\mathrm{obs}}|, \rho_{\max}, l_{\max})$ *consists of all multivariate Gaussian distributions* $\mathcal{N}(\mathbf{0}, \mathbf{\Sigma})$ *such that: (1) The underlying graph* $\mathbb{T} = (\mathcal{V}, \mathcal{E})$ *is a tree* $\mathbb{T} \in \mathcal{T}_{\geq 3}$, *and the size of the set of observed nodes is* $|\mathcal{V}_{\mathrm{obs}}|$. *(2) The distribution* $\mathcal{N}(\mathbf{0}, \mathbf{\Sigma})$ *satisfies Assumptions 1 and 5 with parameters* $l_{\max}$ *and* $\rho_{\max}$.

For the given class of graphical models $\mathcal{T}(|\mathcal{V}_{\mathrm{obs}}|, \rho_{\max}, l_{\max})$, nature chooses some parameter $\theta = \mathbf{\Sigma}$ and generates $n$ i.i.d. samples $\mathbf{X}_1^n$ from $\mathbb{P}_\theta$. The goal of the statistician is to use the observations $\mathbf{X}_1^n$ to learn the underlying graph $\mathbb{T}$, which entails the design of a *decoder* $\phi : \mathbb{R}^{n \times |\mathcal{V}_{\mathrm{obs}}| l_{\max}} \to \mathcal{T}_{|\mathcal{V}_{\mathrm{obs}}|}$, where $\mathcal{T}_{|\mathcal{V}_{\mathrm{obs}}|}$ is the set of latent trees whose size of the observed node set is $|\mathcal{V}_{\mathrm{obs}}|$.

**Theorem 5.** *Consider the class of graphical models* $\mathcal{T}(|\mathcal{V}_{\mathrm{obs}}|, \rho_{\max}, l_{\max})$, *where* $|\mathcal{V}_{\mathrm{obs}}| \geq 3$. *If there exists a graph decoder learns from* $n$ *i.i.d. samples such that*

$$\max_{\theta(\mathbb{T}) \in \mathcal{T}(|\mathcal{V}_{\mathrm{obs}}|, \rho_{\max}, l_{\max})} \mathbb{P}_{\theta(\mathbb{T})}(\phi(\mathbf{X}_1^n) \neq \mathbb{T}) < \delta, \tag{22}$$

*then (as* $\rho_{\max} \to \infty$ *and* $|\mathcal{V}_{\mathrm{obs}}| \to \infty$),

$$n = \max\left\{\Omega\left((1-\delta)e^{\frac{\rho_{\max}}{\lceil \log_3 |\mathcal{V}_{\mathrm{obs}}| \rceil l_{\max}}} \log |\mathcal{V}_{\mathrm{obs}}|\right), \Omega\left((1-\delta)e^{\frac{2\rho_{\max}}{3 l_{\max}}}\right)\right\}. \tag{23}$$

Theorem 5 implies that the optimal sample complexity grows as $\Omega(\log |\mathcal{V}_{\mathrm{obs}}|)$ as $|\mathcal{V}_{\mathrm{obs}}|$ grows. To prove this theorem, we construct several classes of Gaussian latent trees parametrized as linear dynamical systems (see Appendix M) and apply the ubiquitous Fano technique to derive the desired impossibility result. Table 1 indicates that the sample complexity of RCLRG when the underlying latent tree is a full $m$-tree (for $m \geq 3$) or a HMM is optimal in the dependence on $|\mathcal{V}_{\mathrm{obs}}|$. The sample complexity of RNJ is also optimal in $|\mathcal{V}_{\mathrm{obs}}|$ for double binary trees and HMMs. In contrast, the derived sample complexities of RRG and RSNJ are suboptimal in relation to Theorem 5. However, one caveat of our analyses of the latent tree learning algorithms in Section 3 is that we are not claiming that they are the best possible for the given algorithm; there may be room for improvement.

When the maximum information distance $\rho_{\max}$ grows, Theorem 5 indicates that the optimal sample complexity grows as $\Omega(e^{\frac{2\rho_{\max}}{3 l_{\max}}})$. Table 1 shows the sample complexities of RRG, RCLRG and RNJ grow as $O(e^{2\frac{\rho_{\max}}{l_{\max}}})$, which has the alike dependence as the impossibility result. However, the sample complexity of RSNJ grows as $O(e^{2t\rho_{\max}})$, which is larger (looser) than that prescribed by Theorem 5.

## 5 Conclusions and future works

In this paper, we first derived the more refined sample complexities of RG and CLRG. The effectiveness of CLRG was observed to be due to the reduction in the effective length that the error propagates,

i.e., from $L_\mathrm{R}$ to $L_\mathrm{C} \ll L_\mathrm{R}$. Second, to combat potential adversarial corruptions in the data matrix, we robustified RG, CLRG, NJ and SNJ by adopting the truncated inner product technique. The derived sample complexity results showed that all the common latent tree learning algorithms can tolerate level-$O\big(\frac{\sqrt{n_2}}{\log n_2}\big)$ arbitrary corruptions. The varying efficacies of these robustified algorithms were then corroborated through extensive simulations with different types of corruptions and on different graphs. Finally, we derived the first known instance-dependent impossibility result for learning latent trees. The optimalities of RCLRG and RNJ in their dependencies on $|\mathcal{V}_\mathrm{obs}|$ were also discussed in the context of various latent tree structures.

There are several promising avenues for future research. First, the design and analysis of the initialization process of CLRG can be further improved. The correctness of CLRG relies only on the fact that if a hidden node is contracted to an observed node, then *all* the hidden nodes on the path between the hidden node and the observed nodes are contracted to the *same* observed node. One can conceive of a more general initialization algorithm other than that using the MST of the weighted graph with weights being the information distances. Second, the analysis of RG can be tightened with more sophisticated concentration bounds. In particular, the exponential behavior of the sample complexity of RG can also refined by performing a more careful analysis of the error propagation through the learned tree.

**Acknowledgements**    We would like to thank the NeurIPS reviewers for their valuable and detailed reviews. This work is supported by a National University of Singapore (NUS) President's Graduate Fellowship, Singapore National Research Foundation (NRF) Fellowship (R-263-000-D02-281), a Singapore Ministry of Education (MoE) AcRF Tier 1 Grant (R-263-000-E80-114), and a Singapore MoE AcRF Tier 2 Grant (R-263-000-C83-112).

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
