# Supplementary materials for the NeurIPS 2021 submission "Robustifying Algorithms of Learning Latent Trees with Vector Variables"

## A    Illustrations of corruption patterns in Section 2.1

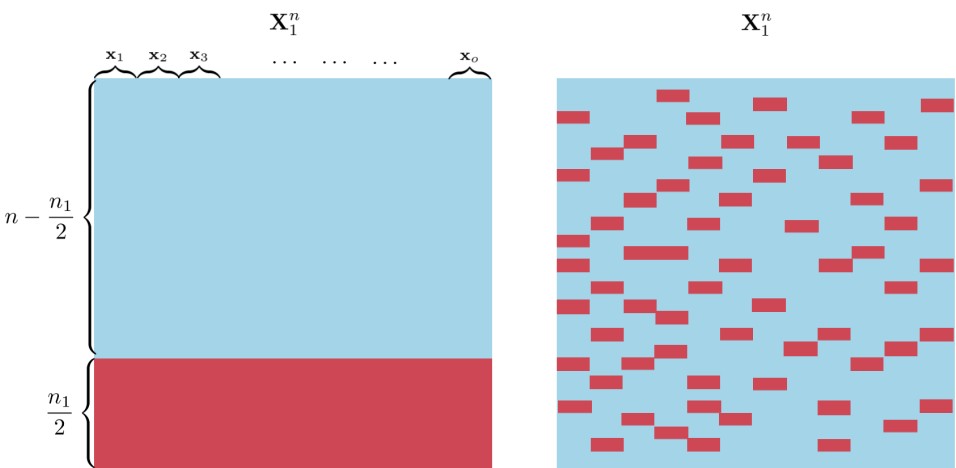

Figure 3: The left figure shows the corruption pattern that corrupted terms lie *in the same rows*. This corruption pattern is known as *outliers*. The right figure shows an *arbitrary corruption pattern* where corrupted entries in each column can be in *any $n_1/2$ rows*.

## B    Illustrations of active sets defined in Section 3.2

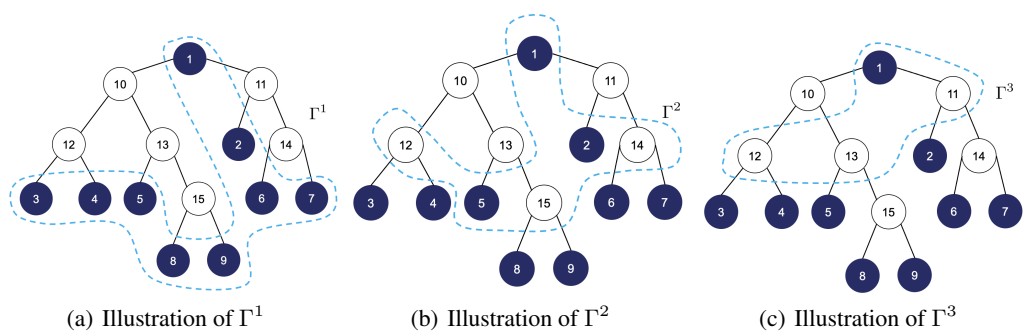

(a) Illustration of $\Gamma^1$    (b) Illustration of $\Gamma^2$    (c) Illustration of $\Gamma^3$

Figure 4: Illustration of active sets.

# C    Pseudo-code of RRG in Section 3.2

---

**Algorithm 1** RRG

---

**Input:** Data matrix $\mathbf{X}$, corruption level $n_1$, threshold $\varepsilon$
**Output:** Adjacency matrix $\mathbf{A}$
**Procedure:**

1: Active set $\Gamma^1 \leftarrow$ all the observed nodes
2: Implement truncated inner product to compute $\hat{\mathrm{d}}(x_i, x_j)$ for all $x_i, x_j \in \mathcal{V}_{\mathrm{obs}}$.
3: **while** $|\Gamma^i| > 2$ **do**
4:     Update $\hat{\mathrm{d}}(x_{\mathrm{new}}, x_i)$ for all $x_i \in \Gamma^i$ for all new hidden nodes.
5:     Compute $\hat{\Phi}_{ijk} = \hat{\mathrm{d}}(x_i, x_k) - \hat{\mathrm{d}}(x_j, x_k)$ for all $x_i, x_j, x_k \in \Gamma^i$
6:     **for** all nodes $x_i$ and $x_j$ in $\Gamma^i$ **do**
7:         **if** $|\hat{\Phi}_{ijk} - \hat{\Phi}_{ijk'}| < \varepsilon$ for all $x_k, x_{k'} \in \Gamma^i$ **then**
8:             **if** $|\hat{\Phi}_{ijk} - \hat{\mathrm{d}}(x_i, x_j)| < \varepsilon$ for all $x_k \in \Gamma^i$ **then**
9:                 $x_j$ is the parent of $x_i$.
10:                Eliminate $x_i$ from $\Gamma^i$
11:            **else**
12:                $x_j$ and $x_i$ are siblings.
13:                Create a hidden node $x_{\mathrm{new}}$ as the parent of $x_j$ and $x_i$
14:                Add $x_{\mathrm{new}}$ and eliminate $x_j$ and $x_i$ from $\Gamma^i$
15:            **end if**
16:        **end if**
17:    **end for**
18: **end while**

---

# D    Pseudo-code of RSNJ in Section 3.3

---

**Algorithm 2** RSNJ

---

**Input:** Data matrix $\mathbf{X}$, corruption level $n_1$
**Output:** Adjacent matrix $\mathbf{A}$
**Procedure:**

1: Implement truncated inner product to compute $\hat{\mathrm{d}}(x_i, x_j)$ for all $x_i, x_j \in \mathcal{V}_{\mathrm{obs}}$.
2: Compute the symmetric affinity matrix $\hat{\mathbf{R}}$ as $\hat{\mathbf{R}}(i, j) = \exp(-\hat{\mathrm{d}}(x_i, x_j))$ for all $x_i, x_j \in \mathcal{V}_{\mathrm{obs}}$
3: Set $B_i = \{x_i\}$ for all $x_i \in \Omega$
4: Compute the matrix $\mathbf{S}$ as $\hat{\mathbf{S}}(i, j) = \sigma_2(\hat{\mathbf{R}}^{B_i \cup B_j})$
5: **while** The number of $B_i$'s is larger than 3 **do**
6:     Find $(\hat{i}, \hat{j}) = \arg\min_{i,j} \hat{\mathbf{S}}(i, j)$.
7:     Merge $B_{\hat{i}}$ and $B_{\hat{j}}$ as $B_{\hat{i}} = B_{\hat{i}} \cup B_{\hat{j}}$ and delete $B_{\hat{j}}$.
8:     Update $\hat{\mathbf{S}}(k, \hat{i}) = \sigma_2(\hat{\mathbf{R}}^{B_k \cup B_{\hat{i}}})$.
9: **end while**

---

# E    Pseudo-code of RCLRG in Section 3.4

---
**Algorithm 3** RCLRG

---
**Input:** Data matrix $\mathbf{X}$, corruption level $n_1$, threshold $\varepsilon$
**Output:** Adjacency matrix $\mathbf{A}$
**Procedure:**
 1: Construct a Chow-Liu tree with $\hat{\mathrm{d}}(x_j, x_k)$ for observed nodes $x_j, x_k \in \mathcal{V}_{\mathrm{obs}}$
 2: Identify the set of internal nodes of the Chow-Liu tree
 3: **for** all internal nodes $x_i$ of the Chow-Liu tree **do**
 4:     Implement RRG algorithm on the closed neighborhood of $x_i$
 5:     Replace the closed neighborhood of $x_i$ with the output of RRG
 6: **end for**

---

# F    Illustrations of representative trees in Section 3.5

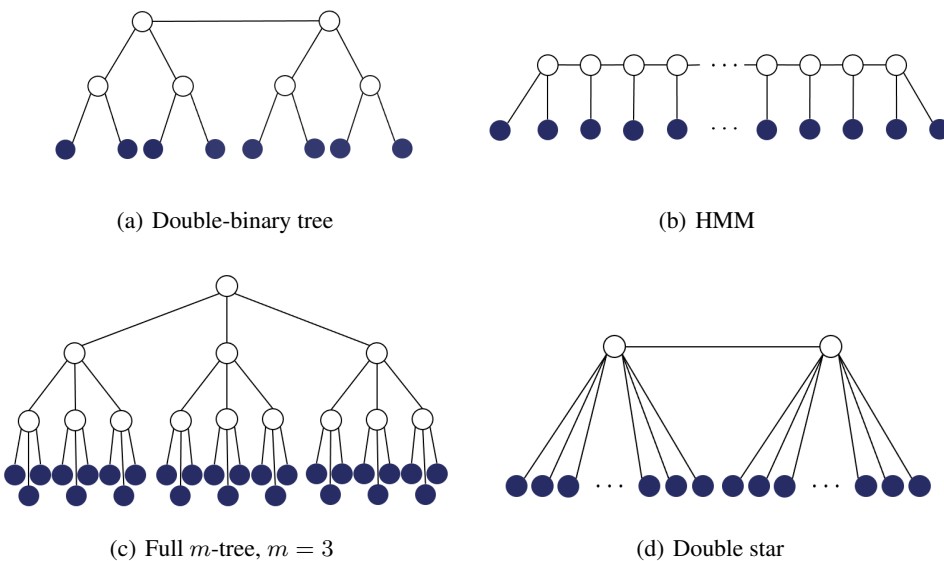

(a) Double-binary tree

(b) HMM

(c) Full $m$-tree, $m = 3$

(d) Double star

Figure 5: Representative tree structures.

# G    Proofs of results in Section 3.1

*Proof of Proposition 1.* For the sake of brevity, we prove the additivity property for paths of length 2. The proof for the general cases can be derived similarly. We consider the case $x_j$ is on the path connected $x_i$ and $x_k$ and $x_i, x_j, x_k \in \mathcal{V}$.

For any square matrix $\mathbf{A} \in \mathbb{R}^{n \times n}$, the determinant of $\mathbf{A}$ is denoted as $|\mathbf{A}| = \det(\mathbf{A})$.

Then we can write information distance as

$$\mathrm{d}(x_i, x_k) = -\frac{1}{2} \log \left| \boldsymbol{\Sigma}_{ik} \boldsymbol{\Sigma}_{ik}^{\top} \right| + \frac{1}{4} \log \left| \boldsymbol{\Sigma}_{ii} \boldsymbol{\Sigma}_{ii}^{\top} \right| + \frac{1}{4} \log \left| \boldsymbol{\Sigma}_{kk} \boldsymbol{\Sigma}_{kk}^{\top} \right| \tag{G.1}$$

Note that $\mathbb{E}[\mathbf{x}_i | \mathbf{x}_j] = \boldsymbol{\Sigma}_{ij} \boldsymbol{\Sigma}_{jj}^{-1} \mathbf{x}_j$ and $\boldsymbol{\Sigma}_{ij}$ is of full rank by Assumption 2, and

$$\mathbf{A}_{i|j} = \boldsymbol{\Sigma}_{ij} \boldsymbol{\Sigma}_{jj}^{-1} \tag{G.2}$$

is also of full rank.

Furthermore, we have

$$\boldsymbol{\Sigma}_{ik} = \mathbf{A}_{i|j} \boldsymbol{\Sigma}_{jj} \mathbf{A}_{k|j}^{\top} \quad \text{and} \quad \boldsymbol{\Sigma}_{ik} \boldsymbol{\Sigma}_{ik}^{\top} = \mathbf{A}_{i|j} \boldsymbol{\Sigma}_{jj} \mathbf{A}_{k|j}^{\top} \mathbf{A}_{k|j} \boldsymbol{\Sigma}_{jj} \mathbf{A}_{i|j}^{\top}. \tag{G.3}$$

Then we have

$$\left|\boldsymbol{\Sigma}_{ik}\boldsymbol{\Sigma}_{ik}^{\top}\right| = \left|\mathbf{A}_{i|j}\boldsymbol{\Sigma}_{jj}\mathbf{A}_{k|j}^{\top}\mathbf{A}_{k|j}\boldsymbol{\Sigma}_{jj}\mathbf{A}_{i|j}^{\top}\right| \tag{G.4}$$

$$= \left|\mathbf{A}_{i|j}^{\top}\mathbf{A}_{i|j}\boldsymbol{\Sigma}_{jj}\mathbf{A}_{k|j}^{\top}\mathbf{A}_{k|j}\boldsymbol{\Sigma}_{jj}\right| \tag{G.5}$$

$$= \frac{\left|\boldsymbol{\Sigma}_{jj}^{\top}\mathbf{A}_{i|j}^{\top}\mathbf{A}_{i|j}\boldsymbol{\Sigma}_{jj}\right|}{\left|\boldsymbol{\Sigma}_{jj}\right|}\frac{\left|\boldsymbol{\Sigma}_{jj}^{\top}\mathbf{A}_{k|j}^{\top}\mathbf{A}_{k|j}\boldsymbol{\Sigma}_{jj}\right|}{\left|\boldsymbol{\Sigma}_{jj}\right|}. \tag{G.6}$$

Furthermore,

$$\left|\boldsymbol{\Sigma}_{jj}^{\top}\mathbf{A}_{i|j}^{\top}\mathbf{A}_{i|j}\boldsymbol{\Sigma}_{jj}\right| = \left|\mathbf{A}_{i|j}\boldsymbol{\Sigma}_{jj}\boldsymbol{\Sigma}_{jj}^{\top}\mathbf{A}_{i|j}^{\top}\right| = \left|\boldsymbol{\Sigma}_{ij}\boldsymbol{\Sigma}_{ij}^{\top}\right|, \tag{G.7}$$

$$\left|\boldsymbol{\Sigma}_{jj}^{\top}\mathbf{A}_{k|j}^{\top}\mathbf{A}_{k|j}\boldsymbol{\Sigma}_{jj}\right| = \left|\boldsymbol{\Sigma}_{kj}\boldsymbol{\Sigma}_{kj}^{\top}\right|. \tag{G.8}$$

Substituting (G.4) and (G.7) into (G.1), we have

$$\mathrm{d}(x_i, x_k) = -\frac{1}{2}\log\left|\boldsymbol{\Sigma}_{ij}\boldsymbol{\Sigma}_{ij}^{\top}\right| + \frac{1}{4}\log\left|\boldsymbol{\Sigma}_{ii}\boldsymbol{\Sigma}_{ii}^{\top}\right| + \frac{1}{4}\log\left|\boldsymbol{\Sigma}_{jj}\boldsymbol{\Sigma}_{jj}^{\top}\right|$$
$$- \frac{1}{2}\log\left|\boldsymbol{\Sigma}_{kj}\boldsymbol{\Sigma}_{kj}^{\top}\right| + \frac{1}{4}\log\left|\boldsymbol{\Sigma}_{kk}\boldsymbol{\Sigma}_{kk}^{\top}\right| + \frac{1}{4}\log\left|\boldsymbol{\Sigma}_{jj}\boldsymbol{\Sigma}_{jj}^{\top}\right| \tag{G.9}$$
$$= \mathrm{d}(x_i, x_j) + \mathrm{d}(x_j, x_k), \tag{G.10}$$

as desired. □

**Lemma 8.** *(Bernstein-type inequality [22]) Let $X_1, \ldots, X_n$ be $n$ centered sub-exponential random variables, and $K = \max_{1 \le i \le n} \|X_i\|_{\psi_1}$, where $\|\cdot\|_{\psi_1}$ is the sub-exponential norm and is defined as*

$$\|X\|_{\psi_1} := \sup_{p \ge 1} p^{-1}\left(\mathbb{E}|X|^p\right)^{1/p}. \tag{G.11}$$

*Then for every $a = (a_1, \ldots, a_n) \in \mathbb{R}^n$ and every $t > 0$, we have*

$$\mathbb{P}\left(\left|\sum_{i=1}^{n} a_i X_i\right| \ge t\right) \le 2\exp\left[-c\min\left\{\frac{t^2}{K^2\|a\|_2^2}, \frac{t}{K\|a\|_\infty}\right\}\right] \tag{G.12}$$

**Lemma 9.** *Let the estimate of the covariance matrix $\boldsymbol{\Sigma}_{ij}$ based on the truncated inner product be $\hat{\boldsymbol{\Sigma}}_{ij}$. If $t_2 < \kappa = \max\{\sigma_{\max}^2, \rho_{\min}\}$, we have*

$$\mathbb{P}\left(\|\hat{\boldsymbol{\Sigma}}_{ij} - \boldsymbol{\Sigma}_{ij}\|_{\infty,\infty} > t_1 + t_2\right) \le 2l_{\max}^2 e^{-\frac{3n_2}{16\kappa n_1}t_1} + l_{\max}^2 e^{-c\frac{t_2^2 n_2}{\kappa^2}} \quad \forall x_i, x_j \in \mathcal{V}_{\mathrm{obs}}. \tag{G.13}$$

*Proof of Lemma 9.* Let $I_{ij,1}^{st}$ be the set of indexes of the uncorrupted samples of $[\mathbf{x}_i]_s[\mathbf{x}_j]_t$. Without loss of generality, we assume that $|I_{ij,1}^{st}| = n_2$. Let $I_{ij,2}^{st}$ and $I_{ij,3}^{st}$ be the sets of the indexes of truncated uncorrupted samples and the reserved corrupted samples, respectively.

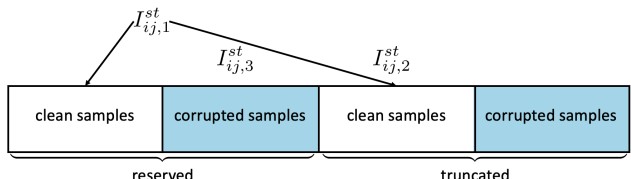

Figure 6: Illustration of the truncated inner product.

Then,

$$[\hat{\boldsymbol{\Sigma}}_{ij}]_{st} = \frac{1}{n_2}\left(\sum_{m \in I_{ij,1}^{st}} [\mathbf{x}_i]_s^{(m)}[\mathbf{x}_j]_t^{(m)} - \sum_{m \in I_{ij,2}^{st}} [\mathbf{x}_i]_s^{(m)}[\mathbf{x}_j]_t^{(m)} + \sum_{m \in I_{ij,3}^{st}} [\mathbf{x}_i]_s^{(m)}[\mathbf{x}_j]_t^{(m)}\right) \tag{G.14}$$

The $(s, t)^{\text{th}}$ entry of the error covariance matrix $\tilde{\mathbf{\Sigma}}_{ij} = \hat{\mathbf{\Sigma}}_{ij} - \mathbf{\Sigma}_{ij} \in \mathbb{R}^{d \times d}$ is defined as

$$[\tilde{\mathbf{\Sigma}}_{ij}]_{st} = \frac{1}{n_2}\left( - \sum_{m \in I_{ij,2}^{st}} [\mathbf{x}_i]_s^{(m)} [\mathbf{x}_j]_t^{(m)} + \sum_{m \in I_{ij,3}^{st}} [\mathbf{x}_i]_s^{(m)} [\mathbf{x}_j]_t^{(m)} \right). \tag{G.15}$$

From the definition of the truncated inner product, we can bound the right-hand side of (G.15) as

$$\left| [\tilde{\mathbf{\Sigma}}_{ij}]_{st} \right| \leq \frac{2}{n_2} \sum_{m \in I_{ij,2}^{st}} \left| [\mathbf{x}_i]_s^{(m)} [\mathbf{x}_j]_t^{(m)} \right|. \tag{G.16}$$

Equipped with the expression of the moment-generating function of a chi-squared distribution, the moment-generating function of each term in the sum of (G.16) can be upper bounded as

$$\mathbb{E}\left[ e^{\lambda | [\mathbf{x}_i]_s^{(m)} [\mathbf{x}_j]_t^{(m)} |} \right] \leq \mathbb{E}\left[ e^{\lambda \frac{([\mathbf{x}_i]_s^{(m)})^2 + ([\mathbf{x}_j]_t^{(m)})^2}{2}} \right] \tag{G.17}$$

$$\leq \sqrt{ \mathbb{E}\left[ e^{\lambda ([\mathbf{x}_i]_s^{(m)})^2} \right] \mathbb{E}\left[ e^{\lambda ([\mathbf{x}_j]_t^{(m)})^2} \right] } \tag{G.18}$$

$$\leq \frac{1}{\sqrt{1 - 2\sigma_{\max}^2 \lambda}}. \tag{G.19}$$

Using the power mean inequality, we have

$$\left( e^{\frac{2\lambda}{n_2} \sum_{m \in I_{ij,2}^{st}} | [\mathbf{x}_i]_s^{(m)} [\mathbf{x}_j]_t^{(m)} |} \right)^{\frac{1}{|I_{ij,2}^{st}|}} \leq \frac{\sum_{m \in I_{ij,2}^{st}} e^{\frac{2\lambda}{n_2} | [\mathbf{x}_i]_s^{(m)} [\mathbf{x}_j]_t^{(m)} |}}{|I_{ij,2}^{st}|} \tag{G.20}$$

$$\leq \left( \frac{\sum_{m \in I_{ij,2}^{st}} e^{\frac{2\lambda n_1}{n_2} | [\mathbf{x}_i]_s^{(m)} [\mathbf{x}_j]_t^{(m)} |}}{|I_{ij,2}^{st}|} \right)^{\frac{1}{n_1}}. \tag{G.21}$$

Thus,

$$\mathbb{E}\left[ e^{\lambda | [\tilde{\mathbf{\Sigma}}_{ij}]_{st} |} \right] \leq \mathbb{E}\left[ e^{\frac{2\lambda}{n_2} \sum_{m \in I_{ij,2}^{st}} \left| [\mathbf{x}_i]_s^{(m)} [\mathbf{x}_j]_t^{(m)} \right|} \right] \leq \max_{m \in I_{ij,2}^{st}} \mathbb{E}\left[ e^{\frac{2\lambda n_1}{n_2} \left| [\mathbf{x}_i]_s^{(m)} [\mathbf{x}_j]_t^{(m)} \right|} \right] \tag{G.22}$$

$$\leq \frac{1}{\sqrt{1 - \frac{4\sigma_{\max}^2 n_1}{n_2} \lambda}} \tag{G.23}$$

and

$$\mathbb{E}\left[ e^{\lambda \max_{s,t} \left| [\tilde{\mathbf{\Sigma}}_{ij}]_{st} \right|} \right] = \mathbb{E}\left[ \max_{s,t} e^{\lambda \left| [\tilde{\mathbf{\Sigma}}_{ij}]_{st} \right|} \right] \leq l_{\max}^2 \mathbb{E}\left[ e^{\lambda \left| [\tilde{\mathbf{\Sigma}}_{ij}]_{st} \right|} \right] \leq \frac{l_{\max}^2}{\sqrt{1 - \frac{4\sigma_{\max}^2 n_1}{n_2} \lambda}}. \tag{G.24}$$

Thus,

$$\mathbb{P}\left( \|\tilde{\mathbf{\Sigma}}_{ij}\|_{\infty,\infty} > t \right) = \mathbb{P}\left( \max_{s,t} \left| [\tilde{\mathbf{\Sigma}}_{ij}]_{st} \right| > t \right) = \mathbb{P}\left( e^{\lambda \max_{s,t} \left| [\tilde{\mathbf{\Sigma}}_{ij}]_{st} \right|} > e^{\lambda t} \right) \tag{G.25}$$

$$\leq e^{-\lambda t} \mathbb{E}\left[ e^{\lambda \max_{s,t} \left| [\tilde{\mathbf{\Sigma}}_{ij}]_{st} \right|} \right] \leq e^{-\lambda t} \frac{l_{\max}^2}{\sqrt{1 - \frac{4\sigma_{\max}^2 n_1}{n_2} \lambda}}. \tag{G.26}$$

Let $\lambda = \frac{3n_2}{16\sigma_{\max}^2 n_1}$, then we have

$$\mathbb{P}\left( \|\tilde{\mathbf{\Sigma}}_{ij}\|_{\infty,\infty} > t \right) \leq 2 l_{\max}^2 e^{-\frac{3n_2}{16\sigma_{\max}^2 n_1} t}. \tag{G.27}$$

According to Lemma 8 (since the involved random variables are sub-exponential), we have

$$\mathbb{P}\left( \left| \frac{1}{n_2} \sum_{m \in I_{ij,1}^{st}} \left( [\mathbf{x}_i]_s^{(m)} [\mathbf{x}_j]_t^{(m)} - [\mathbf{\Sigma}_{ij}]_{st} \right) \right| > t \right) \leq \exp\left( -c \min\left\{ \frac{t^2 n_2}{K^2}, \frac{t n_2}{K} \right\} \right), \tag{G.28}$$

where $K = \sigma_{\max}^2$.

Thus, if $t < \kappa$, we have

$$\mathbb{P}\left( \|\hat{\mathbf{\Sigma}}_{ij} - \mathbf{\Sigma}_{ij}\|_{\infty,\infty} > t_1 + t_2 \right) \leq 2 l_{\max}^2 e^{-\frac{3n_2}{16\kappa n_1} t_1} + l_{\max}^2 e^{-c \frac{t_2^2 n_2}{\kappa^2}}, \tag{G.29}$$

as desired. $\qquad\qquad\qquad\qquad\qquad\qquad\qquad\qquad\qquad\qquad\qquad\qquad\qquad\qquad\qquad\qquad\square$

*Proof of Proposition 2.* From the definition of the information distance, we have

$$\mathrm{d}(x_i, x_j) = -\sum_{n=1}^{l_{\max}} \log \sigma_n\left(\boldsymbol{\Sigma}_{ij}\right) + \frac{1}{2}\log\det\left(\boldsymbol{\Sigma}_{ii}\right) + \frac{1}{2}\log\det\left(\boldsymbol{\Sigma}_{jj}\right). \tag{G.30}$$

According to the inequality $\|\mathbf{A}\|_2 \leq \sqrt{\|\mathbf{A}\|_1 \|\mathbf{A}\|_\infty}$ which holds for all $\mathbf{A} \in \mathbb{R}^{n\times m}$ [23], we have

$$\left|\sigma_k(\hat{\boldsymbol{\Sigma}}_{ij}) - \sigma_k(\boldsymbol{\Sigma}_{ij})\right| \leq \|\hat{\boldsymbol{\Sigma}}_{ij} - \boldsymbol{\Sigma}_{ij}\|_2 \tag{G.31}$$

$$\leq \sqrt{\|\hat{\boldsymbol{\Sigma}}_{ij} - \boldsymbol{\Sigma}_{ij}\|_\infty \|\hat{\boldsymbol{\Sigma}}_{ij} - \boldsymbol{\Sigma}_{ij}\|_1} \leq l_{\max}\|\hat{\boldsymbol{\Sigma}}_{ij} - \boldsymbol{\Sigma}_{ij}\|_{\infty,\infty}. \tag{G.32}$$

Using the triangle inequality, we arrive at

$$\left|\hat{\mathrm{d}}(x_i, x_j) - \mathrm{d}(x_i, x_j)\right| \leq \sum_{n=1}^{l_{\max}} \left|\log\sigma_n(\hat{\boldsymbol{\Sigma}}_{ij}) - \log\sigma_n(\boldsymbol{\Sigma}_{ij})\right| + \frac{1}{2}\sum_{n=1}^{\dim(\mathbf{x}_i)}\left|\log\sigma_n(\hat{\boldsymbol{\Sigma}}_{ii}) - \log\sigma_n(\boldsymbol{\Sigma}_{ii})\right|$$

$$+ \frac{1}{2}\sum_{n=1}^{\dim(\mathbf{x}_j)}\left|\log\sigma_n(\hat{\boldsymbol{\Sigma}}_{jj}) - \log\sigma_n(\boldsymbol{\Sigma}_{jj})\right|. \tag{G.33}$$

Furthermore, since the singular value is lower bounded by $\gamma_{\min}$, using Taylor's theorem and (G.31), we obtain

$$\left|\log\sigma_n(\hat{\boldsymbol{\Sigma}}_{ij}) - \log\sigma_n(\boldsymbol{\Sigma}_{ij})\right| \leq \frac{1}{\gamma_{\min}}\left|\sigma_n(\hat{\boldsymbol{\Sigma}}_{ij}) - \sigma_n(\boldsymbol{\Sigma}_{ij})\right| \leq \frac{l_{\max}}{\gamma_{\min}}\|\hat{\boldsymbol{\Sigma}}_{ij} - \boldsymbol{\Sigma}_{ij}\|_{\infty,\infty}. \tag{G.34}$$

Finally,

$$\left|\hat{\mathrm{d}}(x_i, x_j) - \mathrm{d}(x_i, x_j)\right| \leq \left(l_{\max} + \frac{\dim(\mathbf{x}_i) + \dim(\mathbf{x}_j)}{2}\right)\frac{l_{\max}}{\gamma_{\min}}\|\hat{\boldsymbol{\Sigma}}_{ij} - \boldsymbol{\Sigma}_{ij}\|_{\infty,\infty}$$

$$\leq \frac{2l_{\max}^2}{\gamma_{\min}}\|\hat{\boldsymbol{\Sigma}}_{ij} - \boldsymbol{\Sigma}_{ij}\|_{\infty,\infty}. \tag{G.35}$$

From Lemma 9, the proposition is proved. □

# H   Proofs of results in Section 3.2

**Lemma 10.** *Consider the optimization problem*

$$\mathscr{P}: \quad \max_{\{x_i\}} \quad f(\mathbf{x}) = \sum_{i=1}^{n} x_i(x_i - 1)$$

$$\text{s.t.} \quad \sum_{i=1}^{N} x_i \leq N \quad 0 \leq x_i \leq k \qquad i = 1, \ldots, N. \tag{H.1}$$

*Assume $nk \geq N$. An optimal solution is given by $x_i = k$ for all $i = 1, \ldots, \lfloor\frac{N}{k}\rfloor$ and $x_{\lfloor\frac{N}{k}\rfloor+1} = N - k\lfloor\frac{N}{k}\rfloor$, and $x_i = 0$ for $i = \lfloor\frac{N}{k}\rfloor + 2, \ldots, n$.*

This lemma can be verified by direct calculation, and so we will omit the details.

*Proof of Proposition 4.* We prove the proposition by induction.

Proposition 2 and Eqn. (6) show that at the $0^{\text{th}}$ layer [24]

$$\mathbb{P}(|\Delta_{ij}| > \varepsilon) < f(\varepsilon) = h^{(0)}(\varepsilon). \tag{H.2}$$

Now suppose that the distances related to the nodes created in the $(l-1)^{\text{st}}$ iteration satisfy

$$\mathbb{P}\left(\left|\hat{\mathrm{d}}(x_i, x_h) - \mathrm{d}(x_i, x_h)\right| > \varepsilon\right) < h^{(l-1)}(\varepsilon). \tag{H.3}$$

Since $s > 1$ and $m < 1$, it is obvious that

$$h^{(l)}(\varepsilon) \leq h^{(l+k)}(\varepsilon) \quad \text{for all} \quad l, k \in \mathbb{N} \quad \text{and for all} \quad \varepsilon > 0. \tag{H.4}$$

Then we can deduce that

$$\mathbb{P}(|\hat{\mathrm{d}}(x_i, x_j) - \mathrm{d}(x_i, x_j)| > \varepsilon) < h^{(l-1)}(\varepsilon) \quad \text{for all} \quad x_i, x_j \in \Gamma^l. \tag{H.5}$$

From the update equation of the distance in (9), we have

$$\hat{\mathrm{d}}(x_i, x_h) = \frac{1}{2(|\mathcal{C}(h) - 1|)} \left( \sum_{j \in \mathcal{C}(h)} (\mathrm{d}(x_i, x_j) + \Delta_{ij}) + \frac{1}{|\mathcal{K}_{ij}|} \sum_{k \in \mathcal{K}_{ij}} (\Phi_{ijk} + \Delta_{ik} - \Delta_{jk}) \right) \tag{H.6}$$

and

$$\hat{\mathrm{d}}(x_i, x_h) = \frac{1}{2(|\mathcal{C}(h) - 1|)} \left( \sum_{j \in \mathcal{C}(h)} \Delta_{ij} + \frac{1}{|\mathcal{K}_{ij}|} \sum_{k \in \mathcal{K}_{ij}} (\Delta_{ik} - \Delta_{jk}) \right) + \mathrm{d}(x_i, x_h). \tag{H.7}$$

Using the union bound, we find that

$$\mathbb{P}\left( |\hat{\mathrm{d}}(x_i, x_h) - \mathrm{d}(x_i, x_h)| > \varepsilon \right)$$

$$\leq \mathbb{P}\left( \bigcup_{j \in \mathcal{C}(h)} \left\{ |\Delta_{ij} + \frac{1}{|\mathcal{K}_{ij}|} \sum_{k \in \mathcal{K}_{ij}} (\Delta_{ik} - \Delta_{jk})| > 2\varepsilon \right\} \right) \tag{H.8}$$

$$\leq \sum_{j \in \mathcal{C}(h)} \mathbb{P}\left( |\Delta_{ij} + \frac{1}{|\mathcal{K}_{ij}|} \sum_{k \in \mathcal{K}_{ij}} (\Delta_{ik} - \Delta_{jk})| > 2\varepsilon \right) \tag{H.9}$$

$$\leq \sum_{j \in \mathcal{C}(h)} \left[ \mathbb{P}\left( |\Delta_{ij}| > \frac{2}{3}\varepsilon \right) + \sum_{k \in \mathcal{K}_{ij}} \mathbb{P}\left( |\Delta_{ik}| > \frac{2}{3}\varepsilon \right) + \mathbb{P}\left( |\Delta_{jk}| > \frac{2}{3}\varepsilon \right) \right]. \tag{H.10}$$

The estimates of the distances related to the nodes in the $l^{\text{th}}$ layer satisfy

$$\mathbb{P}\left( |\hat{\mathrm{d}}(x_i, x_h) - \mathrm{d}(x_i, x_h)| > \varepsilon \right) < |\mathcal{C}(h)|(1 + 2|\mathcal{K}_{ij}|) h^{(l-1)}\left( \frac{2}{3}\varepsilon \right) \tag{H.11}$$

$$\leq d_{\max}(1 + 2N_\tau) h^{(l-1)}\left( \frac{2}{3}\varepsilon \right). \tag{H.12}$$

Similarly, from (10), we have

$$\mathbb{P}\left( |\hat{\mathrm{d}}(x_k, x_h) - \mathrm{d}(x_k, x_h)| > \varepsilon \right)$$

$$\leq \begin{cases} \sum_{i \in \mathcal{C}(h)} \mathbb{P}(|\Delta_{ik}| > \frac{1}{2}\varepsilon) + \mathbb{P}(|\Delta_{ih}| > \frac{1}{2}\varepsilon), & \text{if } k \in \mathcal{V}_{\text{obs}} \\ \sum_{(i,j) \in \mathcal{C}(h) \times \mathcal{C}(k)} \mathbb{P}(|\Delta_{ij}| > \frac{1}{3}\varepsilon) + \mathbb{P}(|\Delta_{ih}| > \frac{1}{3}\varepsilon) + \mathbb{P}(|\Delta_{jk}| > \frac{1}{3}\varepsilon), & \text{otherwise.} \end{cases} \tag{H.13}$$

Using the concentration bound at the $(l-1)^{\text{st}}$ layer in inequality (H.3), we have

$$\mathbb{P}\left( |\hat{\mathrm{d}}(x_k, x_h) - \mathrm{d}(x_k, x_h)| > \varepsilon \right)$$

$$\leq \begin{cases} d_{\max} h^{(l-1)}(\frac{1}{2}\varepsilon) + d_{\max}^2 (1 + 2N_\tau) h^{(l-1)}(\frac{1}{3}\varepsilon), & \text{if } k \in \mathcal{V}_{\text{obs}} \\ d_{\max}^2 h^{(l-1)}(\frac{2}{3}\varepsilon) + 2d_{\max}^3 (1 + 2N_\tau) h^{(l-1)}(\frac{2}{9}\varepsilon), & \text{otherwise.} \end{cases} \tag{H.14}$$

Summarizing the above three concentration bounds, we have that for the nodes at the $l^{\text{th}}$ layer, estimates of the information distances (based on the truncated inner product) satisfy

$$\mathbb{P}\left( |\hat{\mathrm{d}}(x_k, x_h) - \mathrm{d}(x_k, x_h)| > \varepsilon \right) < \left[ d_{\max}^2 + 2d_{\max}^3(1 + 2N_\tau) \right] h^{(l-1)}\left( \frac{2}{9}\varepsilon \right) = h^{(l)}(\varepsilon). \tag{H.15}$$

$\square$

**Proposition 11.** *The cardinalities of the active sets in $l^{\text{th}}$ and $(l+1)^{\text{st}}$ iterations admit following relationship*

$$\frac{|\Gamma^l|}{d_{\max}} \leq |\Gamma^{l+1}| \leq |\Gamma^l| - 2. \tag{H.16}$$

*Proof of Proposition 11.* Note that at the $l^{\text{th}}$ iteration, the number of families is $|\Gamma^{l+1}|$, and thus we have

$$\sum_{i=1}^{|\Gamma^{l+1}|} n_i = |\Gamma^l|, \tag{H.17}$$

where $n_i$ is the number of nodes in $\Gamma^l$ in each family. Since $1 \le n_i \le d_{\max}$, we have $\frac{\Gamma^l}{d_{\max}} \le |\Gamma^{l+1}|$.

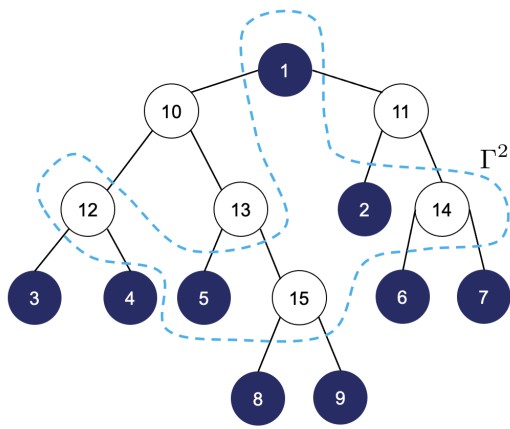

Figure 7: Illustration of RRG. The shaded nodes are the observed nodes and the rest are hidden nodes. $\Gamma^1 = \{x_1, x_2, \ldots, x_9\}$, and $\Gamma^2$ is the nodes in the dotted lines. If we delete the nodes in $\Gamma^2$, the remained unknown hidden nodes are $x_{10}$, $x_{11}$ and $x_{13}$. Nodes $x_{10}$ and $x_{13}$ are at the end of the chain formed by these two nodes, and $x_{11}$ is at the end of the degenerate chain formed by itself.

We next prove that there are at least two of $n_i$'s not less than 2. If we delete the nodes in active set $\Gamma^l$, the remaining hidden nodes form a single tree or a forest. There will at least two nodes at the end of the chain, which means that they only have one neighbor in hidden nodes, as shown in Fig. 7. Since they at least have three neighbors, they have at least two neighbors in $\Gamma^l$. Thus, there are at least two of $n_i$'s not less than 2, and thus $|\Gamma^{l+1}| \le |\Gamma^l| - 2$. $\qquad\square$

**Corollary 1.** *The maximum number of iterations of Algorithm 1, $L_{\text{R}}$, is bounded as*

$$\frac{\log \frac{|\mathcal{V}_{\text{obs}}|}{2}}{\log d_{\max}} \le L_{\text{R}} \le |\mathcal{V}_{\text{obs}}| - 2. \tag{H.18}$$

*Proof.* When Algorithm 1 terminates, $|\Gamma| \le 2$. Combining Proposition 11 and $|\Gamma| \le 2$ proves the corollary. $\qquad\square$

**Theorem 6.** *Under Assumptions 1–5, RRG algorithm constructs the correct latent tree with probability at least $1 - \eta$ if*

$$n_2 \ge \frac{64\lambda^2\kappa^2}{c\varepsilon^2}\left(\frac{9}{2}\right)^{2L_{\text{R}}-2} \log \frac{17l_{\max}^2 s^{L_{\text{R}}-1}|\mathcal{V}_{\text{obs}}|^3}{\eta} \tag{H.19}$$

$$\frac{n_2}{n_1} \ge \frac{128\lambda\kappa}{3\varepsilon}\left(\frac{9}{2}\right)^{L_{\text{R}}-1} \log \frac{34l_{\max}^2 s^{L_{\text{R}}-1}|\mathcal{V}_{\text{obs}}|^3}{\eta}, \tag{H.20}$$

*where*

$$\lambda = \frac{2l_{\max}^2 e^{\rho_{\max}/l_{\max}}}{\delta_{\min}^{1/l_{\max}}} \quad \kappa = \max\{\sigma_{\max}^2, \rho_{\min}\} \quad s = d_{\max}^2 + 2d_{\max}^3(1 + 2N_\tau) \quad \varepsilon = \frac{\rho_{\min}}{2}, \tag{H.21}$$

*$c$ is an absolute constant, and $L_{\text{R}}$ is the number of iterations of RRG needed to construct the tree.*

*Proof of Theorem 6.* It is easy to see by substituting the constants $\lambda$, $\kappa$, $s$ and $\varepsilon$ into (H.19) and (H.20) that Theorem 6 implies Theorem 1, so we provide the proof of Theorem 6 here.

The error events of learning structure in the $l^{\text{th}}$ layer of the latent tree (the $0^{\text{th}}$ layer consists of the observed nodes, and the $(l+1)^{\text{st}}$ layer is the active set formed from $l^{\text{th}}$ layer). The error events could be enumerated as: misclassification of families $\mathcal{E}_{\text{f}}^l$, misclassification of non-families $\mathcal{E}_{\text{nf}}^l$, misclassification of parents $\mathcal{E}_{\text{p}}^l$ and misclassification of siblings $\mathcal{E}_{\text{s}}^l$. We will bound the probabilities of these four error events in the following.

The event representing misclassification of families $\mathcal{E}_{\text{f}}^l$ represents classifying the nodes that are not in the same family as a family. Suppose nodes $x_i$ and $x_j$ are in different families. The event that classifying them to be in the same family $\mathcal{E}_{\text{f},ij}^l$ at layer $l$ can be expressed as

$$\mathcal{E}_{\text{f},ij}^l = \left\{ |\hat{\Phi}_{ijk} - \hat{\Phi}_{ijk'}| < \varepsilon \quad \text{for all} \quad x_k, x_{k'} \in \Gamma^l \right\}. \tag{H.22}$$

We have

$$\mathbb{P}(\mathcal{E}_{\text{f},ij}^l) = \mathbb{P}\left( \bigcap_{x_k, x_{k'} \in \Gamma} \left\{ |\hat{\Phi}_{ijk} - \hat{\Phi}_{ijk'}| < \varepsilon \right\} \right) \leq \min_{x_k, x_{k'} \in \Gamma} \mathbb{P}\left( |\hat{\Phi}_{ijk} - \hat{\Phi}_{ijk'}| < \varepsilon \right), \tag{H.23}$$

$$\mathbb{P}(\mathcal{E}_{\text{f}}^l) = \mathbb{P}\left( \bigcup_{x_i, x_j \text{not in same family}} \mathcal{E}_{\text{f},ij}^l \right) = \mathbb{P}\left( \bigcup_{(x_i, x_j) \in \Gamma_{\text{f}}^l} \mathcal{E}_{\text{f},ij}^l \right). \tag{H.24}$$

We enumerate all possible structural relationships between $x_i$, $x_j$, $x_k$ and $x_{k'}$

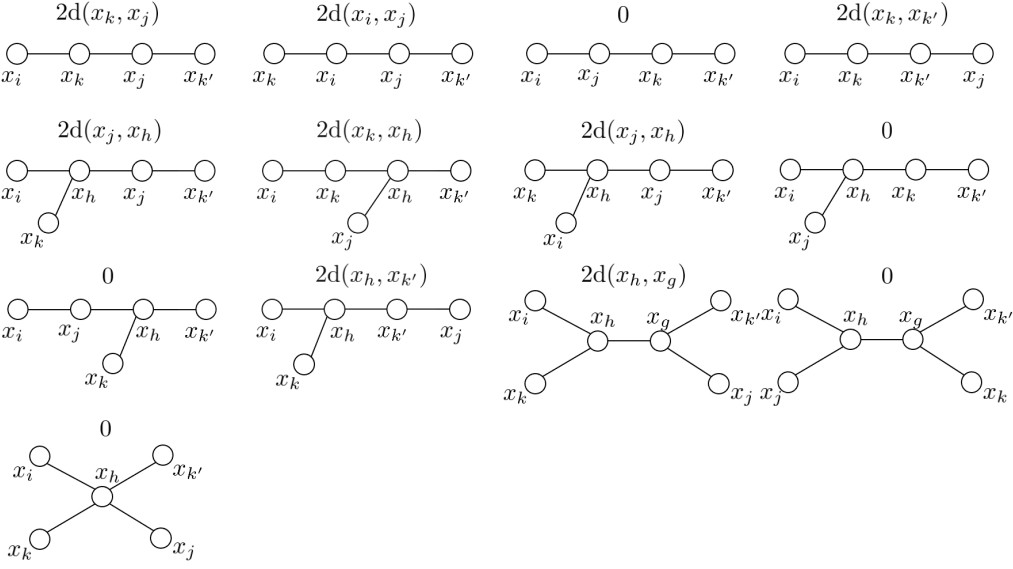

Figure 8: Enumerating of four-node topology and the corresponding $|\Phi_{ijk} - \Phi_{ijk'}|$.

Let $\varepsilon < 2\rho_{\min}$, by decomposing the estimate of the information distance as $\hat{\text{d}}(x_i, x_j) = \text{d}(x_i, x_j) + \Delta_{ij}$, we have

$$\mathbb{P}\left( |\hat{\Phi}_{ijk} - \hat{\Phi}_{ijk'}| < \varepsilon \right) = \mathbb{P}\left( |\Phi_{ijk} - \Phi_{ijk'} + \Delta_{ik} - \Delta_{jk} - \Delta_{ik'} + \Delta_{jk'}| < \varepsilon \right)$$

$$\leq \mathbb{P}\left( \Delta_{ik} - \Delta_{jk} - \Delta_{ik'} + \Delta_{jk'} < \varepsilon - (\Phi_{ijk} - \Phi_{ijk'}) \right)$$

$$\leq \mathbb{P}\left( \Delta_{ik} - \Delta_{jk} - \Delta_{ik'} + \Delta_{jk'} < \varepsilon - 2\rho_{\min} \right)$$

$$\leq \mathbb{P}\left( |\Delta_{ik}| > \frac{2\rho_{\min} - \varepsilon}{4} \right) + \mathbb{P}\left( |\Delta_{jk}| > \frac{2\rho_{\min} - \varepsilon}{4} \right)$$

$$+ \mathbb{P}\left( |\Delta_{jk'}| > \frac{2\rho_{\min} - \varepsilon}{4} \right) + \mathbb{P}\left( |\Delta_{ik'}| > \frac{2\rho_{\min} - \varepsilon}{4} \right). \tag{H.25}$$

The event representing misclassification of the parents $\mathcal{E}_{\mathrm{p}}^l$ represents classifying a sibling relationship as a parent relationship. Following similar procedures, we have

$$\mathbb{P}(\mathcal{E}_{\mathrm{p}}^l) = \mathbb{P}\Big( \bigcup_{x_i, x_j \text{ are siblings}} \mathcal{E}_{\mathrm{p},ij}^l \Big) = \mathbb{P}\Big( \bigcup_{(x_i, x_j) \in \Gamma_{\mathrm{p}}^l} \mathcal{E}_{\mathrm{p},ij}^l \Big) \tag{H.26}$$

$$\mathbb{P}(\mathcal{E}_{\mathrm{p},ij}^l) = \mathbb{P}\Big( \bigcap_{x_k \in \Gamma^l} \big\{ |\hat{\Phi}_{ijk} - \hat{\mathrm{d}}(x_i, x_j)| < \varepsilon \big\} \Big) \leq \min_{x_k \in \Gamma^l} \mathbb{P}\Big( |\hat{\Phi}_{ijk} - \hat{\mathrm{d}}(x_i, x_j)| < \varepsilon \Big) \tag{H.27}$$

$$\leq \mathbb{P}\Big( |\Delta_{ij}| > \frac{2\rho_{\min} - \varepsilon}{3} \Big) + \mathbb{P}\Big( |\Delta_{ik}| > \frac{2\rho_{\min} - \varepsilon}{3} \Big) + \mathbb{P}\Big( |\Delta_{jk}| > \frac{2\rho_{\min} - \varepsilon}{3} \Big) \tag{H.28}$$

The event representing misclassification of non-families $\mathcal{E}_{\mathrm{nf}}^l$ represents classifying family members as non-family members. We have

$$\mathbb{P}(\mathcal{E}_{\mathrm{nf}}^l) = \mathbb{P}\Big( \bigcup_{x_i, x_j \text{ in the same family}} \mathcal{E}_{\mathrm{nf},ij}^l \Big) = \mathbb{P}\Big( \bigcup_{(x_i, x_j) \in \Gamma_{\mathrm{nf}}^l} \mathcal{E}_{\mathrm{nf},ij}^l \Big) \tag{H.29}$$

$$\mathbb{P}(\mathcal{E}_{\mathrm{nf}}^l) = \mathbb{P}\Big( \bigcup_{x_i, x_j \text{ in the same family}} \bigcup_{x_k, x_{k'} \in \Gamma} \big\{ |\hat{\Phi}_{ijk} - \hat{\Phi}_{ijk'}| > \varepsilon \big\} \Big) \tag{H.30}$$

and

$$\mathbb{P}\Big( |\hat{\Phi}_{ijk} - \hat{\Phi}_{ijk'}| \geq \varepsilon \Big) \leq \mathbb{P}\Big( |\Delta_{ik}| > \frac{\varepsilon}{4} \Big) + \mathbb{P}\Big( |\Delta_{jk}| > \frac{\varepsilon}{4} \Big) + \mathbb{P}\Big( |\Delta_{jk'}| > \frac{\varepsilon}{4} \Big) + \mathbb{P}\Big( |\Delta_{ik'}| > \frac{\varepsilon}{4} \Big) \tag{H.31}$$

The event representing misclassification of siblings $\mathcal{E}_{\mathrm{s}}^l$ represents classifying parent relationship as sibling relationship. Similarly, we have

$$\mathbb{P}(\mathcal{E}_{\mathrm{s}}^l) = \mathbb{P}\Big( \bigcup_{x_i \text{ is the parent of } x_j} \mathcal{E}_{\mathrm{s},ij}^l \Big) = \mathbb{P}\Big( \bigcup_{(x_i, x_j) \in \Gamma_{\mathrm{s}}^l} \mathcal{E}_{\mathrm{s},ij}^l \Big) \tag{H.32}$$

$$\mathbb{P}(\mathcal{E}_{\mathrm{s},ij}^l) = \mathbb{P}\Big( \bigcup_{x_k \in \Gamma} \big\{ |\hat{\Phi}_{jik} - \hat{\mathrm{d}}(x_i, x_j)| > \varepsilon \big\} \Big) \tag{H.33}$$

and

$$\mathbb{P}\Big( |\hat{\Phi}_{jik} - \hat{\mathrm{d}}(x_i, x_j)| > \varepsilon \Big) \leq \mathbb{P}\Big( |\Delta_{ij}| > \frac{\varepsilon}{3} \Big) + \mathbb{P}\Big( |\Delta_{ik}| > \frac{\varepsilon}{3} \Big) + \mathbb{P}\Big( |\Delta_{jk}| > \frac{\varepsilon}{3} \Big) \tag{H.34}$$

To bound the probability of error event in $l^{\mathrm{th}}$ layer, we first analyze the cardinalities of $\Gamma_{\mathrm{f}}^l$, $\Gamma_{\mathrm{p}}^l$, $\Gamma_{\mathrm{nf}}^l$ and $\Gamma_{\mathrm{s}}^l$. Note that the definitions of these four sets are

$$\Gamma_{\mathrm{f}}^l = \big\{ (x_i, x_j) : x_i \text{ and } x_j \text{ are not in the same family } x_i, x_j \in \Gamma^l \big\} \tag{H.35}$$

$$\Gamma_{\mathrm{p}}^l = \big\{ (x_i, x_j) : x_i \text{ and } x_j \text{ are siblings } x_i, x_j \in \Gamma^l \big\} \tag{H.36}$$

$$\Gamma_{\mathrm{nf}}^l = \big\{ (x_i, x_j) : x_i \text{ and } x_j \text{ are in the same family } x_i, x_j \in \Gamma^l \big\} \tag{H.37}$$

$$\Gamma_{\mathrm{s}}^l = \big\{ (x_i, x_j) : x_i \text{ and } x_j \text{ is the parent of } x_i, x_j \in \Gamma^l \big\}. \tag{H.38}$$

Clearly, we have

$$|\Gamma_{\mathrm{f}}^l| \leq \binom{|\Gamma^l|}{2} \quad \text{and} \quad |\Gamma_{\mathrm{s}}^l| \leq |\Gamma^l|. \tag{H.39}$$

The cardinality of $\Gamma_{\mathrm{p}}^l$ can be bounded as

$$|\Gamma_{\mathrm{p}}^l| \leq \sum_{i=1}^{|\Gamma^{l+1}|} \binom{n_i}{2} \tag{H.40}$$

where $n_i$ is the size of each family in $\Gamma^l$.

From Lemma 10, we deduce that

$$|\Gamma_{\mathrm{p}}^l| \le \frac{1}{2}d_{\max}(d_{\max}-1)\frac{|\Gamma^l|}{d_{\max}} = \frac{1}{2}|\Gamma^l|(d_{\max}-1). \tag{H.41}$$

Similarly, we have

$$|\Gamma_{\mathrm{nf}}^l| \le \frac{1}{2}|\Gamma^l|(d_{\max}-1). \tag{H.42}$$

The probability of the error event in $l^{\mathrm{th}}$ layer can be bounded as

$$\begin{aligned}
\mathbb{P}(\mathcal{E}^l) &= \mathbb{P}(\mathcal{E}_{\mathrm{f}}^l \cup \mathcal{E}_{\mathrm{p}}^l \cup \mathcal{E}_{\mathrm{nf}}^l \cup \mathcal{E}_{\mathrm{s}}^l) \\
&\le \mathbb{P}(\mathcal{E}_{\mathrm{f}}^l) + \mathbb{P}(\mathcal{E}_{\mathrm{p}}^l) + \mathbb{P}(\mathcal{E}_{\mathrm{nf}}^l) + \mathbb{P}(\mathcal{E}_{\mathrm{s}}^l) \\
&\le 4\binom{|\Gamma^l|}{2}h^{(l)}\Big(\frac{2\rho_{\min}-\varepsilon}{4}\Big) + \frac{3}{2}|\Gamma^l|(d_{\max}-1)h^{(l)}\Big(\frac{2\rho_{\min}-\varepsilon}{3}\Big) \\
&\quad + 3|\Gamma^l|^2 h^{(l)}\Big(\frac{\varepsilon}{3}\Big) + 2|\Gamma^l|^3(d_{\max}-1)h^{(l)}\Big(\frac{\varepsilon}{4}\Big). 
\end{aligned} \tag{H.43}$$

The probability of learning the wrong structure is

$$\mathbb{P}(\mathcal{E}) = \mathbb{P}\Big(\bigcup_l \mathcal{E}^l\Big) \le \sum_l \mathbb{P}(\mathcal{E}^l) \tag{H.44}$$

$$\begin{aligned}
&\le \sum_l 4\binom{|\Gamma^l|}{2}h^{(l)}\Big(\frac{2\rho_{\min}-\varepsilon}{4}\Big) + \frac{3}{2}|\Gamma^l|(d_{\max}-1)h^{(l)}\Big(\frac{2\rho_{\min}-\varepsilon}{3}\Big) \\
&\quad + 3|\Gamma^l|^2 h^{(l)}\Big(\frac{\varepsilon}{3}\Big) + 2|\Gamma^l|^3(d_{\max}-1)h^{(l)}\Big(\frac{\varepsilon}{4}\Big) 
\end{aligned} \tag{H.45}$$

With Proposition 11, we have

$$\begin{aligned}
\mathbb{P}(\mathcal{E}) &\le \sum_{l=0}^{L-1} 4\binom{|\mathcal{V}_{\mathrm{obs}}|-2l}{2}h^{(l)}\Big(\frac{2\rho_{\min}-\varepsilon}{4}\Big) + \frac{3}{2}(|\mathcal{V}_{\mathrm{obs}}|-2l)(d_{\max}-1)h^{(l)}\Big(\frac{2\rho_{\min}-\varepsilon}{3}\Big) \\
&\quad + 3(|\mathcal{V}_{\mathrm{obs}}|-2l)^2 h^{(l)}\Big(\frac{\varepsilon}{3}\Big) + 2(|\mathcal{V}_{\mathrm{obs}}|-2l)^3(d_{\max}-1)h^{(l)}\Big(\frac{\varepsilon}{4}\Big), 
\end{aligned} \tag{H.46}$$

where $L$ is the number of iterations of RRG.

We can separately bound the two parts of the first term in the summation $\binom{|\mathcal{V}_{\mathrm{obs}}|-2l}{2}h^{(l)}\Big(\frac{2\rho_{\min}-\varepsilon}{4}\Big)$ as

$$\frac{4\binom{|\mathcal{V}_{\mathrm{obs}}|-2l}{2}s^l a e^{-wm^l x}}{4\binom{|\mathcal{V}_{\mathrm{obs}}|-2L}{2}s^{L-1}a e^{-wm^{L-1}x}} \le \frac{\big(|\mathcal{V}_{\mathrm{obs}}|-2l\big)\big(|\mathcal{V}_{\mathrm{obs}}|-2l-1\big)}{2s^{L-1-l}} \le \frac{|\mathcal{V}_{\mathrm{obs}}|^2}{2s^{L-1-l}} \text{ for } x>0$$

and

$$\frac{4\binom{|\mathcal{V}_{\mathrm{obs}}|-2l}{2}s^l b e^{-um^{2l}x^2}}{4\binom{|\mathcal{V}_{\mathrm{obs}}|-2L}{2}s^{L-1}b e^{-um^{2L-2}x^2}} \le \frac{\big(|\mathcal{V}_{\mathrm{obs}}|-2l\big)\big(|\mathcal{V}_{\mathrm{obs}}|-2l-1\big)}{2s^{L-1-l}} \le \frac{|\mathcal{V}_{\mathrm{obs}}|^2}{2s^{L-1-l}}. \tag{H.47}$$

These bounds imply that

$$\sum_{l=0}^{L-1} 4\binom{|\mathcal{V}_{\mathrm{obs}}|-2l}{2}h^{(l)}(x) \le \big[1 + \frac{|\mathcal{V}_{\mathrm{obs}}|^2}{2}\sum_{i=1}^{\infty}\frac{1}{s^i}\big]4h^{(L-1)}(x) = 4\Big(1 + \frac{|\mathcal{V}_{\mathrm{obs}}|^2}{2(s-1)}\Big)4h^{(L-1)}(x) \text{ for } x>0$$

Similar procedures could be implemented on other terms, and we will obtain

$$\begin{aligned}
\mathbb{P}(\mathcal{E}) &\le \Big[4\Big(1 + \frac{|\mathcal{V}_{\mathrm{obs}}|^2}{2(s-1)}\Big) + \frac{3}{2}(d_{\max}-1)\Big(1 + \frac{|\mathcal{V}_{\mathrm{obs}}|}{2(s-1)}\Big)\Big]h^{(L-1)}\Big(\frac{2\rho_{\min}-\varepsilon}{4}\Big) \\
&\quad + \Big[3\Big(1 + \frac{|\mathcal{V}_{\mathrm{obs}}|^2}{4(s-1)}\Big) + 2(d_{\max}-1)\Big(1 + \frac{|\mathcal{V}_{\mathrm{obs}}|^3}{8(s-1)}\Big)\Big]h^{(L-1)}\Big(\frac{\varepsilon}{4}\Big) \\
&= \Big[4\Big(1 + \frac{|\mathcal{V}_{\mathrm{obs}}|^2}{2(s-1)}\Big) + \frac{3}{2}(d_{\max}-1)\Big(1 + \frac{|\mathcal{V}_{\mathrm{obs}}|}{2(s-1)}\Big)\Big]\Big(a e^{-wm^{L-1}\frac{2\rho_{\min}-\varepsilon}{4}} + b e^{-um^{2L-2}(\frac{2\rho_{\min}-\varepsilon}{4})^2}\Big) \\
&\quad + \Big[3\Big(1 + \frac{|\mathcal{V}_{\mathrm{obs}}|^2}{4(s-1)}\Big) + 2(d_{\max}-1)\Big(1 + \frac{|\mathcal{V}_{\mathrm{obs}}|^3}{8(s-1)}\Big)\Big]\Big(a e^{-wm^{L-1}\frac{\varepsilon}{4}} + b e^{-um^{2L-2}(\frac{\varepsilon}{4})^2}\Big) \le \eta.
\end{aligned} \tag{H.48}$$

Upper bounding each of the four terms in inequality (H.48) by $\eta/4$, we obtain the following sufficient conditions of $n_1$ and $n_2$ to ensure that $\mathbb{P}(\mathcal{E}) \leq \eta$:

$$n_2 \geq \max\left\{ \frac{64\lambda^2\kappa^2}{c(2\rho_{\min} - \varepsilon)^2}\left(\frac{9}{2}\right)^{2L-2} \log \frac{4l_{\max}^2 s^{L-1}\left[4(1 + \frac{|\mathcal{V}_{\text{obs}}|^2}{2(s-1)}) + \frac{3}{2}(d_{\max} - 1)(1 + \frac{|\mathcal{V}_{\text{obs}}|}{2(s-1)})\right]}{\eta}, \right.$$

$$\left. \frac{64\lambda^2\kappa^2}{c\varepsilon^2}\left(\frac{9}{2}\right)^{2L-2} \log \frac{4l_{\max}^2 s^{L-1}\left[3(1 + \frac{|\mathcal{V}_{\text{obs}}|^2}{4(s-1)}) + 2(d_{\max} - 1)(1 + \frac{|\mathcal{V}_{\text{obs}}|^3}{8(s-1)})\right]}{\eta} \right\},$$

$$\frac{n_2}{n_1} \geq \max\left\{ \frac{128\lambda\kappa}{3(2\rho_{\min} - \varepsilon)}\left(\frac{9}{2}\right)^{L-1} \log \frac{8l_{\max}^2 s^{L-1}\left[4(1 + \frac{|\mathcal{V}_{\text{obs}}|^2}{2(s-1)}) + \frac{3}{2}(d_{\max} - 1)(1 + \frac{|\mathcal{V}_{\text{obs}}|}{2(s-1)})\right]}{\eta}, \right.$$

$$\left. \frac{128\lambda\kappa}{3\varepsilon}\left(\frac{9}{2}\right)^{L-1} \log \frac{8l_{\max}^2 s^{L-1}\left[3(1 + \frac{|\mathcal{V}_{\text{obs}}|^2}{4(s-1)}) + 2(d_{\max} - 1)(1 + \frac{|\mathcal{V}_{\text{obs}}|^3}{8(s-1)})\right]}{\eta} \right\}.$$

Note that

$$\max\left\{ 4\left(1 + \frac{|\mathcal{V}_{\text{obs}}|^2}{2(s-1)}\right) + \frac{3}{2}(d_{\max} - 1)\left(1 + \frac{|\mathcal{V}_{\text{obs}}|}{2(s-1)}\right), \right.$$

$$\left. 3\left(1 + \frac{|\mathcal{V}_{\text{obs}}|^2}{(s-1)}\right) + 2(d_{\max} - 1)\left(1 + \frac{|\mathcal{V}_{\text{obs}}|^3}{8(s-1)}\right) \right\}$$

$$< 4\left(1 + \frac{|\mathcal{V}_{\text{obs}}|^2}{2(s-1)}\right) + 2(d_{\max} - 1)\left(1 + \frac{|\mathcal{V}_{\text{obs}}|^3}{2(s-1)}\right) \tag{H.49}$$

$$\leq 2(d_{\max} - 1)\left(2 + \frac{|\mathcal{V}_{\text{obs}}|^3 + 2|\mathcal{V}_{\text{obs}}|^2}{2(s-1)}\right) \tag{H.50}$$

$$< 2(d_{\max} - 1)\left(2 + \frac{|\mathcal{V}_{\text{obs}}|^3 + 2|\mathcal{V}_{\text{obs}}|^2}{s}\right) \tag{H.51}$$

$$\overset{(a)}{<} 2(d_{\max} - 1)\frac{|\mathcal{V}_{\text{obs}}|^3 + 2|\mathcal{V}_{\text{obs}}|^2 + 7N_\tau|\mathcal{V}_{\text{obs}}|^3}{s} \tag{H.52}$$

$$< 17 d_{\max} N_\tau \frac{|\mathcal{V}_{\text{obs}}|^3}{s} \tag{H.53}$$

$$\overset{(b)}{<} \frac{17}{4}|\mathcal{V}_{\text{obs}}|^3, \tag{H.54}$$

where inequality $(a)$ and $(b)$ result from $s < 7N_\tau|\mathcal{V}_{\text{obs}}|^3$ and $d_{\max}N_\tau < \frac{s}{4}$, respectively. Choosing $\varepsilon < \rho_{\min}$, we then can derive the sufficient conditions to ensure that $\mathbb{P}(\mathcal{E}) \leq \eta$ as

$$n_2 \geq \frac{64\lambda^2\kappa^2}{c\varepsilon^2}\left(\frac{9}{2}\right)^{2L-2} \log \frac{17l_{\max}^2 s^{L-1}|\mathcal{V}_{\text{obs}}|^3}{\eta}, \tag{H.55}$$

$$\frac{n_2}{n_1} \geq \frac{128\lambda\kappa}{3\varepsilon}\left(\frac{9}{2}\right)^{L-1} \log \frac{34l_{\max}^2 s^{L-1}|\mathcal{V}_{\text{obs}}|^3}{\eta}. \tag{H.56}$$

In Theorem 1, we choose $\varepsilon = \frac{\rho_{\min}}{2}$.

Then the following conditions

$$n_2 \geq \frac{64\lambda^2\kappa^2}{c\varepsilon^2}\left(\frac{9}{2}\right)^{2L-2} \log \frac{17l_{\max}^2 s^{L-1}|\mathcal{V}_{\text{obs}}|^3}{\eta}, \tag{H.57}$$

$$n_1 = O\left(\frac{\sqrt{n_2}}{\log n_2}\right). \tag{H.58}$$

are sufficient to guarantee that $\mathbb{P}(\mathcal{E}) \leq \eta$.

We are going to prove that there exists $C' > 0$, such that

$$C'\frac{\sqrt{n_2}}{\log n_2} \leq \frac{n_2}{\frac{128\lambda\kappa}{3\varepsilon}(\frac{9}{2})^{L-1} \log \frac{34l_{\max}^2 s^{L-1}|\mathcal{V}_{\text{obs}}|^3}{\eta}}, \tag{H.59}$$

which is equivalent to

$$C' \frac{128\lambda\kappa}{3\varepsilon} \left(\frac{9}{2}\right)^{L-1} \log \frac{34l_{\max}^2 s^{L-1}|\mathcal{V}_{\mathrm{obs}}|^3}{\eta} \leq \sqrt{n_2} \log n_2. \tag{H.60}$$

Since $n_2$ is lower bounded as in (H.57), it is sufficient to show that there exists $C' > 0$, such that

$$(C')^2 \left(\frac{128\lambda\kappa}{3\varepsilon}\left(\frac{9}{2}\right)^{L-1} \log \frac{34l_{\max}^2 s^{L-1}|\mathcal{V}_{\mathrm{obs}}|^3}{\eta}\right)^2$$
$$\leq \frac{64\lambda^2\kappa^2}{c\varepsilon^2}\left(\frac{9}{2}\right)^{2L-2} \log \frac{17l_{\max}^2 s^{L-1}|\mathcal{V}_{\mathrm{obs}}|^3}{\eta} \log\left[\frac{64\lambda^2\kappa^2}{c\varepsilon^2}\left(\frac{9}{2}\right)^{2L-2} \log \frac{17l_{\max}^2 s^{L-1}|\mathcal{V}_{\mathrm{obs}}|^3}{\eta}\right],$$

which is equivalent to

$$(C')^2 \leq \frac{9}{256c} \frac{\log \frac{17l_{\max}^2 s^{L-1}|\mathcal{V}_{\mathrm{obs}}|^3}{\eta}}{\log \frac{34l_{\max}^2 s^{L-1}|\mathcal{V}_{\mathrm{obs}}|^3}{\eta}} \frac{\left(\log\left(\frac{64\lambda^2\kappa^2}{c\varepsilon^2}(\frac{9}{2})^{2L-2}\right) + \log\log \frac{17l_{\max}^2 s^{L-1}|\mathcal{V}_{\mathrm{obs}}|^3}{\eta}\right)^2}{\log \frac{34l_{\max}^2 s^{L-1}|\mathcal{V}_{\mathrm{obs}}|^3}{\eta}}. \tag{H.61}$$

We have

$$\log \frac{17l_{\max}^2 s^{L-1}|\mathcal{V}_{\mathrm{obs}}|^3}{\eta} \Big/ \log \frac{34l_{\max}^2 s^{L-1}|\mathcal{V}_{\mathrm{obs}}|^3}{\eta} > \frac{1}{2} \quad \text{and} \tag{H.62}$$

$$\frac{\left(\log\left(\frac{64\lambda^2\kappa^2}{c\varepsilon^2}(\frac{9}{2})^{2L-2}\right) + \log\log \frac{17l_{\max}^2 s^{L-1}|\mathcal{V}_{\mathrm{obs}}|^3}{\eta}\right)^2}{\log \frac{34l_{\max}^2 s^{L-1}|\mathcal{V}_{\mathrm{obs}}|^3}{\eta}} > \frac{\left(\log\left(\frac{64\lambda^2\kappa^2}{c\varepsilon^2}(\frac{9}{2})^{2L-2}\right)\right)^2}{\log \frac{34l_{\max}^2 s^{L-1}|\mathcal{V}_{\mathrm{obs}}|^3}{\eta}}. \tag{H.63}$$

Since

$$\lim_{L\to\infty} \frac{\left(\log\left(\frac{64\lambda^2\kappa^2}{c\varepsilon^2}(\frac{9}{2})^{2L-2}\right)\right)^2}{\log \frac{34l_{\max}^2 s^{L-1}|\mathcal{V}_{\mathrm{obs}}|^3}{\eta}} = +\infty, \tag{H.64}$$

we can see that there exists $C' > 0$ that satisfies inequality (H.59). $\qquad\square$

# I  Proofs of results in Section 3.3

**Theorem 7.** *If Assumptions 1 to 5 hold and all the nodes have exactly two children, RNJ constructs the correct latent tree with probability at least $1 - \eta$ if*

$$n_2 > \frac{16\lambda^2\kappa^2}{c\rho_{\min}^2} \log\left(\frac{2|\mathcal{V}_{\mathrm{obs}}|^2 l_{\max}^2}{\eta}\right) \tag{I.1}$$

$$\frac{n_2}{n_1} > \frac{64\lambda\kappa}{3\rho_{\min}} \log\left(\frac{4|\mathcal{V}_{\mathrm{obs}}|^2 l_{\max}^2}{\eta}\right) \tag{I.2}$$

*where*

$$\lambda = \frac{2l_{\max}^2 e^{\rho_{\max}/l_{\max}}}{\delta_{\min}^{1/l_{\max}}} \quad \text{and} \quad \kappa = \max\{\sigma_{\max}^2, \rho_{\min}\}, \tag{I.3}$$

*and $c$ is an absolute constant.*

*Proof of Theorem 7.* It is easy to see by substituting the constants $\lambda$ and $\kappa$ into (I.1) and (I.2) that Theorem 7 implies Theorem 2, so we provide the proof of Theorem 7 here.

With the sufficient condition in Proposition 5, we can bound the probability of error event by the union bound as follows

$$\mathbb{P}(\mathcal{E}) \leq \mathbb{P}\left(\max_{x_i,x_j \in \mathcal{V}_{\mathrm{obs}}} \left|\hat{\mathrm{d}}(x_i,x_j) - \mathrm{d}(x_i,x_j)\right| > \frac{\rho_{\min}}{2}\right) \tag{I.4}$$

$$\leq |\mathcal{V}_{\mathrm{obs}}|^2 \mathbb{P}\left(\left|\hat{\mathrm{d}}(x_i,x_j) - \mathrm{d}(x_i,x_j)\right| > \frac{\rho_{\min}}{2}\right). \tag{I.5}$$

We bound two terms in the tail probability separately as

$$2l_{\max}^2 e^{-\frac{3n_2}{64\lambda\kappa n_1}\rho_{\min}} < \frac{\eta}{2|\mathcal{V}_{\text{obs}}|^2} \tag{I.6}$$

$$l_{\max}^2 e^{-c\frac{n_2}{16\lambda^2\kappa^2}\rho_{\min}^2} < \frac{\eta}{2|\mathcal{V}_{\text{obs}}|^2}. \tag{I.7}$$

Then we have

$$n_2 > \frac{16\lambda^2\kappa^2}{c\rho_{\min}^2} \log\left(\frac{2|\mathcal{V}_{\text{obs}}|^2 l_{\max}^2}{\eta}\right), \tag{I.8}$$

$$\frac{n_2}{n_1} > \frac{64\lambda\kappa}{3\rho_{\min}} \log\left(\frac{4|\mathcal{V}_{\text{obs}}|^2 l_{\max}^2}{\eta}\right). \tag{I.9}$$

The proof that $n_1 = O(\sqrt{n_2}/\log n_2)$ can be derived by following the similar procedures in the proof of Theorem 1. $\square$

**Proposition 12.** *If Assumption 1 to 5 hold and the truncated inner product is adopted to estimate the information distances,*

$$\mathbb{P}\left(\|\hat{\mathbf{R}} - \mathbf{R}\|_2 > t\right) \leq |\mathcal{V}_{\text{obs}}|^2 f\left(e^{\rho_{\min}}\frac{t}{|\mathcal{V}_{\text{obs}}|}\right), \tag{I.10}$$

*where the function $f$ is defined as*

$$f(x) \triangleq 2l_{\max}^2 e^{-\frac{3n_2}{32\lambda\kappa n_1}x} + l_{\max}^2 e^{-c\frac{n_2}{4\lambda^2\kappa^2}x^2} = ae^{-wx} + be^{-ux^2}, \tag{I.11}$$

*with $\lambda = 2l_{\max}^2 e^{\rho_{\max}/l_{\max}}/\delta_{\min}^{1/l_{\max}}$, $w = \frac{3n_2}{32\lambda\kappa n_1}$, $u = c\frac{n_2}{4\lambda^2\kappa^2}$, $a = 2l_{\max}^2$ and $b = l_{\max}^2$.*

*Proof of Proposition 12.* Noting that $\mathbf{R}_{ij} = \exp\left(-\mathrm{d}(x_i, x_j)\right)$, we have

$$\mathbb{P}\left(|\hat{\mathbf{R}}_{ij} - \mathbf{R}_{ij}| > t\right) = \mathbb{P}\left(\left|\exp\left(-\hat{\mathrm{d}}(x_i, x_j)\right) - \exp\left(-\mathrm{d}(x_i, x_j)\right)\right| > t\right) \tag{I.12}$$

$$\overset{(a)}{\leq} \mathbb{P}\left(\left|\hat{\mathrm{d}}(x_i, x_j) - \mathrm{d}(x_i, x_j)\right| > e^{\rho_{\min}}t\right) \tag{I.13}$$

$$< f(e^{\rho_{\min}}t), \tag{I.14}$$

where inequality $(a)$ is derived from Taylor's Theorem.

Since

$$\|\hat{\mathbf{R}} - \mathbf{R}\|_2 \leq |\mathcal{V}_{\text{obs}}| \max_{i,j} |\hat{\mathbf{R}}_{ij} - \mathbf{R}_{ij}|, \tag{I.15}$$

we have

$$\mathbb{P}\left(\|\hat{\mathbf{R}} - \mathbf{R}\|_2 > t\right) \leq \mathbb{P}\left(\max_{i,j} |\hat{\mathbf{R}}_{ij} - \mathbf{R}_{ij}| > \frac{t}{|\mathcal{V}_{\text{obs}}|}\right) \leq |\mathcal{V}_{\text{obs}}|^2 f\left(e^{\rho_{\min}}\frac{t}{|\mathcal{V}_{\text{obs}}|}\right) \tag{I.16}$$

as desired. $\square$

**Theorem 8.** *If Assumptions 1 to 5 hold and all the nodes have exactly two children, RSNJ constructs the correct latent tree with probability at least $1 - \eta$ if*

$$n_2 \geq \frac{16\lambda^2\kappa^2|\mathcal{V}_{\text{obs}}|^2}{ce^{2\rho_{\min}}g(|\mathcal{V}_{\text{obs}}|, \rho_{\min}, \rho_{\max})^2} \log\frac{2|\mathcal{V}_{\text{obs}}|^2 l_{\max}^2}{\eta} \tag{I.17}$$

$$\frac{n_2}{n_1} \geq \frac{64\lambda\kappa|\mathcal{V}_{\text{obs}}|}{3e^{\rho_{\min}}g(|\mathcal{V}_{\text{obs}}|, \rho_{\min}, \rho_{\max})} \log\frac{4|\mathcal{V}_{\text{obs}}|^2 l_{\max}^2}{\eta} \tag{I.18}$$

*where*

$$g(x, \rho_{\min}, \rho_{\max}) = \begin{cases} \frac{1}{2}(2e^{-\rho_{\max}})^{\log_2(x/2)}e^{-\rho_{\max}}(1 - e^{-2\rho_{\min}}), & e^{-2\rho_{\max}} \leq 0.5 \\ e^{-3\rho_{\max}}(1 - e^{-2\rho_{\min}}), & e^{-2\rho_{\max}} > 0.5 \end{cases} \tag{I.19}$$

$$\lambda = \frac{2l_{\max}^2 e^{\rho_{\max}/l_{\max}}}{\delta_{\min}^{1/l_{\max}}} \quad \kappa = \max\{\sigma_{\max}^2, \rho_{\min}\}, \tag{I.20}$$

*and $c$ is an absolute constant.*

*Proof of Theorem 8.* It is easy to see by substituting the constants $\lambda$ and $\kappa$ into (I.17) and (I.18) that Theorem 8 implies Theorem 3, so we provide the proof of Theorem 8 here.

Proposition 6 shows that the probability of learning the wrong tree $\mathbb{P}(\mathcal{E})$ could be bounded as

$$\mathbb{P}(\mathcal{E}) \leq \mathbb{P}\big(\|\hat{\mathbf{R}} - \mathbf{R}\|_2 > g(|\mathcal{V}_{\mathrm{obs}}|, \rho_{\min}, \rho_{\max})\big) \leq |\mathcal{V}_{\mathrm{obs}}|^2 f\Big(e^{\rho_{\min}} \frac{g(|\mathcal{V}_{\mathrm{obs}}|, \rho_{\min}, \rho_{\max})}{|\mathcal{V}_{\mathrm{obs}}|}\Big). \quad \text{(I.21)}$$

Substituting the expression of $f$ and bounding the right-hand-side of inequality (I.21) by $\eta$, we have

$$n_2 \geq \frac{16\lambda^2 \kappa^2 |\mathcal{V}_{\mathrm{obs}}|^2}{ce^{2\rho_{\min}} g(|\mathcal{V}_{\mathrm{obs}}|, \rho_{\min}, \rho_{\max})^2} \log \frac{2|\mathcal{V}_{\mathrm{obs}}|^2 l_{\max}^2}{\eta} \quad \text{and} \quad \text{(I.22)}$$

$$\frac{n_2}{n_1} \geq \frac{64\lambda\kappa |\mathcal{V}_{\mathrm{obs}}|}{3e^{\rho_{\min}} g(|\mathcal{V}_{\mathrm{obs}}|, \rho_{\min}, \rho_{\max})} \log \frac{4|\mathcal{V}_{\mathrm{obs}}|^2 l_{\max}^2}{\eta}. \quad \text{(I.23)}$$

The proof that $n_1 = O(\sqrt{n_2}/\log n_2)$ can be derived by following the similar procedures in the proof of Theorem 1. $\qquad\square$

## J Proofs of results in Section 3.4

**Lemma 13.** *The MST of a weighted graph $\mathbb{T}$ has the following properties:*

*(1) For any cut $C$ of the graph, if the weight of an edge $e$ in the cut-set of $C$ is strictly smaller than the weights of all other edges of the cut-set of $C$, then this edge belongs to all MSTs of the graph.*

*(2) If $\mathbb{T}'$ is a tree of MST edges, then we can contract $\mathbb{T}'$ into a single vertex while maintaining the invariant that the MST of the contracted graph plus $\mathbb{T}'$ gives the MST for the graph before contraction [25].*

*Proof of Proposition 7.* We prove this argument by induction. Choosing any node as the root node, we first prove that the edges which are related to the observed nodes with the largest depth are identified or contracted correctly.

Since we consider the edges which involve at least one observed node, we only need to discuss the edges formed by two observed nodes and one observed node and one hidden node. We first consider the identification of the edges between two observed nodes.

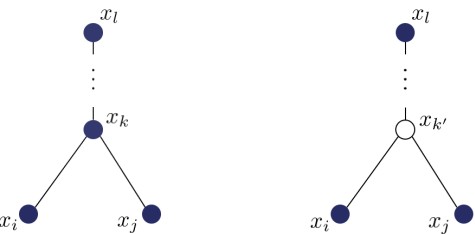

Figure 9: Two kinds of edges related at least one observed node.

To correctly identify the edge $(x_k, x_j)$ in Fig. 9, we consider the cut of the graph which splits the nodes into $\{x_j\}$ and all the other nodes. Lemma 13 says that the condition that

$$\hat{\mathrm{d}}(x_k, x_j) < \hat{\mathrm{d}}(x_l, x_j) \quad \forall x_l \in \mathcal{V}_{\mathrm{obs}}, x_l \neq x_k, x_j \quad \text{(J.1)}$$

is sufficient to guarantee that this edge is identified correctly. This condition is equivalent to

$$\Delta_{kj} < \Delta_{lj} + \mathrm{d}(x_l, x_k) \quad \forall x_l \in \mathcal{V}_{\mathrm{obs}}, x_l \neq x_k, x_j, \quad \text{(J.2)}$$

which is guaranteed by choosing $\Delta_{\mathrm{MST}} = \mathrm{d}_{\mathrm{ct}}(x_k; \mathbb{T}, \mathcal{V}_{\mathrm{obs}})$.

Furthermore, we need to guarantee that $x_j$ is not connected to other nodes except $x_k$. We consider the cut of the graph which split the nodes into $\{x_j, x_k\}$ and all the other nodes. Lemma 13 says that the condition that

$$\hat{\mathrm{d}}(x_l, x_k) < \hat{\mathrm{d}}(x_l, x_j) \quad \forall x_l \in \mathcal{V}_{\mathrm{obs}}, x_l \neq x_k, x_j \tag{J.3}$$

is sufficient to guarantee $x_j$ is not connected to other nodes. This condition is equivalent to

$$\Delta_{lk} < \Delta_{lj} + \mathrm{d}(x_k, x_j) \quad \forall x_l \in \mathcal{V}_{\mathrm{obs}}, x_l \neq x_k, x_j, \tag{J.4}$$

which is guaranteed by choosing $\Delta_{\mathrm{MST}} = \mathrm{d}_{\mathrm{ct}}(x_k; \mathbb{T}, \mathcal{V}_{\mathrm{obs}})$.

A similar proof can be used to guarantee $(x_i, x_k)$ can be identified correctly. Then we can contract $x_i, x_j$ to $x_k$ to form a super node in the subsequent edges identification for Lemma 13.

Now we discuss the edges involving one observed node and one hidden node. There are two cases: (i) The hidden node $x_{k'}$ should be contracted to either $x_i$ or $x_j$. (ii) The hidden node $x_{k'}$ should be contracted to $x_l \in \mathcal{V}_{\mathrm{obs}}, x_l \neq x_i, x_j$.

We first consider the case (i). Without loss of generality, we assume that $x_{k'}$ should be contracted to $x_j$. Contracting $x_{k'}$ to $x_j$ is equivalent to that $x_i$ is not connected to other nodes except $x_j$. Lemma 13 shows that

$$\hat{\mathrm{d}}(x_i, x_j) < \hat{\mathrm{d}}(x_i, x_l) \quad \hat{\mathrm{d}}(x_j, x_l) < \hat{\mathrm{d}}(x_i, x_l) \qquad \forall x_l \in \mathcal{V}_{\mathrm{obs}}, x_l \neq x_i, x_j \tag{J.5}$$

is sufficient to achieve that $x_i$ is not connected to other nodes except $x_j$. This condition is equivalent to

$$\Delta_{ij} + \mathrm{d}(x_{k'}, x_j) < \Delta_{lj} + \mathrm{d}(x_{k'}, x_l) \quad \text{and} \quad \Delta_{jl} + \mathrm{d}(x_{k'}, x_j) < \Delta_{il} + \mathrm{d}(x_{k'}, x_i)$$
$$\forall x_l \in \mathcal{V}_{\mathrm{obs}}, x_l \neq x_k, x_j, \tag{J.6}$$

which is guaranteed by choosing $\Delta_{\mathrm{MST}} = \mathrm{d}_{\mathrm{ct}}(x_{k'}; \mathbb{T}, \mathcal{V}_{\mathrm{obs}})$. Then we can contract $x_i$ to $x_j$ to form a super node in the subsequent edges identification for Lemma 13.

Then we consider the case (ii). Here we need to prove that $x_{k'}$ will not be contracted to $x_i$ or $x_j$. Without loss of generality, we assume that $x_{k'}$ is contracted to $x_l$, which guaranteed by that there is no edge between $x_i$ and $x_j$. Lemma 13 shows that

$$\hat{\mathrm{d}}(x_i, x_l) < \hat{\mathrm{d}}(x_i, x_j) \quad \hat{\mathrm{d}}(x_j, x_l) < \hat{\mathrm{d}}(x_i, x_j) \tag{J.7}$$

is sufficient to guarantee that there is no edge between $x_i$ and $x_j$. This condition is equivalent to

$$\Delta_{il} + \mathrm{d}(x_{k'}, x_l) < \Delta_{ij} + \mathrm{d}(x_{k'}, x_j) \quad \Delta_{jl} + \mathrm{d}(x_{k'}, x_l) < \Delta_{ij} + \mathrm{d}(x_{k'}, x_i), \tag{J.8}$$

which is guaranteed by choosing $\Delta_{\mathrm{MST}} = \mathrm{d}_{\mathrm{ct}}(x_{k'}; \mathbb{T}, \mathcal{V}_{\mathrm{obs}})$.

Assume that all the edges related to the nodes with depths larger than $l$ are identified or contracted correctly. We now consider the edges related to the nodes with depths $l$. For the edges between two observed nodes and edges of case (i) and (ii), similar procedures can be adopted to prove the statements. Here we discuss the case where $x_l$ should contract the hidden nodes which are its descendants. Contracting $x_{k'}$ to $x_l$ is equivalent to that there are edges between $(x_l, x_i)$ and $(x_l, x_j)$, and there is no other edges related to $x_i$ and $x_j$. Recall that condition (J.7) is satisfied by the induction hypothesis. Lemma 13 shows that

$$\hat{\mathrm{d}}(x_i, x_l) < \hat{\mathrm{d}}(x_i, x_k) \quad \hat{\mathrm{d}}(x_k, x_l) < \hat{\mathrm{d}}(x_k, x_i) \quad \forall x_k \in \mathcal{V}_{\mathrm{obs}}, x_k \neq x_i, x_j, x_l \tag{J.9}$$

$$\hat{\mathrm{d}}(x_j, x_l) < \hat{\mathrm{d}}(x_j, x_k) \quad \hat{\mathrm{d}}(x_k, x_l) < \hat{\mathrm{d}}(x_k, x_j) \quad \forall x_k \in \mathcal{V}_{\mathrm{obs}}, x_k \neq x_i, x_j, x_l \tag{J.10}$$

is sufficient to guarantee that $x_{k'}$ is contracted to $x_l$. This condition is equivalent to

$$\Delta_{il} < \Delta_{ik} + \mathrm{d}(x_k, x_l) \quad \Delta_{kl} < \Delta_{ki} + \mathrm{d}(x_i, x_l) \quad \forall x_k \in \mathcal{V}_{\mathrm{obs}}, x_k \neq x_i, x_j, x_l \tag{J.11}$$

$$\Delta_{jl} < \Delta_{jk} + \mathrm{d}(x_k, x_l) \quad \Delta_{kl} < \Delta_{kj} + \mathrm{d}(x_j, x_l) \quad \forall x_k \in \mathcal{V}_{\mathrm{obs}}, x_k \neq x_i, x_j, x_l \tag{J.12}$$

which is guaranteed by choosing $\Delta_{\mathrm{MST}} = \mathrm{d}_{\mathrm{ct}}(x_l; \mathbb{T}, \mathcal{V}_{\mathrm{obs}})$. Then we can contract $x_i$ and $x_j$ to $x_l$ to form a super node in the subsequent edges identification for Lemma 13.

Thus, the results are proved by induction. $\qquad\square$

**Theorem 9.** *If Assumptions 1–4, RCLRG constructs the correct latent tree with probability at least* $1 - \eta$ *if*

$$n_2 \geq \max\left\{\frac{4}{\varepsilon^2}\left(\frac{9}{2}\right)^{2L_\mathrm{C}-2}, \frac{1}{\Delta_\mathrm{MST}^2}\right\}\frac{16\lambda^2\kappa^2}{c}\log\frac{17l_\mathrm{max}^2 s^{L_\mathrm{C}-1}|\mathcal{V}_\mathrm{obs}|^3 + l_\mathrm{max}^2|\mathcal{V}_\mathrm{obs}|^2}{\eta}, \quad \text{(J.13)}$$

$$\frac{n_2}{n_1} \geq \max\left\{\frac{2}{\varepsilon}\left(\frac{9}{2}\right)^{L_\mathrm{C}-1}, \frac{1}{\Delta_\mathrm{MST}}\right\}\frac{64\lambda\kappa}{3}\log\frac{34l_\mathrm{max}^2 s^{L_\mathrm{C}-1}|\mathcal{V}_\mathrm{obs}|^3 + 2l_\mathrm{max}^2|\mathcal{V}_\mathrm{obs}|^2}{\eta}, \quad \text{(J.14)}$$

*where*

$$\lambda = \frac{2l_\mathrm{max}^2 e^{\rho_\mathrm{max}/l_\mathrm{max}}}{\delta_\mathrm{min}^{1/l_\mathrm{max}}} \quad \kappa = \max\{\sigma_\mathrm{max}^2, \rho_\mathrm{min}\} \quad s = d_\mathrm{max}^2 + 2d_\mathrm{max}^3(1 + 2N_\tau) \quad \varepsilon = \frac{\rho_\mathrm{min}}{2},$$
$$\text{(J.15)}$$

$c$ *is an absolute constant, and* $L_\mathrm{C}$ *is the number of iterations of RCLRG needed to construct the tree.*

*Proof of Theorem 9.* It is easy to see by substituting the constants $\lambda$, $\kappa$, $s$ and $\varepsilon$ into (J.13) and (J.14) that Theorem 9 implies Theorem 4, so we provide the proof of Theorem 9 here.

The RCLRG algorithm consists of two stages: Calculation of MST and implementation of RRG on internal nodes. The probability of error of RCLRG could be decomposed as

$$\mathbb{P}(\mathcal{E}) = \mathbb{P}\big(\mathcal{E}_\mathrm{MST} \cup (\mathcal{E}_\mathrm{MST}^c \cap \mathcal{E}_\mathrm{RRG})\big) = \mathbb{P}(\mathcal{E}_\mathrm{MST}) + \mathbb{P}(\mathcal{E}_\mathrm{MST}^c \cap \mathcal{E}_\mathrm{RRG}) \leq \mathbb{P}(\mathcal{E}_\mathrm{MST}) + \mathbb{P}(\mathcal{E}_\mathrm{RRG})$$

We define the correct event of calculation of the MST as

$$\mathcal{C}_\mathrm{MST} = \bigcap_{x_i, x_j \in \mathcal{V}_\mathrm{obs}}\left\{|\hat{\mathrm{d}}(x_i, x_j) - \mathrm{d}(x_i, x_j)| < \frac{\Delta_\mathrm{MST}}{2}\right\} = \bigcap_{x_i, x_j \in \mathcal{V}_\mathrm{obs}}\mathcal{C}_{ij} \quad \text{(J.16)}$$

Proposition 7 shows that

$$\mathbb{P}(\mathcal{E}_\mathrm{MST}) \leq 1 - \mathbb{P}(\mathcal{C}_\mathrm{MST}) = \mathbb{P}\big((\bigcap_{x_i, x_j \in \mathcal{V}_\mathrm{obs}}\mathcal{C}_{ij})^c\big) \quad \text{(J.17)}$$

$$= \mathbb{P}\bigg(\bigcup_{x_i, x_j \in \mathcal{V}_\mathrm{obs}}\mathcal{C}_{ij}^c\bigg) \leq \sum_{x_i, x_j \in \mathcal{V}_\mathrm{obs}}\mathbb{P}(\mathcal{C}_{ij}^c) \leq \binom{|\mathcal{V}_\mathrm{obs}|}{2}f\Big(\frac{\Delta_\mathrm{MST}}{2}\Big) \quad \text{(J.18)}$$

We define the event that RRG yields the correct subtree based on $\mathrm{nbd}[x_i, \mathbb{T}]$

$$\mathcal{C}_\mathrm{RRG} = \bigcap_{x_i \in \mathrm{Int}(\mathrm{MST}(\mathcal{V}_\mathrm{obs}; \hat{\mathbf{D}}))}\{\text{Output of RRG is correct with input } \mathrm{nbd}[x_i, \mathbb{T}]\} \quad \text{(J.19)}$$

$$= \bigcap_{x_i \in \mathrm{Int}(\mathrm{MST}(\mathcal{V}_\mathrm{obs}; \hat{\mathbf{D}}))}\mathcal{C}_i \quad \text{(J.20)}$$

Then we have

$$\mathbb{P}(\mathcal{E}_\mathrm{RRG}) = 1 - \mathbb{P}(\mathcal{C}_\mathrm{RRG}) = \mathbb{P}\bigg(\Big(\bigcap_{x_i \in \mathrm{Int}(\mathrm{MST}(\mathcal{V}_\mathrm{obs}; \hat{\mathbf{D}}))}\mathcal{C}_i\Big)^c\bigg) = \mathbb{P}\bigg(\bigcup_{x_i \in \mathrm{Int}(\mathrm{MST}(\mathcal{V}_\mathrm{obs}; \hat{\mathbf{D}}))}\mathcal{C}_i^c\bigg). \quad \text{(J.21)}$$

By defining $L_\mathrm{C} = \lceil\frac{\mathrm{Deg}(\mathrm{MST}(\mathcal{V}_\mathrm{obs}; \hat{\mathbf{D}}))}{2} - 1\rceil$, we have

$$\mathbb{P}(\mathcal{E}) \leq \mathbb{P}(\mathcal{E}_\mathrm{MST}) + \mathbb{P}(\mathcal{E}_\mathrm{RRG}) \quad \text{(J.22)}$$

$$\leq \binom{|\mathcal{V}_\mathrm{obs}|}{2}f\Big(\frac{\Delta_\mathrm{MST}}{2}\Big) + (|\mathcal{V}_\mathrm{obs}| - 2)\bigg\{\Big[4\Big(1 + \frac{|\mathcal{V}_\mathrm{obs}|^2}{2(s-1)}\Big) + \frac{3}{2}(d_\mathrm{max} - 1)\Big(1 + \frac{|\mathcal{V}_\mathrm{obs}|}{2(s-1)}\Big)\Big]$$

$$\times h^{(L_\mathrm{C}-1)}\Big(\frac{2\rho_\mathrm{min} - \varepsilon}{4}\Big) + \Big[3\Big(1 + \frac{|\mathcal{V}_\mathrm{obs}|^2}{4(s-1)}\Big) + 2(d_\mathrm{max} - 1)\Big(1 + \frac{|\mathcal{V}_\mathrm{obs}|^3}{8(s-1)}\Big)\Big]h^{(L_\mathrm{C}-1)}\Big(\frac{\varepsilon}{4}\Big)\bigg\}$$

$$\text{(J.23)}$$

To derive the sufficient conditions of $\mathbb{P}(\mathcal{E}) \leq \eta$, we consider the following conditions

$$\mathbb{P}(\mathcal{E}_{\mathrm{MST}}) \leq (1-r)\eta \quad \text{and} \quad \mathbb{P}(\mathcal{E}_{\mathrm{RRG}}) \leq r\eta \quad \text{for some} \quad r \in (0,1) \tag{J.24}$$

Following the same calculations with inequalities (13), we have

$$n_2 \geq \max \left\{ \frac{64\lambda^2\kappa^2}{c\varepsilon^2} \left(\frac{9}{2}\right)^{2L_{\mathrm{C}}-2} \log \frac{17l_{\max}^2 s^{L_{\mathrm{C}}-1}|\mathcal{V}_{\mathrm{obs}}|^3}{r\eta}, \frac{16\lambda^2\kappa^2}{c\Delta_{\mathrm{MST}}^2} \log \frac{l_{\max}^2|\mathcal{V}_{\mathrm{obs}}|^2}{(1-r)\eta} \right\} \tag{J.25}$$

$$\frac{n_2}{n_1} \geq \max \left\{ \frac{128\lambda\kappa}{3\varepsilon} \left(\frac{9}{2}\right)^{L_{\mathrm{C}}-1} \log \frac{34l_{\max}^2 s^{L_{\mathrm{C}}-1}|\mathcal{V}_{\mathrm{obs}}|^3}{r\eta}, \frac{64\lambda\kappa}{3\Delta_{\mathrm{MST}}} \log \frac{2l_{\max}^2|\mathcal{V}_{\mathrm{obs}}|^2}{(1-r)\eta} \right\} \tag{J.26}$$

By choosing $r = \frac{17s^{L_{\mathrm{C}}-1}|\mathcal{V}_{\mathrm{obs}}|^3}{17s^{L_{\mathrm{C}}-1}|\mathcal{V}_{\mathrm{obs}}|^3+|\mathcal{V}_{\mathrm{obs}}|^2}$, we have

$$n_2 \geq \max \left\{ \frac{4}{\varepsilon^2} \left(\frac{9}{2}\right)^{2L_{\mathrm{C}}-2}, \frac{1}{\Delta_{\mathrm{MST}}^2} \right\} \frac{16\lambda^2\kappa^2}{c} \log \frac{17l_{\max}^2 s^{L_{\mathrm{C}}-1}|\mathcal{V}_{\mathrm{obs}}|^3 + l_{\max}^2|\mathcal{V}_{\mathrm{obs}}|^2}{\eta}, \tag{J.27}$$

$$\frac{n_2}{n_1} \geq \max \left\{ \frac{2}{\varepsilon} \left(\frac{9}{2}\right)^{L_{\mathrm{C}}-1}, \frac{1}{\Delta_{\mathrm{MST}}} \right\} \frac{64\lambda\kappa}{3} \log \frac{34l_{\max}^2 s^{L_{\mathrm{C}}-1}|\mathcal{V}_{\mathrm{obs}}|^3 + 2l_{\max}^2|\mathcal{V}_{\mathrm{obs}}|^2}{\eta} \tag{J.28}$$

Following a similar proof as that for RRG, we claim that

$$n_2 \geq \max \left\{ \frac{4}{\varepsilon^2} \left(\frac{9}{2}\right)^{2L_{\mathrm{C}}-2}, \frac{1}{\Delta_{\mathrm{MST}}^2} \right\} \frac{16\lambda^2\kappa^2}{c} \log \frac{17l_{\max}^2 s^{L_{\mathrm{C}}-1}|\mathcal{V}_{\mathrm{obs}}|^3 + l_{\max}^2|\mathcal{V}_{\mathrm{obs}}|^2}{\eta}, \tag{J.29}$$

$$n_1 = O\left(\frac{\sqrt{n_2}}{\log n_2}\right) \tag{J.30}$$

are sufficient to guarantee $\mathbb{P}(\mathcal{E}) \leq \eta$. $\qquad \square$

## K Discussions and Proofs of results in Section 3.5

In this section, we provide more discussions of the results in Table 1. We also provide the proofs of results listed in Table 1.

The sample complexities of RRG and RCLRG are achieved w.h.p., since the number of iterations $L_{\mathrm{R}}$ and $L_{\mathrm{C}}$ depend on the quality of the estimates of the information distances. The parameter $t$ for RSNJ scales as $O(\frac{1}{l_{\max}} + \log |\mathcal{V}_{\mathrm{obs}}|)$. For the dependence on $\mathrm{Diam}(\mathbb{T})$, RRG and RSNJ have the worst performance. This is because RRG constructs new hidden nodes and estimates the information distances related to them in each iteration (or layer), which results in more severe error propagation on larger and deeper graphs. In contrast, our impossibility result in Theorem 5 suggests that RNJ has the optimal dependence on $\mathrm{Diam}(\mathbb{T})$. RCLRG also has the optimal dependence on the diameter of graphs on HMM, which demonstrates that the Chow-Liu initialization procedure greatly reduces the sample complexity from $O\left((\frac{9}{2})^{\mathrm{Diam}(\mathbb{T})}\right)$ to $O\left(\log \mathrm{Diam}(\mathbb{T})\right)$. Since the dependence on $\rho_{\max}$ only relies on the parameters, the dependence of $\rho_{\max}$ of all these algorithms remains the same for graphical models with different underlying structures. RRG, RCLRG and RNJ have the same dependence $O(e^{2\frac{\rho_{\max}}{l_{\max}}})$, while RSNJ has a worse dependence on $\rho_{\max}$.

### K.1 Proofs of entries in Table 1

**Double-binary tree** For RRG, the number of iterations needed to construct the tree $L_{\mathrm{R}} = \frac{1}{2}(\mathrm{Diam}(\mathbb{T})-1)$. Thus, the sample complexity of RRG is $O\left(e^{2\frac{\rho_{\max}}{l_{\max}}} (\frac{9}{2})^{\mathrm{Diam}(\mathbb{T})}\right)$.

For RCLRG, as mentioned previously, the MST can be obtained by contracting the hidden nodes to its closest observed node. For example, the MST of the double-binary tree with $\mathrm{Diam}(\mathbb{T}) = 5$ could be derived by contracting hidden nodes as Fig. 10. Then $L_{\mathrm{C}} = \lceil \frac{\mathrm{Diam}(\mathbb{T})+1}{4} \rceil - 1$, and the number of observed nodes is $|\mathcal{V}_{\mathrm{obs}}| = 2^{\frac{\mathrm{Diam}(\mathbb{T})+1}{2}}$. Thus, the sample complexity is $O\left(e^{2\frac{\rho_{\max}}{l_{\max}}} (\frac{9}{2})^{\frac{\mathrm{Diam}(\mathbb{T})}{2}}\right)$.

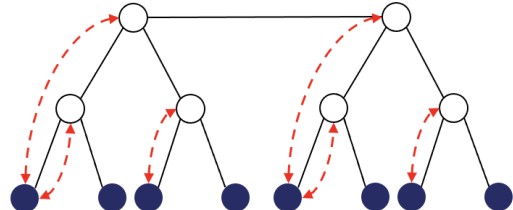

Figure 10: The contraction of hidden nodes in double-binary trees.

For RSNJ, the number of observed nodes is $|\mathcal{V}_{\mathrm{obs}}| = 2^{\frac{\mathrm{Diam}(\mathbb{T})+1}{2}}$, so the sample complexity is $O\big(e^{2t\rho_{\max}}\mathrm{Diam}(\mathbb{T})\big)$.

For RNJ, the number of observed nodes is $|\mathcal{V}_{\mathrm{obs}}| = 2^{\frac{\mathrm{Diam}(\mathbb{T})+1}{2}}$, so the sample complexity is $O\big(e^{2\frac{\rho_{\max}}{l_{\max}}}\mathrm{Diam}(\mathbb{T})\big)$.

**HMM** For RRG, the number of iterations needed to construct the tree $L_{\mathrm{R}} = \lceil \frac{\mathrm{Diam}\mathbb{T}}{2} - 1 \rceil$. Thus, the sample complexity of RRG is $O\big(e^{2\frac{\rho_{\max}}{l_{\max}}}(\frac{9}{2})^{\mathrm{Diam}(\mathbb{T})}\big)$.

For RCLRG, MST could be derived as contracting hidden nodes as shown in Fig. 11. Then $L_{\mathrm{C}} = 1$ and $|\mathcal{V}_{\mathrm{obs}}| = \mathrm{Diam}(\mathbb{T}) + 1$. The sample complexity is thus $O\big(e^{2\frac{\rho_{\max}}{l_{\max}}}\log\mathrm{Diam}(\mathbb{T})\big)$.

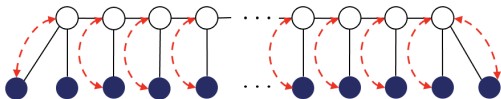

Figure 11: The contraction of hidden nodes in HMMs.

For RSNJ, the number of observed nodes is $|\mathcal{V}_{\mathrm{obs}}| = \mathrm{Diam}(\mathbb{T}) + 1$, so the sample complexity is $O\big(e^{2t\rho_{\max}}\log\mathrm{Diam}(\mathbb{T})\big)$.

For RNJ, the number of observed nodes is $|\mathcal{V}_{\mathrm{obs}}| = \mathrm{Diam}(\mathbb{T}) + 1$, so the sample complexity is $O\big(e^{2\frac{\rho_{\max}}{l_{\max}}}\log\mathrm{Diam}(\mathbb{T})\big)$.

**Full $m$-tree** For RRG, the number of iterations needed to construct the tree $L_{\mathrm{R}} = \frac{1}{2}\mathrm{Diam}(\mathbb{T})$. Thus, the sample complexity of RRG is $O\big(e^{2\frac{\rho_{\max}}{l_{\max}}}(\frac{9}{2})^{\mathrm{Diam}(\mathbb{T})}\big)$.

For RCLRG, the MST can be derived by contracting hidden nodes as shown in Fig. 12. Then $L_{\mathrm{C}} = 2$ and $|\mathcal{V}_{\mathrm{obs}}| = m^{\mathrm{Diam}(\mathbb{T})/2}$. Thus, its sample complexity is $O\big(e^{2\frac{\rho_{\max}}{l_{\max}}}\mathrm{Diam}(\mathbb{T})\big)$.

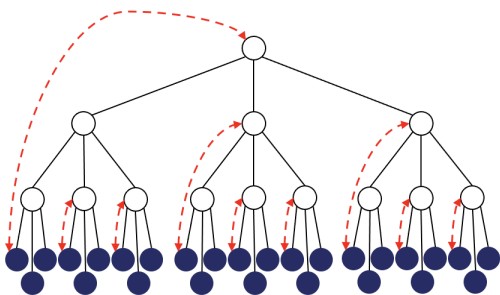

Figure 12: The contraction of hidden nodes in full $m$-trees.

**Double star**   For RRG, the number of iterations needed to construct the tree $L_R = 1$. Thus, the sample complexity of RRG is $O(e^{2\frac{\rho_{\max}}{t_{\max}}})$.

For RCLRG, the maximum number of iterations over each RRG step (over each internal node of the constructed Chow-Liu tree) in RCLRG is. $L_C = 1$ and $|\mathcal{V}_{\mathrm{obs}}| = 2d_{\max}$, so the sample complexity of RCLRG is $O(e^{2\frac{\rho_{\max}}{t_{\max}}}\log d_{\max})$.

## L   Additional numerical details and results

### L.1   Standard deviations of results in Fig. 2

We first report the standard deviations of the results presented in Fig. 2 in the main paper. All results are averaged over 100 independent runs.

Constant magnitude corruptions (Fig. 2(a))

| $\sigma/(\sigma/\text{AVG})\times 100$     # Samples
Algorithm | 500 | 1000 | 1500 | 2000 | 5000 | 10000 | 20000 |
|---|---|---|---|---|---|---|---|
| RRG | 9.7/9.3 | 4.4/5.0 | 3.7/4.5 | 4.0/5.0 | 5.0/6.8 | 14.4/24.1 | 21.0/75.0 |
| RSNJ | 3.3/3.8 | 3.0/7.0 | 3.9/28.8 | 0.3/703.5 | 0.0/0.0 | 0.0/0.0 | 0.0/0.0 |
| RCLRG | 2.1/52.0 | 0.5/229.1 | 0.0/0.0 | 0.0/0.0 | 0.0/0.0 | 0.0/0.0 | 0.0/0.0 |
| RNJ | 5.6/4.3 | 9.0/7.1 | 12.3/10.0 | 17.1/15.1 | 28.4/28.3 | 35.4/47.9 | 32.2/68.1 |
| RG | 9.2/9.5 | 8.8/8.8 | 8.3/8.3 | 7.8/7.8 | 5.7/6.2 | 4.1/4.9 | 1.9/2.3 |
| SNJ | 0.4/0.3 | 0.6/0.4 | 1.4/0.9 | 2.7/1.8 | 3.2/2.6 | 3.6/3.7 | 3.2/5.9 |
| CLRG | 3.0/2.2 | 3.5/2.6 | 3.4/2.5 | 4.0/3.0 | 11.2/19.9 | 4.7/22.4 | 2.1/43.5 |
| NJ | 1.8/1.3 | 2.2/1.6 | 2.2/1.5 | 3.1/2.3 | 6.0/4.8 | 11.5/9.8 | 17.9/16.35 |

Table 2: The standard deviations and standard deviations divided by the means of the Robinson-Foulds distances for different algorithms

Uniform corruptions (Fig. 2(b))

| $\sigma/(\sigma/\text{AVG})\times 100$     # Samples
Algorithm | 500 | 1000 | 1500 | 2000 | 5000 | 10000 | 20000 |
|---|---|---|---|---|---|---|---|
| RRG | 4.5/5.0 | 3.3/4.0 | 3.8/4.6 | 3.1/4.0 | 4.3/5.9 | 10.9/17.4 | 23.0/103.2 |
| RSNJ | 3.3/4.0 | 2.9/6.7 | 5.0/30.1 | 0.7/230.3 | 0.0/0.0 | 0.0/0.0 | 0.0/0.0 |
| RCLRG | 4.6/9.7 | 2.5/35.8 | 0.6/197.1 | 0.1/1971.0 | 0.0/0.0 | 0.0/0.0 | 0.0/0.0 |
| RNJ | 9.2/6.9 | 11.5/9.4 | 16.4/14.1 | 18.7/16.6 | 31.1/35.0 | 31.4/50.9 | 33.7/74.5 |
| RG | 9.2/9.0 | 9.8/9.6 | 8.0/7.8 | 8.1/8.0 | 9.0/9.0 | 7.8/7.4 | 5.9/6.1 |
| SNJ | 0.0/0.0 | 0.0/0.0 | 0.0/0.0 | 0.2/0.1 | 0.5/0.3 | 4.4/3.0 | 3.5/3.0 |
| CLRG | 3.3/2.4 | 3.4/2.5 | 3.3/2.4 | 3.5/2.5 | 3.0/2.2 | 6.0/4.5 | 8.0/17.5 |
| NJ | 1.7/1.2 | 1.9/1.3 | 2.0/1.4 | 2.0/1.4 | 2.1/1.5 | 3.9/2.8 | 6.1/4.8 |

Table 3: The standard deviations and standard deviations divided by the means of the Robinson-Foulds distances for different algorithms

HMM corruptions (Fig. 2(c))

| $\sigma/(\sigma/\text{AVG})\times 100$     # Samples
Algorithm | 500 | 1000 | 1500 | 2000 | 5000 | 10000 | 20000 |
|---|---|---|---|---|---|---|---|
| RRG | 4.0/4.5 | 5.3/5.8 | 3.8/4.5 | 3.3/4.0 | 3.4/4.6 | 7.5/10.9 | 21.1/49.6 |
| RSNJ | 5.9/6.0 | 3.5/6.4 | 3.4/9.7 | 3.6/35.2 | 0.0/0.0 | 0.0/0.0 | 0.0/0.0 |
| RCLRG | 13.1/17.4 | 4.9/14.1 | 3.4/29.1 | 1.6/54.6 | 0.0/0.0 | 0.0/0.0 | 0.0/0.0 |
| RNJ | 6.7/4.5 | 11.3/8.9 | 12.7/10.5 | 19.2/16.7 | 29.3/30.7 | 38.0/48.7 | 32.3/65.4 |
| RG | 9.3/9.1 | 8.4/8.3 | 8.7/8.5 | 8.6/8.3 | 9.0/8.8 | 8.6/8.2 | 5.5/5.7 |
| SNJ | 0.3/0.2 | 0.4/0.3 | 0.4/0.3 | 0.5/0.3 | 2.0/1.2 | 4.8/3.4 | 3.9/3.2 |
| CLRG | 3.4/2.5 | 3.3/2.4 | 3.2/2.3 | 3.1/2.3 | 3.5/2.6 | 15.0/12.7 | 8.2/13.7 |
| NJ | 1.8/1.3 | 1.6/1.2 | 2.0/1.4 | 1.9/1.3 | 2.8/2.0 | 4.5/3.3 | 5.7/4.4 |

Table 4: The standard deviations and standard deviations divided by the means of the Robinson-Foulds distances for different algorithms

We note that most of the standard deviations (relative to the means) are reasonably small. However, some entries in Tables 2–4 appear to be rather large, for example $0.5/229.1$. The reason is that the

mean value of the errors are already quite small in these cases, so any deviation from the small means result in large standard deviations. This, however, seems unavoidable.

## L.2 More simulation results complementing those in Section 3.6

In the following more extensive simulations, we consider eight corruption patterns:

- Uniform corruptions: Uniform corruptions are independent additive noises in $[-2A, 2A]$ and distributed randomly in the data matrix $\mathbf{X}_1^n$.

- Constant magnitude corruptions: Constant magnitude corruptions are independent additive noises but taking values in $\{-A, +A\}$ with probability $0.5$ and distributed randomly in $\mathbf{X}_1^n$.

- Gaussian corruptions: Gaussian corruptions are independent additive Gaussian noises $\mathcal{N}(0, A^2)$ and distributed randomly in $\mathbf{X}_1^n$.

- HMM corruptions: HMM corruptions are generated by a HMM which shares the same structure as the original HMM but has different parameters. They replace the entries in $\mathbf{X}_1^n$ with the samples generated by the variables in the same positions.

- Double binary corruptions: Double binary corruptions are generated by a double binary tree-structured graphical model which shares the same structure as the original double binary graphical model but has different parameters. They replace the entries in $\mathbf{X}_1^n$ with the samples generated by the variables in the same positions.

- Gaussian outliers: Gaussian outliers are outliers that are generated by independent Gaussian random variables distributed as $\mathcal{N}(0, A^2)$.

- HMM outliers: HMM outliers are outliers that are generated by a HMM that shares the same structure as the original HMM but has different parameters.

- Double binary outliers: Double binary outliers are outliers that are generated by a double binary tree-structured graphical model which shares the same structure as the original HMM but has different parameters.

In all our experiments, the parameter $A$ is set to $60$ and the number of corruptions $n_1$ is set to $100$.

Samples are generated from two graphical models: HMM (Fig. 5(b)) and double binary tree (Fig. 5(a)). The dimensions of the random vectors at each node are $l_{\max} = 3$. The Robinson-Foulds distance [21] between the nominal tree and the estimate and the error rate (zero-one loss) are adopted to measure the performance of learning algorithms. These are computed based on $100$ independent trials. We use the code for RG and CLRG provided by Choi et al. [4]. All our experiments are run on an Intel(R) Xeon(R) CPU E5-2697 v4 @ 2.30 GHz.

### L.2.1 HMM

Just as in the experiments in Choi et al. [4], the diameter of the HMM (Fig. 5(b)) is chosen to be $\mathrm{Diam}(\mathbb{T}) = 80$. The matrices $(\mathbf{A}, \mathbf{\Sigma}_{\mathrm{r}}, \mathbf{\Sigma}_{\mathrm{n}})$ are chosen so that the condition in Proposition 14 are satisfied with $\alpha = 1$, and we set $\mathbf{A}$ commutable with $\mathbf{\Sigma}_{\mathrm{r}}$. The information distances between neighboring nodes are chosen to be the same value $0.24$, which implies that $\rho_{\min} = 0.24$ and $\rho_{\max} = 0.24 \cdot \mathrm{Diam}(\mathbb{T}) = 19.2$.

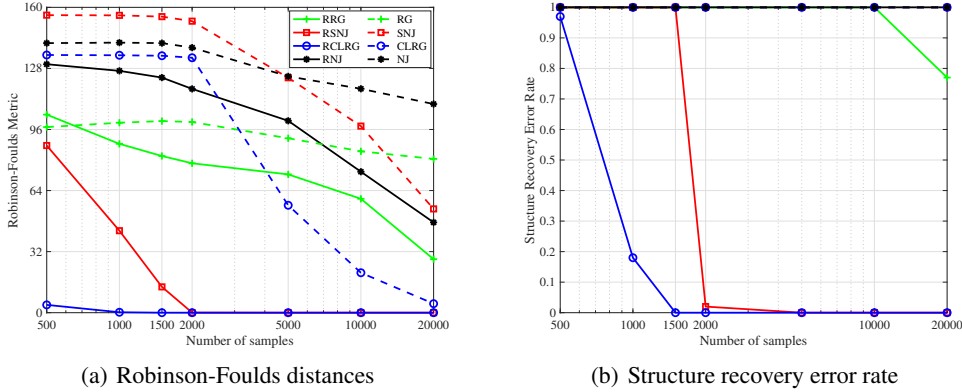

(a) Robinson-Foulds distances

(b) Structure recovery error rate

Figure 13: Performances of robustified and original learning algorithms with constant magnitude corruptions

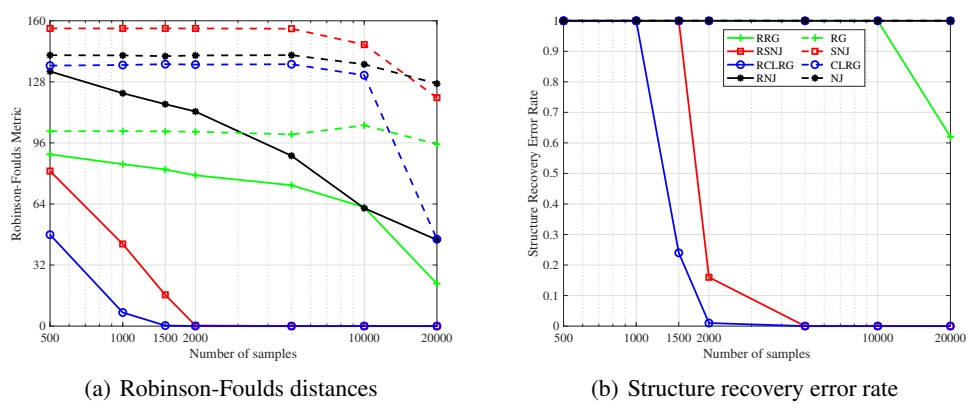

(a) Robinson-Foulds distances

(b) Structure recovery error rate

Figure 14: Performances of robustified and original learning algorithms with uniform corruptions

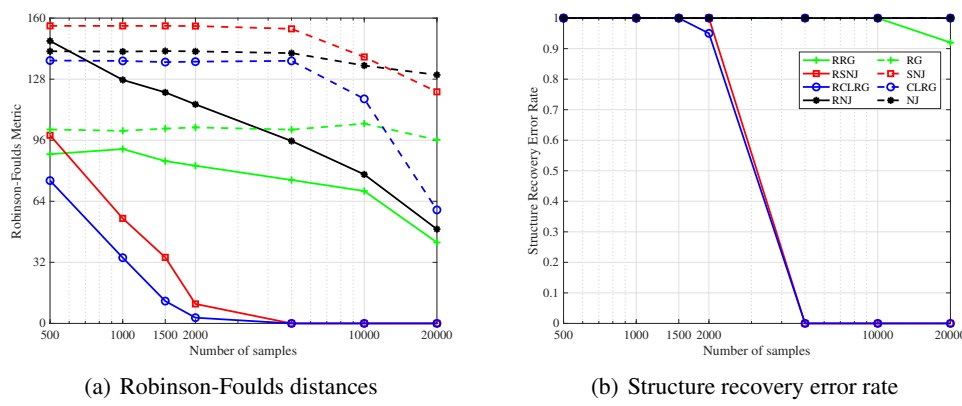

(a) Robinson-Foulds distances

(b) Structure recovery error rate

Figure 15: Performances of robustified and original learning algorithms with HMM corruptions

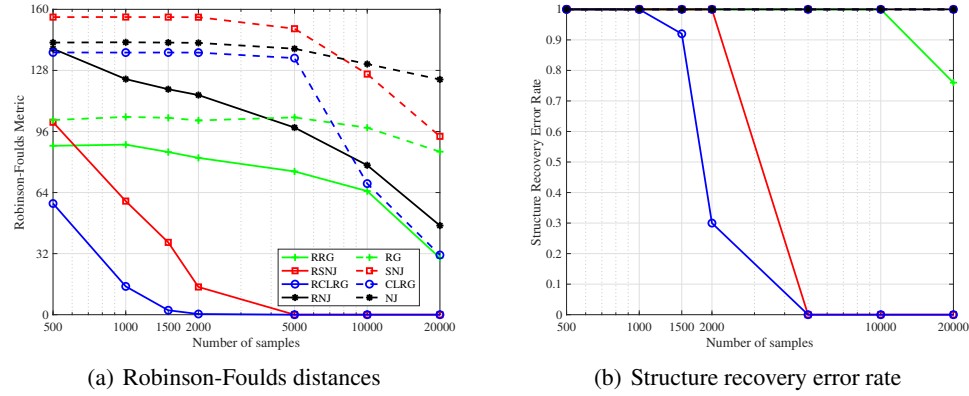

(a) Robinson-Foulds distances

(b) Structure recovery error rate

Figure 16: Performances of robustified and original learning algorithms with Gaussian corruptions

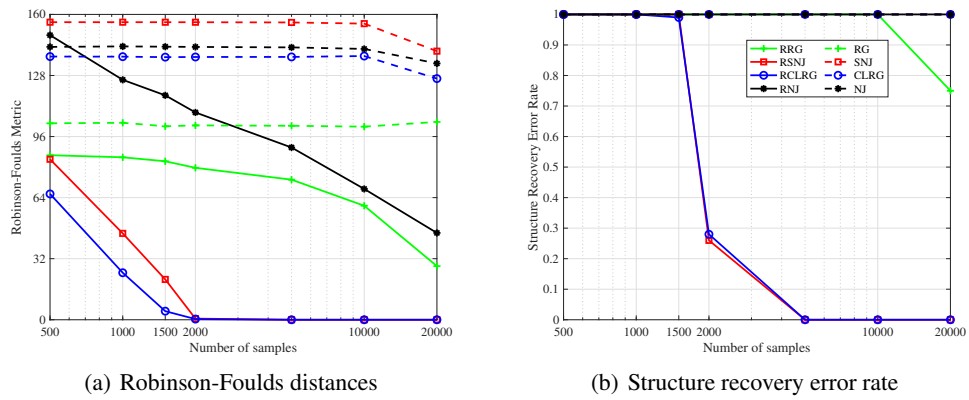

(a) Robinson-Foulds distances

(b) Structure recovery error rate

Figure 17: Performances of robustified and original learning algorithms with double binary corruptions

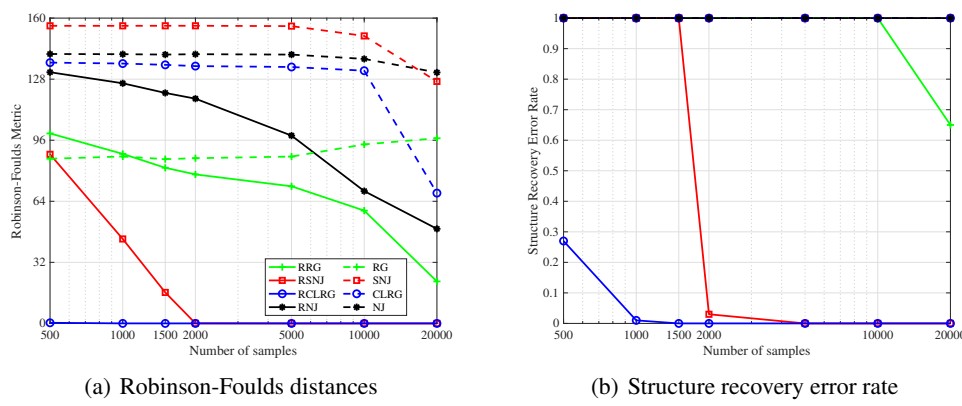

(a) Robinson-Foulds distances

(b) Structure recovery error rate

Figure 18: Performances of robustified and original learning algorithms with Gaussian outliers

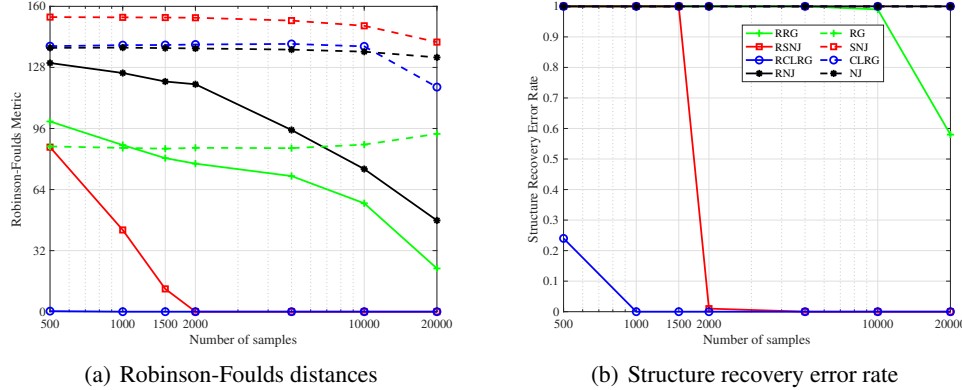

(a) Robinson-Foulds distances       (b) Structure recovery error rate

Figure 19: Performances of robustified and original learning algorithms with HMM outliers

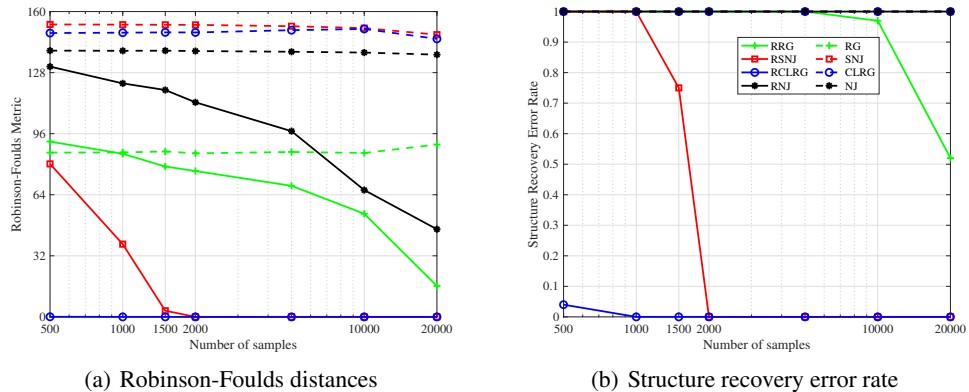

(a) Robinson-Foulds distances       (b) Structure recovery error rate

Figure 20: Performances of robustified and original learning algorithms with double binary outliers

These figures show that for the HMM, RCLRG performs best among all these algorithms. The reason is that the Chow-Liu initialization greatly reduces the effective depth of the original tree, which mitigates the error propagation. These simulation results also corroborate the effectiveness of the truncated inner product in combating any form of corruptions. We observe that the errors of robustified algorithms are significantly less that those of original algorithms.

Table 1 shows that for the HMM, RCLRG and RNJ both have optimal dependence on the diameter of the tree. In fact, by changing the parameters $\rho_{\min}$ and $\rho_{\max}$, we find that RNJ can sometimes perform better than RCLRG when $\rho_{\min}$ and $\rho_{\max}$ are both very small. In the experiments shown above, the parameters favor RCLRG.

Finally, it is also instructive to observe the effect of the different corruption patterns. By comparing the simulation results of HMM (resp. Gaussian and double binary) corruptions and HMM (resp. Gaussian and double binary) outliers, we can see that the algorithms perform worse in the presence of HMM (resp. Gaussian and double binary) corruptions. Since the truncated inner product truncates the samples with large absolute values, if corruptions appear in the *same* positions for all the samples, i.e., they appear as outliers, it is easier for the truncated inner product to identify these outliers and truncate them, resulting in higher quality estimates.

### L.2.2 Double binary tree

The diameter of the double binary tree (Fig. 5(a)) is $\mathrm{Diam}(\mathbb{T}) = 11$. The matrices $(\mathbf{A}, \mathbf{\Sigma}_{\mathrm{r}}, \mathbf{\Sigma}_{\mathrm{n}})$ are chosen so that the condition in Proposition 14 are satisfied with $\alpha = 1$, and we set $\mathbf{A}$ commutable

with $\mathbf{\Sigma}_r$. The information distance between neighboring nodes is 1, which implies that $\rho_{\min} = 1$ and $\rho_{\max} = \text{Diam}(\mathbb{T}) = 11$.

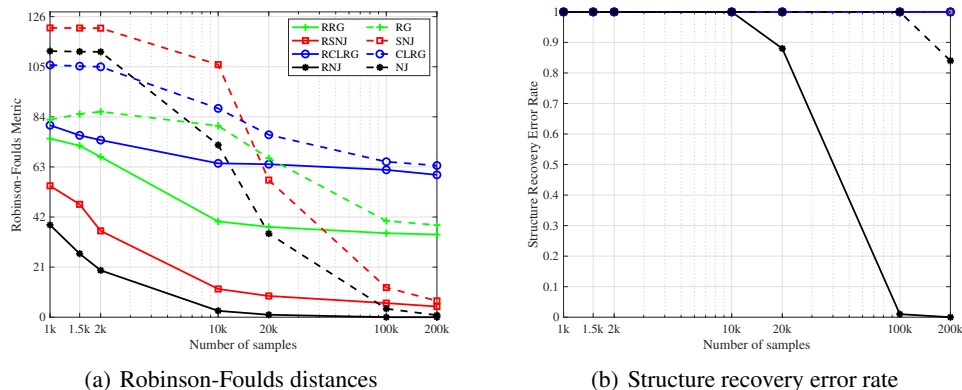

(a) Robinson-Foulds distances

(b) Structure recovery error rate

Figure 21: Performances of robustified and original learning algorithms with constant magnitude corruptions

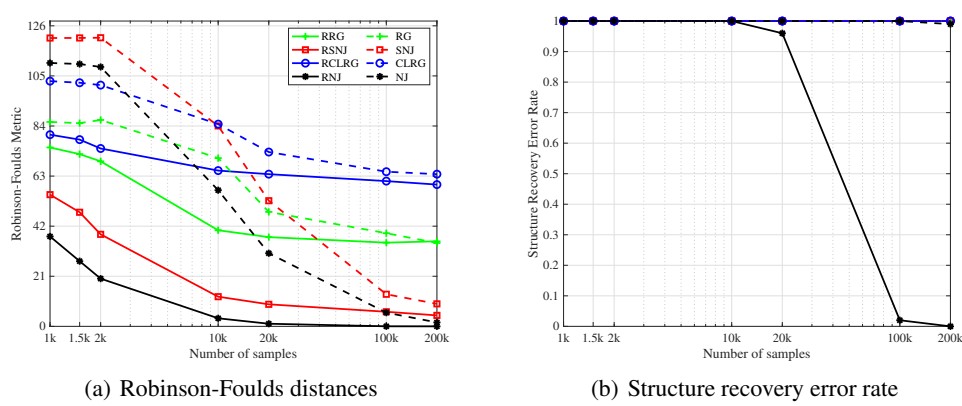

(a) Robinson-Foulds distances

(b) Structure recovery error rate

Figure 22: Performances of robustified and original learning algorithms with uniform corruptions

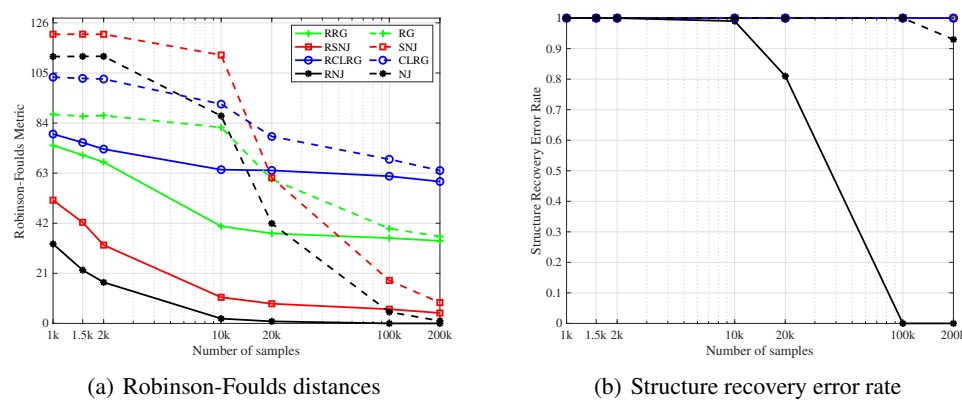

(a) Robinson-Foulds distances

(b) Structure recovery error rate

Figure 23: Performances of robustified and original learning algorithms with Gaussian corruptions

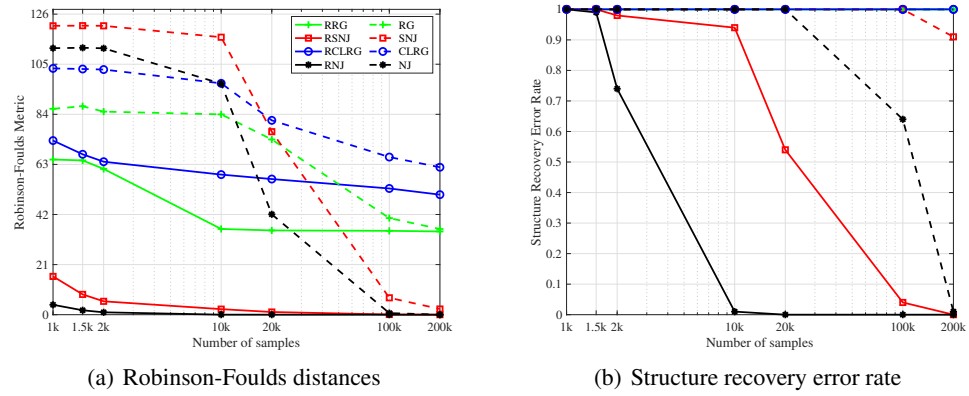

(a) Robinson-Foulds distances

(b) Structure recovery error rate

Figure 24: Performances of robustified and original learning algorithms with HMM corruptions

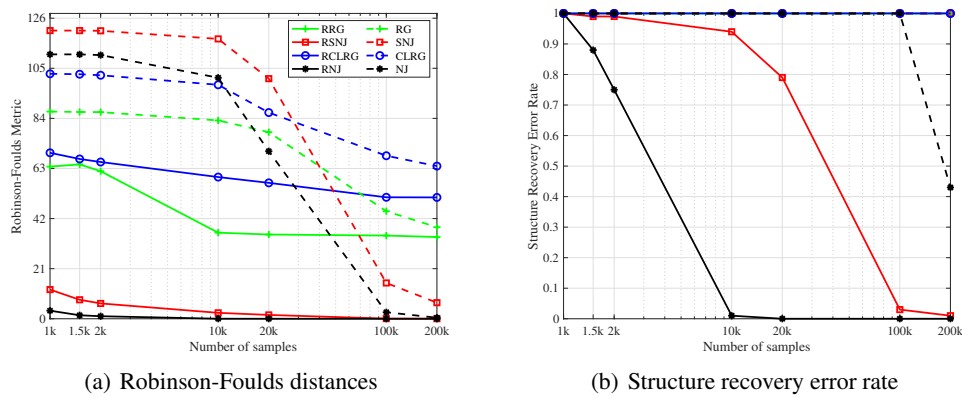

(a) Robinson-Foulds distances

(b) Structure recovery error rate

Figure 25: Performances of robustified and original learning algorithms with double binary corruptions

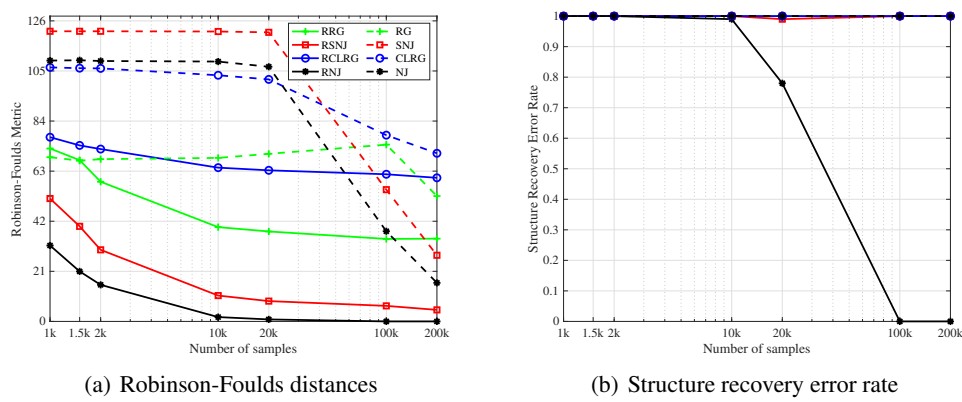

(a) Robinson-Foulds distances

(b) Structure recovery error rate

Figure 26: Performances of robustified and original learning algorithms with Gaussian outliers

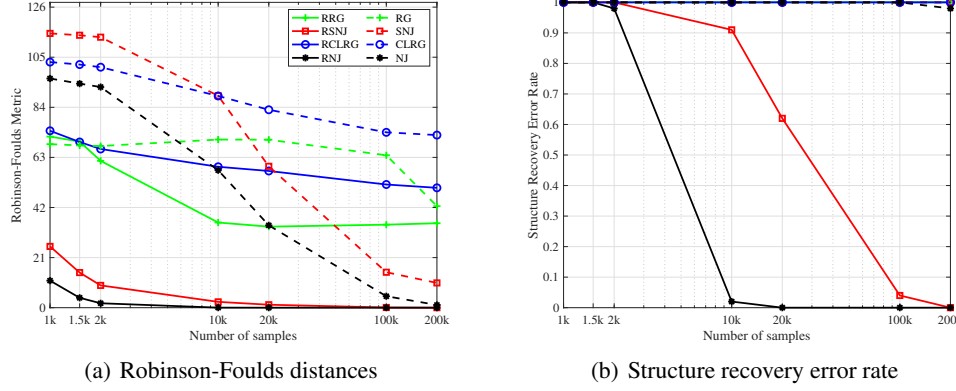

(a) Robinson-Foulds distances      (b) Structure recovery error rate

Figure 27: Performances of robustified and original learning algorithms with HMM outliers

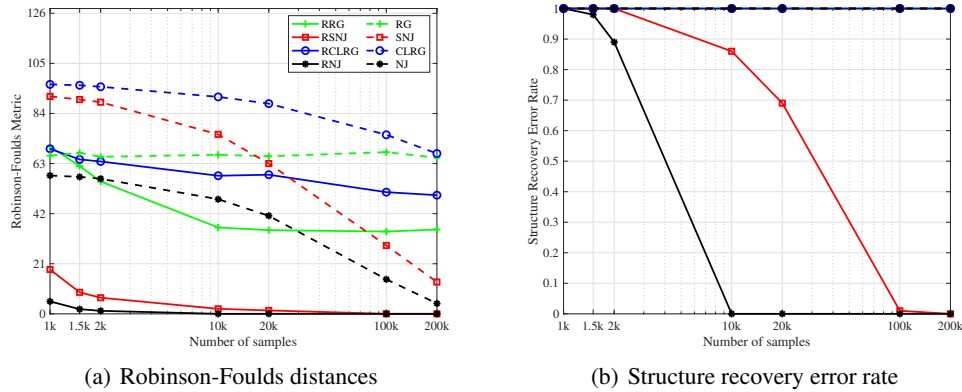

(a) Robinson-Foulds distances      (b) Structure recovery error rate

Figure 28: Performances of robustified and original learning algorithms with double binary outliers

These figures reinforce that the robustification procedure is highly effective in combating the corruptions. Furtheremore, we observe that RNJ performs the best among all these algorithms for the double binary tree. However, the simulation results in Jaffe et al. [6] shows that SNJ performs better than NJ. This does not contradict our observations here. The reason lies on the choice of the parameters of the model $\rho_{\min}$ and $\rho_{\max}$. In the simulations of [6], the parameter $\delta$ (defined in therein) is set to $0.9$, but in our simulation, the equivalent parameter $e^{-2\rho_{\max}/\mathrm{Diam}(\mathbb{T})}$ is $0.1$. The exponential dependence on $\rho_{\max}$ of RSNJ listed in Table 1 explains the difference between simulation results in [6] and our simulation results.

## M Proofs of results in Section 4

To derive the impossibility results, we will apply Fano's inequality on two special families of graphical models, each contained in $\mathcal{T}(|\mathcal{V}_{\mathrm{obs}}|, \rho_{\max}, l_{\max})$. Each graphical model in the families is parameterized by a quartet $(\mathbf{A}, \mathbf{\Sigma}_{\mathrm{r}}, \mathbf{\Sigma}_{\mathrm{n}}, \alpha)$. This quartet defines the Gaussian graphical model as follows. We choose a node in the tree as the root node $\mathbf{x}_{\mathrm{r}}$, and define the parent node and set of children nodes (in the rooted tree) of any node $x_i$ as $\mathrm{pa}(i)$ and $\mathcal{C}(x_i)$ respectively. The depth of a node $x_i$ (with respect to the root node $x_{\mathrm{r}}$) is $\mathrm{d}_{\mathbb{T}}(x_i, x_{\mathrm{r}})$. We specify the model in which

$$\mathbf{x}_i = \mathbf{A}\mathbf{x}_{\mathrm{pa}(i)} + \mathbf{n}_i \quad \text{for all} \quad x_i \in \mathcal{V} \tag{M.1}$$

where $\mathbf{A} \in \mathbb{R}^{l_{\max} \times l_{\max}}$ is non-singular, $\mathbf{n}_i \sim \mathcal{N}(\mathbf{0}, \alpha^{\mathrm{d}_{\mathbb{T}}(x_i, x_{\mathrm{r}})-1}\mathbf{\Sigma}_{\mathrm{n}})$ and $\mathbf{n}_i$'s are mutually independent. Since the root node has no parent, it is natural to set $\mathbf{x}_{\mathrm{pa}(\mathrm{r})} = \mathbf{0}$ and $\mathbf{n}_{\mathrm{r}} \sim \mathcal{N}(\mathbf{0}, \mathbf{\Sigma}_{\mathrm{r}})$. It is easy

to verify that the model specified by (M.1) and this initial condition is an undirected GGM. Then the covariance matrix of the random vector $\mathbf{x}_i$ is $\alpha^{d_T(x_i,x_r)}\boldsymbol{\Sigma}_r$.

**Proposition 14.** *If $\mathbf{n}_i$'s for the variables at depth $l$ are distributed as $\mathcal{N}(0, \alpha^{l-1}\boldsymbol{\Sigma}_n)$, and*

$$\mathbf{A}\boldsymbol{\Sigma}_r\mathbf{A}^\top + \boldsymbol{\Sigma}_n = \alpha\boldsymbol{\Sigma}_r \tag{M.2}$$

*where $\alpha > 0$ is a constant, then the covariance matrix of the variable at depth $l$ is $\alpha^l\boldsymbol{\Sigma}_r$.*

We term (M.2) as the $(\mathbf{A}, \boldsymbol{\Sigma}_r, \boldsymbol{\Sigma}_n)$-*homogenous* condition, which guarantees that covariance matrices of the random vectors in the tree are same up to a scale factor.

*Proof of Proposition 14.* The statement in Proposition 14 is equivalent to

$$\mathbf{A}^l\boldsymbol{\Sigma}_r(\mathbf{A}^l)^\top + \sum_{i=1}^{l}\alpha^{i-1}\mathbf{A}^{l-i}\boldsymbol{\Sigma}_n(\mathbf{A}^{l-i})^\top = \alpha^l\boldsymbol{\Sigma}_r. \tag{M.3}$$

We prove (M.3) by induction.

When $l = 1$, the homogenous condition guarantees that $\mathbf{A}\boldsymbol{\Sigma}_r\mathbf{A}^\top + \boldsymbol{\Sigma}_n = \alpha\boldsymbol{\Sigma}_r$.

If (M.3) holds for $l = 1, \ldots, n$, then for $l = n + 1$

$$\mathbf{A}^{n+1}\boldsymbol{\Sigma}_r(\mathbf{A}^{n+1})^\top + \sum_{i=1}^{n+1}\alpha^{i-1}\mathbf{A}^{n+1-i}\boldsymbol{\Sigma}_n(\mathbf{A}^{n+1-i})^\top$$

$$= \mathbf{A}(\mathbf{A}^n\boldsymbol{\Sigma}_r(\mathbf{A}^n)^\top + \sum_{i=1}^{n+1}\alpha^{i-1}\mathbf{A}^{n-i}\boldsymbol{\Sigma}_n(\mathbf{A}^{n-i})^\top)\mathbf{A}^\top \tag{M.4}$$

$$= \mathbf{A}(\alpha^n\boldsymbol{\Sigma}_r + \alpha^n\mathbf{A}^{-1}\boldsymbol{\Sigma}_n\mathbf{A}^{-\top})\mathbf{A}^\top \tag{M.5}$$

$$= \alpha^n(\mathbf{A}\boldsymbol{\Sigma}_r\mathbf{A}^\top + \boldsymbol{\Sigma}_n) \tag{M.6}$$

$$= \alpha^{n+1}\boldsymbol{\Sigma}_r \tag{M.7}$$

as desired. $\qquad\square$

**Proposition 15.** *The undirected graphical model specified by (M.1) and the initial condition $\mathbf{x}_{pa(r)} = 0$, $\mathbf{n}_r \sim \mathcal{N}(\mathbf{0}, \boldsymbol{\Sigma}_r)$ is GGM.*

*Proof of Proposition 15.* To prove that the specified model is a GGM, we need to prove that the joint distribution of all variables is Gaussian and that the conditional independence relationship induced by the edges is achieved.

According to (M.1) and the initial condition, it is easy to see that any linear combination of variables is the linear combination of independent Gaussian variables, which is Gaussian. Thus, the joint distribution of all variables is indeed Gaussian.

To show that the conditional independence is guaranteed, we show that

$$A \perp\!\!\!\perp B \mid S \text{ for any } S \text{ separates } A \text{ and } B. \tag{M.8}$$

where $S$, $A$ and $B$ are all sets of nodes, and $S$ separates $A$ and $B$ means that any path connected nodes in $A$ and $B$ goes through a node in $S$.

Without loss of generality, we consider the case where $S$, $A$ and $B$ consist of a single node for conciseness of the proof. The case where these sets consist of multiple nodes can be easily proved by generalizing the proof we show here.

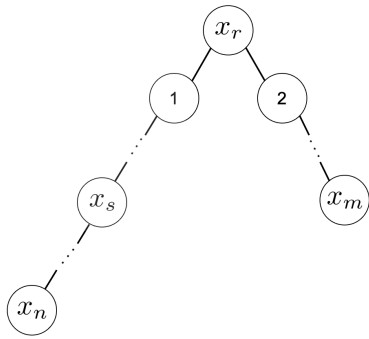

Figure 29: Illustration of the relationship among $x_n$, $x_m$ and $x_s$.

We first consider the case where $x_n$ and $x_m$ belong to different branches, as shown in Fig. 29, and the depths of $x_n$ and $x_m$ are $n$ and $m$, respectively. The separator node $x_s$ can be anywhere along the path connecting $x_n$ and $x_m$. Without loss of generality, we assume it sits in the same branch as $x_n$, and its depth is $s$, where $s < n$. Then we have

$$\mathbb{E}[\mathbf{x}_n \mathbf{x}_n^\top] = \mathbf{A}^n \boldsymbol{\Sigma}_{\mathrm{r}} (\mathbf{A}^n)^\top + \sum_{i=1}^n \mathbf{A}^{n-i} \boldsymbol{\Sigma}_i (\mathbf{A}^{n-i})^\top \tag{M.9}$$

$$\mathbb{E}[\mathbf{x}_m \mathbf{x}_m^\top] = \mathbf{A}^m \boldsymbol{\Sigma}_{\mathrm{r}} (\mathbf{A}^m)^\top + \sum_{i=1}^m \mathbf{A}^{m-i} \boldsymbol{\Sigma}_i' (\mathbf{A}^{m-i})^\top \tag{M.10}$$

$$\mathbb{E}[\mathbf{x}_t \mathbf{x}_n^\top] = \mathbf{A}^t \boldsymbol{\Sigma}_{\mathrm{r}} (\mathbf{A}^n)^\top + \sum_{i=1}^t \mathbf{A}^{t-i} \boldsymbol{\Sigma}_i (\mathbf{A}^{n-i})^\top, \tag{M.11}$$

where $\boldsymbol{\Sigma}_i$ and $\boldsymbol{\Sigma}_i'$ are the covariance matrices of the independent noises in each branch.

Then we calculate the distribution of conditional distribution

$$\begin{bmatrix} \mathbf{x}_n \\ \mathbf{x}_m \end{bmatrix} \mid \mathbf{x}_t \sim \mathcal{N}(\tilde{\mu}, \tilde{\boldsymbol{\Sigma}}), \tag{M.12}$$

where

$$\tilde{\boldsymbol{\Sigma}} = \begin{bmatrix} \tilde{\boldsymbol{\Sigma}}_{11} & \tilde{\boldsymbol{\Sigma}}_{12} \\ \tilde{\boldsymbol{\Sigma}}_{21} & \tilde{\boldsymbol{\Sigma}}_{22} \end{bmatrix}. \tag{M.13}$$

We have

$$\tilde{\boldsymbol{\Sigma}}_{12} = \mathbf{A}^n \boldsymbol{\Sigma}_{\mathrm{r}} (\mathbf{A}^m)^\top - \left( \mathbf{A}^n \boldsymbol{\Sigma}_{\mathrm{r}} (\mathbf{A}^t)^\top + \sum_{i=1}^t \mathbf{A}^{n-i} \boldsymbol{\Sigma}_i \mathbf{A}^{(t-i)\top} \right)$$

$$\times \left( \mathbf{A}^t \boldsymbol{\Sigma}_{\mathrm{r}} (\mathbf{A}^t)^\top + \sum_{i=1}^t \mathbf{A}^{t-i} \boldsymbol{\Sigma}_i \mathbf{A}^{(t-i)\top} \right)^{-1} \mathbf{A}^t \boldsymbol{\Sigma}_{\mathrm{r}} (\mathbf{A}^m)^\top = \mathbf{0}. \tag{M.14}$$

Thus, the conditional independence of $x_n$ and $x_m$ given $x_s$ is proved.

When $x_n$ and $x_m$ are on the same branch, a similar calculation can be performed to prove the conditional independence property. $\qquad \square$

**Proposition 16.** *For a tree graph $\mathbb{T} = (\mathcal{V}, \mathcal{E})$ where $\mathcal{V} = \{x_1, x_2, \ldots, x_p\}$ and any symmetric matrix $\mathbf{A} \in \mathbb{R}^{d \times d}$ whose absolute values of all the eigenvalues are less than 1, the determinant of the matrix $\bar{\mathbf{D}}(\mathbb{T}, \mathbf{A})$, which is defined below, is $\left[ \det(\mathbf{I} - \mathbf{A}^2) \right]^{p-1}$*

$$\bar{\mathbf{D}}(\mathbb{T}, \mathbf{A}) = \begin{bmatrix} \mathbf{A}^{d_{\mathbb{T}}(x_1, x_1)} & \mathbf{A}^{d_{\mathbb{T}}(x_1, x_2)} & \cdots & \mathbf{A}^{d_{\mathbb{T}}(x_1, x_p)} \\ \mathbf{A}^{d_{\mathbb{T}}(x_2, x_1)} & \mathbf{A}^{d_{\mathbb{T}}(x_2, x_2)} & \cdots & \mathbf{A}^{d_{\mathbb{T}}(x_2, x_p)} \\ \vdots & \vdots & \ddots & \vdots \\ \mathbf{A}^{d_{\mathbb{T}}(x_p, x_1)} & \mathbf{A}^{d_{\mathbb{T}}(x_p, x_2)} & \cdots & \mathbf{A}^{d_{\mathbb{T}}(x_p, x_p)} \end{bmatrix}, \tag{M.15}$$

*Proof of Proposition 16.* Since the underlying structure is a tree, we can always find a leaf and its neighbor. Without loss of generality, we assume $x_p$ is a leaf and $x_{p-1}$ is $x_p$'s neighbor, otherwise we can exchange the rows and columns of $\bar{\mathbf{D}}(\mathbb{T}, \mathbf{A})$ to satisfy this assumption. Then we have

$$\mathrm{d}_{\mathbb{T}}(x_{p-1}, x_p) = 1 \quad \text{and} \quad \mathrm{d}_{\mathbb{T}}(x_p, x_i) = \mathrm{d}_{\mathbb{T}}(x_{p-1}, x_i) + 1 \quad \text{for all} \quad i \in [p-2]. \qquad \text{(M.16)}$$

Thus, we have

$$\bar{\mathbf{D}}(\mathbb{T}, \mathbf{A}) = \begin{bmatrix} \mathbf{A}^0 & \mathbf{A}^{\mathrm{d}_{\mathbb{T}}(x_1,x_2)} & \cdots & \mathbf{A}^{\mathrm{d}_{\mathbb{T}}(x_1,x_{p-1})} & \mathbf{A}^{\mathrm{d}_{\mathbb{T}}(x_1,x_{p-1})+1} \\ \mathbf{A}^{\mathrm{d}_{\mathbb{T}}(x_2,x_1)} & \mathbf{A}^0 & \cdots & \mathbf{A}^{\mathrm{d}_{\mathbb{T}}(x_2,x_{p-1})} & \mathbf{A}^{\mathrm{d}_{\mathbb{T}}(x_2,x_{p-1})+1} \\ \vdots & \vdots & \ddots & \vdots & \vdots \\ \mathbf{A}^{\mathrm{d}_{\mathbb{T}}(x_{p-1},x_1)} & \mathbf{A}^{\mathrm{d}_{\mathbb{T}}(x_{p-1},x_2)} & \cdots & \mathbf{A}^0 & \mathbf{A}^1 \\ \mathbf{A}^{\mathrm{d}_{\mathbb{T}}(x_{p-1},x_1)+1} & \mathbf{A}^{\mathrm{d}_{\mathbb{T}}(x_{p-1},x_2)+1} & \cdots & \mathbf{A}^1 & \mathbf{A}^0 \end{bmatrix}. \qquad \text{(M.17)}$$

Subtracting $\mathbf{A}$ times the penultimate row of $\bar{\mathbf{D}}(\mathbb{T}, \mathbf{A})$ from the last row of $\bar{\mathbf{D}}(\mathbb{T}, \mathbf{A})$, we have

$$\begin{bmatrix} \mathbf{A}^0 & \mathbf{A}^{\mathrm{d}_{\mathbb{T}}(x_1,x_2)} & \cdots & \mathbf{A}^{\mathrm{d}_{\mathbb{T}}(x_1,x_{p-1})} & \mathbf{A}^{\mathrm{d}_{\mathbb{T}}(x_1,x_{p-1})+1} \\ \mathbf{A}^{\mathrm{d}_{\mathbb{T}}(x_2,x_1)} & \mathbf{A}^0 & \cdots & \mathbf{A}^{\mathrm{d}_{\mathbb{T}}(x_2,x_{p-1})} & \mathbf{A}^{\mathrm{d}_{\mathbb{T}}(x_2,x_{p-1})+1} \\ \vdots & \vdots & \ddots & \vdots & \vdots \\ \mathbf{A}^{\mathrm{d}_{\mathbb{T}}(x_{p-1},x_1)} & \mathbf{A}^{\mathrm{d}_{\mathbb{T}}(x_{p-1},x_2)} & \cdots & \mathbf{A}^0 & \mathbf{A}^1 \\ \mathbf{0} & \mathbf{0} & \cdots & \mathbf{0} & \mathbf{A}^0 - \mathbf{A}^2 \end{bmatrix}. \qquad \text{(M.18)}$$

Applying the similar column transformation, we have

$$\begin{bmatrix} \mathbf{A}^0 & \mathbf{A}^{\mathrm{d}_{\mathbb{T}}(x_1,x_2)} & \cdots & \mathbf{A}^{\mathrm{d}_{\mathbb{T}}(x_1,x_{p-1})} & \mathbf{0} \\ \mathbf{A}^{\mathrm{d}_{\mathbb{T}}(x_2,x_1)} & \mathbf{A}^0 & \cdots & \mathbf{A}^{\mathrm{d}_{\mathbb{T}}(x_2,x_{p-1})} & \mathbf{0} \\ \vdots & \vdots & \ddots & \vdots & \vdots \\ \mathbf{A}^{\mathrm{d}_{\mathbb{T}}(x_{p-1},x_1)} & \mathbf{A}^{\mathrm{d}_{\mathbb{T}}(x_{p-1},x_2)} & \cdots & \mathbf{A}^0 & \mathbf{0} \\ \mathbf{0} & \mathbf{0} & \cdots & \mathbf{0} & \mathbf{A}^0 - \mathbf{A}^2 \end{bmatrix}. \qquad \text{(M.19)}$$

By repeating these row and column transformations, we will acquire

$$\mathrm{diag}(\mathbf{I}, \mathbf{I} - \mathbf{A}^2, \ldots, \mathbf{I} - \mathbf{A}^2), \qquad \text{(M.20)}$$

which has the same determinant as $\bar{\mathbf{D}}(\mathbb{T}, \mathbf{A})$. Thus, $\det(\bar{\mathbf{D}}(\mathbb{T}, \mathbf{A})) = \left[ \det(\mathbf{I} - \mathbf{A}^2) \right]^{p-1}$. $\qquad \square$

The proof of Theorem 5 follows from the following non-asymptotic result.

**Theorem 10.** *Consider the class of graphs* $\mathcal{T}(|\mathcal{V}_{\mathrm{obs}}|, \rho_{\max}, l_{\max})$*, where* $|\mathcal{V}_{\mathrm{obs}}| \geq 3$*. If the number of i.i.d. samples* $n$ *is upper bounded as follows,*

$$n < \max \left\{ \frac{2(1-\delta)\left(\log 3^{1/3} \lfloor \log_3(|\mathcal{V}_{\mathrm{obs}}|) \rfloor - 1\right) - \frac{2}{|\mathcal{V}_{\mathrm{obs}}|}}{-l_{\max} \log\left(1 - e^{-\frac{\rho_{\max}}{\lceil \log_3(|\mathcal{V}_{\mathrm{obs}}|) \rceil l_{\max}}}\right)}, \frac{(1-\delta)/5 - \frac{2}{|\mathcal{V}_{\mathrm{obs}}|}}{-l_{\max} \log\left(1 - e^{-\frac{2\rho_{\max}}{3 l_{\max}}}\right)} \right\} \qquad \text{(M.21)}$$

*then for any graph decoder* $\phi : \mathbb{R}^{n|\mathcal{V}_{\mathrm{obs}}| l_{\max}} \to \mathcal{T}(|\mathcal{V}_{\mathrm{obs}}|, \rho_{\max}, l_{\max})$

$$\max_{\theta(\mathbb{T}) \in \mathcal{T}(|\mathcal{V}_{\mathrm{obs}}|, \rho_{\max}, l_{\max})} \mathbb{P}_{\theta(\mathbb{T})}(\phi(\mathbf{X}_1^n) \neq \mathbb{T}) \geq \delta. \qquad \text{(M.22)}$$

*Proof of Theorem 5.* To prove Theorem 5, we simply implement the Taylor expansion $\log(1+x) = \sum_{k=1}^{\infty} (-1)^{k+1} \frac{x^k}{k}$ on (M.21) in Theorem 10 taking $\rho_{\max} \to \infty$ and $|\mathcal{V}_{\mathrm{obs}}| \to \infty$. $\qquad \square$

It remains to prove Theorem 10.

*Proof of Theorem 10.* To prove this non-asymptotic converse bound, we consider $M$ models in $\mathcal{T}(|\mathcal{V}_{\mathrm{obs}}|, \rho_{\max}, l_{\max})$, whose parameters are enumerated as $\{\theta^{(1)}, \theta^{(2)}, \ldots, \theta^{(M)}\}$. We choose a model $K = k$ uniformly in $\{1, \ldots, M\}$ and generate $n$ i.i.d. samples $\mathbf{X}_1^n$ from $\mathbb{P}_{\theta^{(k)}}$. A latent tree learning algorithm is a decoder $\phi : \mathbb{R}^{n|\mathcal{V}_{\mathrm{obs}}| l_{\max}} \to \{1, \ldots, M\}$.

Two families are built to derive the converse bound. We separately describe the families of $M$ graphical models we consider here.

**Graphical model family A** We specify the structure of trees as full-$m$ trees, except the top layer, as shown in Fig. 30. All the observed nodes are leaves. The parameters of each tree are set to satisfy the conditions in Proposition 14. Additionally, we set $\alpha = 1$ in the homogeneous condition (M.2) and set $\mathbf{A}$ to be a symmetric matrix that commutes with $\mathbf{\Sigma}_r$. We set $m = 3$ and $L = \lfloor \log_3(|\mathcal{V}_{\text{obs}}|) \rfloor$, then the number of residual nodes is $r = |\mathcal{V}_{\text{obs}}| - 3^L$. All these residual nodes are connected to one of parents of the observed nodes.

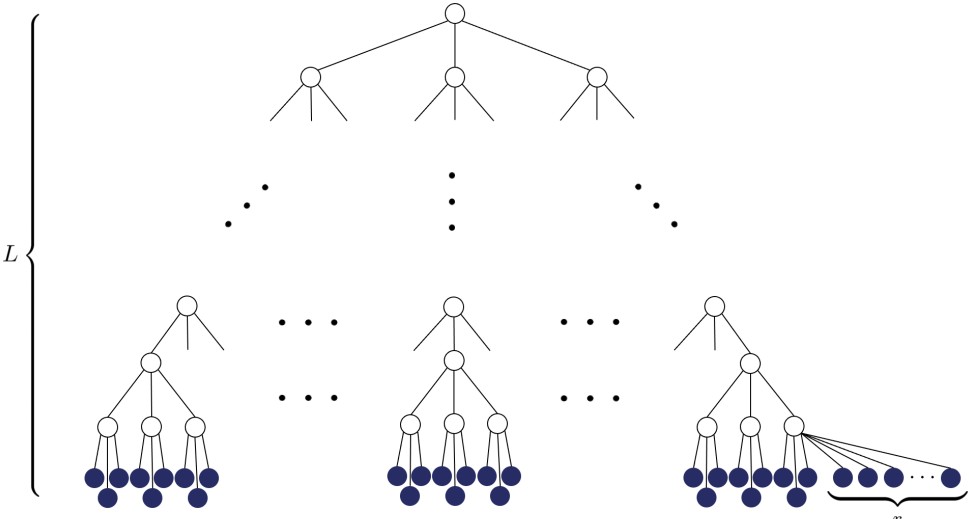

(a) The full 3-tree. All the observed nodes are leaves, and residual nodes are connected to one of parents of the observed nodes.

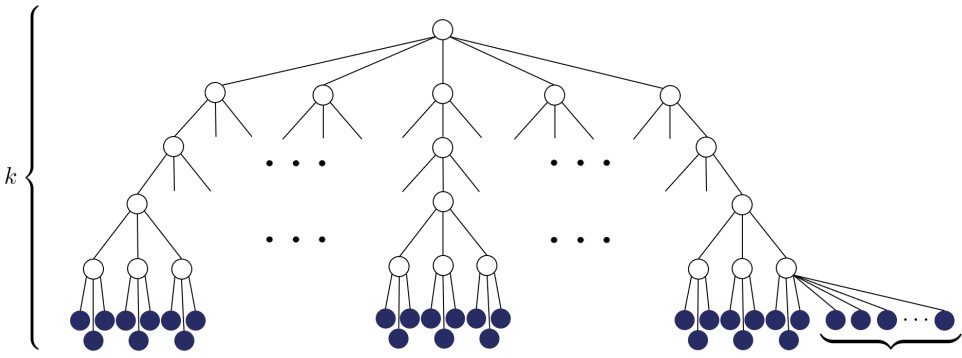

(b) The full tree with depth $k$, where all the internal nodes have three children except the root node.

Figure 30: The family A of graphical models considered in the impossibility result.

To derive the converse result, we use the Fano's method. Namely, Fano's method says that if the sample size

$$n < \frac{(1-\delta)\log M}{I(\mathbf{X}_1; K)},$$

(M.23)

then for any decoder

$$\max_{k=1,\dots,M} \mathbb{P}_{\theta^{(k)}} \left[ \phi(\mathbf{X}_1^n) \neq k \right] \geq \delta - \frac{1}{\log M}.$$

(M.24)

We first evaluate the cardinality of this family of graphical models. We first count the number of graphical models with depth $1 \leq k \leq L$ in Fig. 30. For a specific order of labels (e.g., $1, 2, \dots, m^L$), exchanging the labels in a family does not change the topology of the tree. For instance, exchanging the position of node 1 and node $m$, we obtain an identical tree. By changing the orders in the last

layer, it is obvious that there are $(m!)^{m^{L-1}}$ different orders representing the same structure. For the penultimate layer, there are $(m!)^{m^{L-2}}$ different orders represent an identical structure. Thus, for a specific graphical model with depth $k$, there are

$$(m^{L-k+1})! \prod_{i=L-1}^{L-k+1} (m!)^{m^i} = (m^{L-k+1})!(m!)^{\frac{m^L-m^{L-k+1}}{m-1}} \qquad \text{(M.25)}$$

graphical models with the same distribution.

Then the number of different structures of graphical models with depth $k$ can be calculated as

$$\frac{(m^L)!}{(m^{L-k+1})!(m!)^{\frac{m^L-m^{L-k+1}}{m-1}}}. \qquad \text{(M.26)}$$

The total number of different graphical models in the family we consider is

$$M = \sum_{k=1}^{L} \frac{(m^L)!}{(m^{L-k+1})!(m!)^{\frac{m^L-m^{L-k+1}}{m-1}}}. \qquad \text{(M.27)}$$

Using Stirling's formula, we have the following simplification of $M$:

$$M \geq \sum_{k=1}^{L} \frac{\sqrt{2\pi}(m^L)^{m^L+1/2}e^{-m^L}}{e(m^{L-k+1})^{m^{L-k+1}+1/2}e^{-m^{L-k+1}}} \frac{1}{(e^{-(m-1)}m^{m+1/2})^{(m^L-m^{L-k+1})/(m-1)}} \qquad \text{(M.28)}$$

$$= \sum_{k=1}^{L} \frac{\sqrt{2\pi}}{e} m^{Lm^L-(L-k+1)m^{L-k+1}+(k-1)/2-(m+1/2)(m^L-m^{L-k+1})/(m-1)} \qquad \text{(M.29)}$$

$$= \frac{\sqrt{2\pi}}{e} m^{(L-(m+1/2)/(m-1))m^L} \sum_{k=1}^{L} m^{-(L-k+1-(m+1/2)/(m-1))m^{L-k+1}+(k-1)/2} \qquad \text{(M.30)}$$

$$> \frac{\sqrt{2\pi}}{e} m^{(L-(m+1/2)/(m-1))m^L} m^{3m/(2m-2)+(L-1)/2} \qquad \text{(M.31)}$$

and

$$\log M > \log\left(\frac{\sqrt{2\pi}}{e}\right) + m^L\left(L - \frac{m+\frac{1}{2}}{m-1}\right)\log m + \left(\frac{3m}{2m-2} + \frac{L-1}{2}\right)\log m \qquad \text{(M.32)}$$

Thus,

$$\frac{\log M}{|\mathcal{V}_{\text{obs}}|} > \frac{1}{|\mathcal{V}_{\text{obs}}|}\log\left(\frac{\sqrt{2\pi}}{e}\right) + \frac{m^L}{|\mathcal{V}_{\text{obs}}|}\left(L - \frac{m+\frac{1}{2}}{m-1}\right)\log m + \frac{1}{|\mathcal{V}_{\text{obs}}|}\left(\frac{3m}{2m-2} + \frac{L-1}{2}\right)\log m \qquad \text{(M.33)}$$

$$> \frac{\log m}{m}\left(L - \frac{m+\frac{1}{2}}{m-1}\right) + \frac{1}{|\mathcal{V}_{\text{obs}}|}\log\left(\frac{\sqrt{2\pi}}{e}\right) \qquad \text{(M.34)}$$

$$\overset{(a)}{>} \frac{\log 3}{3}L - 1, \qquad \text{(M.35)}$$

where inequality $(a)$ is derived by substituting $m = 3$.

Next we calculate an upper bound of $I(\mathbf{X}_1; K)$. Since $\mathbb{P}_{\mathbb{T}_k} = \mathcal{N}(0, \boldsymbol{\Sigma}_{\text{obs}}(\mathbb{T}_k))$, where $\boldsymbol{\Sigma}_{\text{obs}}$ is the covariance matrix of observed variables, we have [26]

$$I(\mathbf{X}_1; K) \leq \mathbb{E}_{\mathbb{T}_k}\left[D(\mathbb{P}_{\mathbb{T}_k}\|\mathbb{Q})\right], \qquad \text{(M.36)}$$

for any distribution $\mathbb{Q}$. By choosing $\mathbb{Q} = \mathcal{N}(0, \mathbf{I}_{l_{\max}|\mathcal{V}_{\text{obs}}|\times l_{\max}|\mathcal{V}_{\text{obs}}|})$, we have

$$D(\mathbb{P}_{\mathbb{T}_k}\|\mathbb{Q}) = \frac{1}{2}\left\{\log\left(\det\left(\boldsymbol{\Theta}_{\text{obs}}(\mathbb{T}_k)\right)\right) + \text{trace}\left(\boldsymbol{\Sigma}_{\text{obs}}(\mathbb{T}_k)\right) - l_{\max}|\mathcal{V}_{\text{obs}}|\right\} \qquad \text{(M.37)}$$

$$= \frac{1}{2}\left\{-\log\left(\det\left(\boldsymbol{\Sigma}_{\text{obs}}(\mathbb{T}_k)\right)\right) + \text{trace}\left(\boldsymbol{\Sigma}_{\text{obs}}(\mathbb{T}_k)\right) - l_{\max}|\mathcal{V}_{\text{obs}}|\right\} \qquad \text{(M.38)}$$

Since we consider models that satisfy the conditions in Proposition 14, the covariance matrix of any two variables is

$$\mathbb{E}\left[\mathbf{x}_i \mathbf{x}_j^\top\right] = \mathbf{\Sigma}_{\mathrm{r}} \mathbf{A}^{d_{\mathbb{T}}(x_i, x_j)}. \tag{M.39}$$

The covariance matrix $\mathbf{\Sigma}(\mathbb{T}_k)$ for all the observed variables and latent variables $\mathcal{V}_{\mathrm{obs}} \cup \mathcal{V}_{\mathrm{hid}}$ is

$$\begin{bmatrix} \mathbf{\Sigma}_{\mathrm{r}}\mathbf{A}^{d_{\mathbb{T}}(x_1,x_1)} & \cdots & \mathbf{\Sigma}_{\mathrm{r}}\mathbf{A}^{d_{\mathbb{T}}(x_1,x_{|\mathcal{V}_{\mathrm{obs}}|})} & \mathbf{\Sigma}_{\mathrm{r}}\mathbf{A}^{d_{\mathbb{T}}(x_1,y_1)} & \cdots & \mathbf{\Sigma}_{\mathrm{r}}\mathbf{A}^{d_{\mathbb{T}}(x_1,y_{|\mathcal{V}_{\mathrm{hid}}|})} \\ \vdots & \ddots & \vdots & \vdots & \ddots & \vdots \\ \mathbf{\Sigma}_{\mathrm{r}}\mathbf{A}^{d_{\mathbb{T}}(x_{|\mathcal{V}_{\mathrm{obs}}|},x_1)} & \cdots & \mathbf{\Sigma}_{\mathrm{r}}\mathbf{A}^{d_{\mathbb{T}}(x_{|\mathcal{V}_{\mathrm{obs}}|},x_{|\mathcal{V}_{\mathrm{obs}}|})} & \mathbf{\Sigma}_{\mathrm{r}}\mathbf{A}^{d_{\mathbb{T}}(x_{|\mathcal{V}_{\mathrm{obs}}|},y_1)} & \cdots & \mathbf{\Sigma}_{\mathrm{r}}\mathbf{A}^{d_{\mathbb{T}}(x_{|\mathcal{V}_{\mathrm{obs}}|},x_{|\mathcal{V}_{\mathrm{hid}}|})} \\ \mathbf{\Sigma}_{\mathrm{r}}\mathbf{A}^{d_{\mathbb{T}}(y_1,x_1)} & \cdots & \mathbf{\Sigma}_{\mathrm{r}}\mathbf{A}^{d_{\mathbb{T}}(y_1,x_{|\mathcal{V}_{\mathrm{obs}}|})} & \mathbf{\Sigma}_{\mathrm{r}}\mathbf{A}^{d_{\mathbb{T}}(y_1,y_1)} & \cdots & \mathbf{\Sigma}_{\mathrm{r}}\mathbf{A}^{d_{\mathbb{T}}(y_1,y_{|\mathcal{V}_{\mathrm{hid}}|})} \\ \vdots & \ddots & \vdots & \vdots & \ddots & \vdots \\ \mathbf{\Sigma}_{\mathrm{r}}\mathbf{A}^{d_{\mathbb{T}}(y_{|\mathcal{V}_{\mathrm{hid}}|},x_1)} & \cdots & \mathbf{\Sigma}_{\mathrm{r}}\mathbf{A}^{d_{\mathbb{T}}(y_{|\mathcal{V}_{\mathrm{hid}}|},x_{|\mathcal{V}_{\mathrm{obs}}|})} & \mathbf{\Sigma}_{\mathrm{r}}\mathbf{A}^{d_{\mathbb{T}}(y_{|\mathcal{V}_{\mathrm{hid}}|},y_1)} & \cdots & \mathbf{\Sigma}_{\mathrm{r}}\mathbf{A}^{d_{\mathbb{T}}(y_{|\mathcal{V}_{\mathrm{hid}}|},y_{|\mathcal{V}_{\mathrm{hid}}|})} \end{bmatrix}$$

$$= \left(\mathbf{I}_{|\mathcal{V}|\times|\mathcal{V}|} \otimes \mathbf{\Sigma}_{\mathrm{r}}\right) \begin{bmatrix} \mathbf{V} & \mathbf{B} \\ \mathbf{B}^\top & \mathbf{H} \end{bmatrix} \tag{M.40}$$

$$= \left(\mathbf{I}_{|\mathcal{V}|\times|\mathcal{V}|} \otimes \mathbf{\Sigma}_{\mathrm{r}}\right) \bar{\mathbf{D}}(\mathbb{T}_k, \mathbf{A}) \tag{M.41}$$

where $\mathbf{A} \otimes \mathbf{B}$ is the Kronecker product of matrices $\mathbf{A}$ and $\mathbf{B}$. Letting $\mathbf{\Sigma}_{\mathrm{r}} = \mathbf{I}$, it is obvious that $\bar{\mathbf{D}}(\mathbb{T}, \mathbf{A})$ is a positive definite matrix. Furthermore, $\mathbf{H} - \mathbf{B}^\top \mathbf{V}^{-1} \mathbf{B}$ is positive semi-definite matrix, since it is the inverse of the principal minor of $\bar{\mathbf{D}}^{-1}$. Thus we have

$$\det\left(\bar{\mathbf{D}}(\mathbb{T}_k, \mathbf{A})\right) = \det(\mathbf{V})\det\left(\mathbf{H} - \mathbf{B}^\top \mathbf{V}^{-1}\mathbf{B}\right) \tag{M.42}$$

$$\overset{(a)}{\leq} \det(\mathbf{V})\det\left(\mathbf{H}\right) \overset{(b)}{=} \det(\mathbf{V})\left[\det(\mathbf{I} - \mathbf{A}^2)\right]^{|\mathcal{V}_{\mathrm{hid}}|-1}, \tag{M.43}$$

where inequality $(a)$ is derived from Minkowski determinant theorem [27], and $(b)$ comes from the fact that all the latent variables themselves form a tree. Also, we have that

$$\det\left(\bar{\mathbf{D}}(\mathbb{T}_k, \mathbf{A})\right) = \left[\det(\mathbf{I} - \mathbf{A}^2)\right]^{|\mathcal{V}_{\mathrm{hid}}|+|\mathcal{V}_{\mathrm{obs}}|-1}. \tag{M.44}$$

Thus, we have

$$\det(\mathbf{V}) \geq \left[\det(\mathbf{I} - \mathbf{A}^2)\right]^{|\mathcal{V}_{\mathrm{obs}}|}, \tag{M.45}$$

which implies that

$$\log\left(\det\left(\mathbf{\Sigma}_{\mathrm{obs}}(\mathbb{T}_k)\right)\right) \geq \log\left(\left(\det(\mathbf{\Sigma}_{\mathrm{r}})\right)^{|\mathcal{V}_{\mathrm{obs}}|}\left[\det(\mathbf{I} - \mathbf{A}^2)\right]^{|\mathcal{V}_{\mathrm{obs}}|}\right) \tag{M.46}$$

$$= |\mathcal{V}_{\mathrm{obs}}|\log\left(\det(\mathbf{\Sigma}_{\mathrm{r}})\det(\mathbf{I} - \mathbf{A}^2)\right). \tag{M.47}$$

The mutual information can thus be upper bounded as

$$I(\mathbf{X}_1; K) \leq \frac{1}{2}|\mathcal{V}_{\mathrm{obs}}|\left(-\log\left(\det(\mathbf{I} - \mathbf{A}^2)\right) + \mathrm{trace}(\mathbf{\Sigma}_{\mathrm{r}}) - \log\left(\det(\mathbf{\Sigma}_{\mathrm{r}})\right) - l_{\max}\right) \tag{M.48}$$

Combining inequalities (M.28) and (M.48), we can deduce that the any decoder will construct the wrong tree with probability at least $\delta$ if

$$n < \frac{2(1-\delta)\left(\log 3^{1/3}L - 1\right)}{-\log\left(\det(\mathbf{I} - \mathbf{A}^2)\right) + \mathrm{trace}(\mathbf{\Sigma}_{\mathrm{r}}) - \log\left(\det(\mathbf{\Sigma}_{\mathrm{r}})\right) - l_{\max}} \tag{M.49}$$

By choosing $\mathbf{\Sigma}_{\mathrm{r}} = \mathbf{I}$ and letting the eigenvalues of $\mathbf{A}$ are all the same, we have

$$\rho_{\max} = -\frac{2L}{2}\log\left(\det(\mathbf{A}^2)\right) = -2l_{\max}L\log\left(\lambda(\mathbf{A})\right) \tag{M.50}$$

and

$$\mathrm{trace}(\mathbf{\Sigma}_{\mathrm{r}}) - \log\left(\det(\mathbf{\Sigma}_{\mathrm{r}})\right) - l_{\max} = 0. \tag{M.51}$$

Furthermore, we have

$$\log\left(\det(\mathbf{I} - \mathbf{A}^2)\right) = l_{\max}\log\left(1 - \lambda(\mathbf{A})^2\right) = l_{\max}\log\left(1 - e^{-\frac{\rho_{\max}}{L l_{\max}}}\right). \tag{M.52}$$

By choosing $\delta' = \delta + \frac{1}{\log(M)}$, we have that the condition

$$n < \frac{2(1-\delta')\left(\log 3^{1/3}L - 1\right) - \frac{2}{|\mathcal{V}_{\mathrm{obs}}|}}{-l_{\max}\log\left(1 - e^{-\frac{\rho_{\max}}{L l_{\max}}}\right)} \tag{M.53}$$

guarantees that

$$\max_{\theta(\mathbb{T})\in\mathcal{T}(|\mathcal{V}_{\mathrm{obs}}|,\rho_{\max},l_{\max})} \mathbb{P}_{\theta(\mathbb{T})}(\phi(\mathbf{X}_1^n) \neq \mathbb{T}) \geq \delta' \tag{M.54}$$

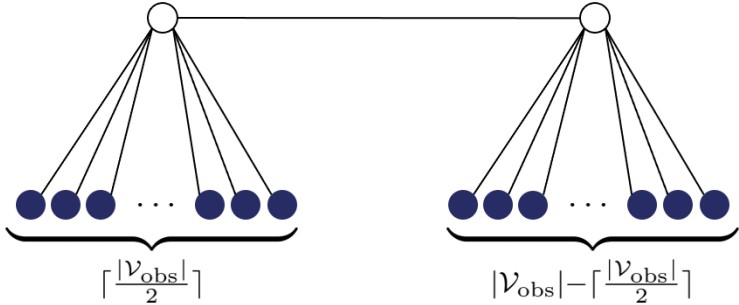

Figure 31: The family B of graphical models considered in the impossibility result.

**Graphical model family B** We consider the family of graphical models with double-star substructures, as shown in Fig. 31. Then the number of graphical models $M$ in this family is lower bounded as

$$
M > \frac{1}{2}\binom{|\mathcal{V}_{\mathrm{obs}}|}{\lceil|\mathcal{V}_{\mathrm{obs}}|/2\rceil} > \frac{\sqrt{2\pi}|\mathcal{V}_{\mathrm{obs}}|^{|\mathcal{V}_{\mathrm{obs}}|+1/2}e^{-|\mathcal{V}_{\mathrm{obs}}|}}{2\big(en^{n+1/2}e^{-n}\big)\big(e(|\mathcal{V}_{\mathrm{obs}}|-n)^{|\mathcal{V}_{\mathrm{obs}}|-n+1/2}e^{-(|\mathcal{V}_{\mathrm{obs}}|-n)}\big)}
$$
$$
= \frac{\sqrt{2\pi}}{2e^2}\frac{|\mathcal{V}_{\mathrm{obs}}|^{|\mathcal{V}_{\mathrm{obs}}|+1/2}}{n^{n+1/2}(|\mathcal{V}_{\mathrm{obs}}|-n)^{|\mathcal{V}_{\mathrm{obs}}|-n+1/2}}
\tag{M.55}
$$

when $n = \lceil|\mathcal{V}_{\mathrm{obs}}|/2\rceil$. Since $n \geq |\mathcal{V}_{\mathrm{obs}}| - n$, we further have

$$
M > \frac{\sqrt{2\pi}}{2e^2}\frac{|\mathcal{V}_{\mathrm{obs}}|^{|\mathcal{V}_{\mathrm{obs}}|+1/2}}{n^{|\mathcal{V}_{\mathrm{obs}}|+1}} = \sqrt{\frac{\pi}{2e^4|\mathcal{V}_{\mathrm{obs}}|}}\Big(\frac{|\mathcal{V}_{\mathrm{obs}}|}{n}\Big)^{|\mathcal{V}_{\mathrm{obs}}|+1}
\tag{M.56}
$$

and

$$
\frac{\log(M)}{|\mathcal{V}_{\mathrm{obs}}|} > \frac{|\mathcal{V}_{\mathrm{obs}}|+1}{|\mathcal{V}_{\mathrm{obs}}|}\log\Big(\frac{|\mathcal{V}_{\mathrm{obs}}|}{\lceil|\mathcal{V}_{\mathrm{obs}}|/2\rceil}\Big) + \frac{1}{|\mathcal{V}_{\mathrm{obs}}|}\log\Big(\sqrt{\frac{\pi}{2e^4|\mathcal{V}_{\mathrm{obs}}|}}\Big)
\tag{M.57}
$$
$$
> \frac{|\mathcal{V}_{\mathrm{obs}}|+1}{|\mathcal{V}_{\mathrm{obs}}|}\log 2 + \frac{1}{|\mathcal{V}_{\mathrm{obs}}|}\log\Big(\sqrt{\frac{\pi}{2e^4|\mathcal{V}_{\mathrm{obs}}|}}\Big) > \frac{1}{10}
\tag{M.58}
$$

By choosing $\mathbf{\Sigma}_{\mathrm{r}} = \mathbf{I}$ and letting all the eigenvalues of $\mathbf{A}$ to be the same, we have

$$
\rho_{\max} = -\frac{3}{2}\log\big(\det(\mathbf{A}^2)\big) = -3l_{\max}\log\big(\lambda(\mathbf{A})\big)
\tag{M.59}
$$

and

$$
\mathrm{trace}(\mathbf{\Sigma}_{\mathrm{r}}) - \log\big(\det(\mathbf{\Sigma}_{\mathrm{r}})\big) - l_{\max} = 0.
\tag{M.60}
$$

Furthermore, we have

$$
\log\big(\det(\mathbf{I}-\mathbf{A}^2)\big) = l_{\max}\log\big(1-\lambda(\mathbf{A})^2\big) = l_{\max}\log\big(1-e^{-\frac{2\rho_{\max}}{3l_{\max}}}\big).
\tag{M.61}
$$

By choosing $\delta' = \delta + \frac{1}{\log(M)}$, we have that the condition

$$
n < \frac{(1-\delta')/5 - \frac{2}{|\mathcal{V}_{\mathrm{obs}}|}}{-l_{\max}\log\big(1-e^{-\frac{2\rho_{\max}}{3l_{\max}}}\big)}
\tag{M.62}
$$

guarantees that

$$
\max_{\theta(\mathbb{T})\in\mathcal{T}(|\mathcal{V}_{\mathrm{obs}}|,\rho_{\max},l_{\max})}\mathbb{P}_{\theta(\mathbb{T})}\big(\phi(\mathbf{X}_1^n)\neq\mathbb{T}\big) \geq \delta'
\tag{M.63}
$$

as desired. $\qquad\square$