# OpenReview forum: "Robustifying Algorithms of Learning Latent Trees with Vector Variables"
_NeurIPS.cc/2021/Conference — NeurIPS 2021 Poster_

### Official Review · Reviewer_jt2c · 2021-07-15

**Rating:** 8
**Confidence:** 3

**Summary:**

The authors introduce various theoretical results for learning latent trees including the following: a) extends RG to be applicable to GGMs (sec 3.2), b) reduce sample complexity for RG using Chow-Liu initialization, which depends of random vectors that fulfill a monotonic property (sec 3.4), and c) present an impossibility result that is useful for comparing theoretical results and directing future effort on this topic. Experimental results with synthetic data mirror the theoretical results, and both RCLRG and RSNJ perform markedly better than the baselines.

**Limitations And Societal Impact:**

The authors thoroughly address limitations by the nature of their theoretical results, but also clearly describe assumptions where applicable.

**Main Review:**

Strengths

- All claims seem to be valid, although I may have missed any subtle errors.
- This is clearly a substantial extension of previous work, and is relevant to many researchers who work on closely or loosely related areas. Graphical models are relevant to most if not all of the ML community.
- The related work is well organized and presented clearly to highlight the exact contributions of this paper. The entire paper is highly technical, yet well written and easy to follow.
- The detailed appendix is incredibly helpful. It includes additional details for proofs, algorithms, and results.

Weaknesses

- Minor complaint. It would benefit the reader to briefly explain why noisy samples are important in this context even if it is touched upon in related work (e.g. the intro from Choi et al.).


**Time Spent Reviewing:**

4

---

> ### Author Response · Authors · 2021-08-09
> **Response to reviewer jt2c**
>
> We thank the reviewer for the careful reading and insightful comments.
>
> We will supplement the importance of learning with noisy samples in the revised version.

---

### Official Review · Reviewer_YapK · 2021-07-16

**Rating:** 7
**Confidence:** 2

**Summary:**

Summary:

The paper studies the structural learning problems on latent tree models, in which a subset of vector observations have been arbitrarily corrupted. The paper proposes a series of "robustified" algorithms, including the robustified recursive grouping (RRG), robustified Chow-Liu recursive grouping (CLRG), robustified neighbor joining (RNJ), and robustified spectral neighbor joining (RSNJ). The authors analyze the sample complexity of the algorithms and show that a number of corruptions up to the square root of the number of clean samples can be tolerated. The paper also provides an information theoretic lower bound for this class of graphical models.

**Limitations And Societal Impact:**

Yes.

**Main Review:**

Strength:

It seems there are a series of solid results presented in this paper, namely the robustified algorithms and the corresponding upper limits for recovery.

The assumption of an arbitrary corruption pattern is interesting and, arguably, more challenging in analysis compared to, say, uniform corruptions. This is also more desirable from an application point of view. The related discussion in the appendix is interesting.

According to the paper the information theoretic lower bound in Theorem 5 is novel for its class, although it seems the Fano's inequality proof is standard.


Weakness:

Given the current state the paper suffers from the writing, which can be improved a lot.

- This paper deals with corruptions in graphical models, however I cannot find the formal definition of corruption. This is never defined in Section 2. Please let me know if I misunderstood something.

- Boldface x_i: Is this a vector or an entry? See Line 67 vs Line 75.

- Use the full names of the algorithms instead of the acronyms for their first appearance in the introduction section.
    Here are a few:
    - Line 23: NJ -> Neighbor Joining (NJ)
    - Line 26: RG -> Recursive Grouping (RG)
    - Line 46: CLRG ->  Chow-Liu Recursive Grouping

- Consider rephrasing Line 75 - 78.

Regarding Line 343 - 349: What else can be said about the gap in RRG and RSNJ? Is it because of the analysis not being tight, or the original RG and SNJ algorithm not being optimal?


**Time Spent Reviewing:**

2

---

> ### Author Response · Authors · 2021-08-09
> **Response to reviewer YapK**
>
> We thank the reviewer for the careful reading and insightful comments.
>
> The definition of corruptions is described by words in the introduction and illustrated by figures in this version. Please see Fig. 3 in the supplementary. Basically, a subset of entries of the data matrix is not assumed to follow the latent tree statistical model and can be arbitrary. We will provide precise mathematical definitions in the revised version.
>
> In our paper, we denote vectors as boldface characters. The entries in the vectors are denoted as boldface characters enclosed by square brackets. For example, the $i^{\rm{th}}$ entry of the vector $\mathbf{x}$ is written as $[\mathbf{x}]_{i}$.
>
> The reason why RRG and RSNJ are suboptimal is because RG and SNJ are themselves not optimal. For RRG, the exponential relationship with $L_{\rm{R}}$ results from the need to estimate distances across layers which is a salient feature of the original RG algorithm in Choi et al. (2011). The estimation of long distances results in severe error propagation from the bottom layer to the top layer and this in turn results in the sample complexity being exponential in $L_{\rm{R}}$. RSNJ uses a certain spectral property of the "symmetric affinity matrix" $\mathbf{S}$. However, this spectral property is closely related to the size of the graph, as described in Theorem 4.1 in Jaffe et al.(2021). When $e^{-2\rho_{\max}}\geq 0.5$, the lower bound of the second largest singular value of a submatrix of $\mathbf{S}$ is linear in the number of nodes, which dramatically increases the sample complexity of RSNG, resulting in it being suboptimal. Thus, both RRG and RSNG are suboptimal but in different ways.

---

### Official Review · Reviewer_fher · 2021-07-16

**Rating:** 5
**Confidence:** 4

**Summary:**

In this paper, the authors consider the problem of learning Gaussian latent tree models when the nodes of the tree are vector variables and the observations are arbitrarily corrupted. The paper uses the algorithms of Choi et al. (2011) as the base and extends their results (which are for the case when the nodes in the tree are scalar and there is no corruption) to vector variables with an arbitrarily corrupted subset of samples. Furthermore, they provide a more refined sample complexity analysis and highlight the advantage of using Chow-Liu based initialization in the sample complexity. They also provide novel impossibility results.

**Limitations And Societal Impact:**

The paper does not seem to have any negative societal impact.

**Main Review:**

Originality:

Not novel– Graph-theoretic aspects of the algorithms RG, NJ, SNJ, and CLRG have all been exactly used as they exist in the literature.

Incremental Novelty- Once we have an additive distance metric these algorithms are straightforward to follow. Thus the extension of the results to vector random variables is a straightforward case of plugging the distance measure (proposed in Huang et al. (2020)) into the existing algorithms. Similarly imparting robustness is also a straightforward application of Chen et al. (2013) in the estimation of distance.

Novelty- (i) Sample complexity results that highlight the exponential dependence on the number of iterations for RRG. (ii) The potential improvement in the HMM case when initialized with Chow-Liu tree. (iii) The insight that the Chow-Liu tree is not the same as the MST of the distance metrics and the class of graphical models for which they are the same.

Quality:

My current understanding is that in the RRG algorithm, the authors should consider only a subset of the active nodes which have a distance below a given threshold at each iteration following on the lines relaxed RG algorithm of Choi et al. (2011).
In that direction, it would be great if the authors can answer this question – The proposed RRG algorithm considers all the nodes in the active set while making decisions about sibling or parent child relationships in the nodes (line 5,7,8 of Algorithm 1) which seems to be motivated by the RG algorithm of Choi et al. (2011). However, shouldn’t the authors build upon the relaxed RG algorithm of Choi et al. (2011) which is developed for the finite sample setting and only considers nodes in the active set that have distance smaller than a threshold while determining parent child or sibling relationships? Would this change the sample complexity – in particular, will the exponential dependence on the number of iterations of RRG continue to be the case?

Furthermore, my understanding of the RG algorithm of Choi et al. (2011) is that it is the steppingstone to the (better) CLRG algorithm. Thus, the extensive discussion of the RG algorithm in this paper would only make sense if the CLRG algorithm does not apply to some settings. Indeed, as per the discussion in the paper about the potential non-monotonicity of the distance with the mutual information, it seems that CLRG might not be applicable in general. However, I could not really understand why do we have to initialize with the MST of the mutual information. My understanding is that the algorithm would go through just fine if we simply initialized with the MST of the empirical distances and did not worry about the mutual information (yes, this tree would not be the Chow-Liu tree). I might be missing something and I would really appreciate it if the authors can point to what would break down if I used the above approach.

A minor point is that the authors seem to paint the picture that their setting is more challenging compared to [7-11] because they consider arbitrary corruption (lines 40-41, 99). While the setting of arbitrary corruptions is a very interesting setting especially when the corruptions can be present in the different components, it feels misleading to present it as a more difficult problem. While in the setting of [7-11] there are restrictions on the noise model, $all$ the samples are noisy, unlike the setting in this paper where only a small ($\mathcal{O}(\sqrt{n})$) $subset$ of the samples are noisy.

Clarity:

The paper is clearly written and easy to follow.

Significance:

According to my understanding of the paper right now, the results in this paper are incremental. Answering the questions in the Quality section would help in highlighting the significance of the contributions of this paper.

I am happy to change my evaluation if the authors can answer the above questions.


**Time Spent Reviewing:**

20

---

> ### Author Response · Authors · 2021-08-09
> **Response to reviewer fher**
>
> We thank the reviewer for the careful reading and insightful comments.
>
> Q1: If we consider a robustified version of relaxed RG (as in Choi et al. (2011)), there would only be a potential improvement only in the dependence on a polynomial in $L_{\rm{R}}$ in the sample complexity in (H.19) (and the theorem statement). This is due to the application of the union bound over possibly fewer nodes, whereas our bound is more conservative, and the union bound is applied over more nodes in general. However, the precise analysis of relaxed RG is more involved because the cardinality (and in fact identities) of the nodes in the active set that are sufficiently close to $x_i$ and $x_j$ (i.e., $|\mathcal{K_{ij}}|$) are random. The exponential dependence on $L_{\rm{R}}$ originates from the calculation of the distances between new nodes, i.e., equations (8) and (9) in which errors propagate from the bottom layer to the top layer. This exponential dependence cannot be avoided by using the refined analysis, hence, the analysis of relaxed RRG is probably not worth the effort.
>
> Q2: We sincerely thank that the reviewer for this astute observation that initialization of the MST with the robustified mutual information (MI) quantities is not necessary. Indeed, using the robustified distances suffices and there is no need to establish the monotonicity of the MI and distances for successful structure recovery. This is because the CLRG algorithm in Choi et al. (2011) depends only on the distances and not the MIs. Hence, our sufficient condition that neighboring nodes are connected by linear Gaussian channels parametrized by $\mathbf{A}$ (call this Condition A) is, in fact, not necessary for the success of RCLRG. However, the discussion and analysis of the RRG algorithm are indeed crucial, because they serve as a key stepping stone for the analysis of RCLRG—RRG is employed on the closed neighborhood of every internal node of the initialized Chow-Liu tree. Also, the analysis of the error propagation in RRG helps us to understand why RCLRG reduces the sample complexity in some cases (e.g., the hidden Markov model).
>
> Even though Condition A is not necessary for RCLRG, this (linear) parametrization turns out to be useful for us to construct instances to derive the novel impossibility result in Section 4. We will remove Condition A for CLRG in the revised version of the paper and move the linear Gaussian model to the impossibility section, where it is needed.
>
> Q3: We believe that [7-11] and our work study two different noise models, so they are not comparable. The papers [7-11] consider the *random* noise model that corrupts every entry, but the corruptions are generally small in magnitude. In contrast, our work studies the gross corruption model on a small set of entries $O(\sqrt{n})$ where the amplitudes of the corruptions may not be bounded. Such unbounded errors, if left unchecked, will disastrously decrease the accuracy of classical structure learning algorithms. Thus, these two different noise models capture different perspectives of robust structure learning.
>
> We sincerely thank the reviewer for the many astute comments and for being open to changing her/his evaluation. We hope to have addressed all your concerns and will make the changes to our paper to incorporate your comments. We reiterate that we refine the analyses of many existing latent tree learning algorithms exposing their dependencies of the sample complexity on the problem parameters (which had not been done systematically before). We also study the effect of adversarial corruptions on the data matrix. Finally, we also derive the first known impossibility result for learning latent trees, showing that some algorithms are not optimal and others are for some latent trees.

---

> > ### Comment · Reviewer_fher · 2021-09-20
> > **Thank you for the response**
> >
> > I thank the authors for their response. When I revisited the paper after the response, I realized that the authors are in fact extending the relaxed RG. As a reader, I am not convinced about the utility of the analysis of RRG, which is clearly a suboptimal algorithm compared to RCLRG. It could potentially be a small subset of the analysis of RCLRG but does not need an extensive discussion. Therefore, in the current form, I believe that the paper is incremental.
> >
> > Furthermore, the extent of the gain of RCLRG over RRG could potentially be a side effect of the analysis of RRG. In particular, going from H.7 to H.8 and H.9 to H.10, the authors drop the dividing constants which then later show up as multiplicative constants in H.12. However, this is not a contributing factor to my evaluation of the paper.

---

> > > ### Author Response · Authors · 2021-09-21
> > > **Response to reviewer fher**
> > >
> > > We thank the reviewer for the attention and the detailed reply.
> > >
> > > We feel that the reviewer is overly harsh on the point concerning RRG. Let us clarify. The discussion of RRG, which spans only 1.5 pages in the main text, serves as a *stepping stone* for the discussion and analysis of RCLRG (as RRG is performed on each of the internal nodes of the Chow-Liu initialized tree). We use the former to prove that the latter algorithm performs well; namely that its sample complexity depends on $L_{\mathrm{C}}$, which is significantly smaller than $L_{\mathrm{R}}$. We do not feel that the discussion of RRG makes the paper incremental as there are still a number of results presented herein *go well beyond* RRG, including the sample complexity analysis of RCLRG, neighbor joining and its variants, experimental results, and importantly, the lower bound.

---

### Decision · Program_Chairs · 2021-09-27

**Decision:**

Accept (Poster)

**Comment:**

This paper studies the problem of learning Gaussian latent tree models where the nodes are vector-valued random variables and there are corruptions. There were differing opinions among the reviewers, but in aggregate they agreed that the extension of existing works (notably the algorithms of Choi et al.) from the scalar-valued to vector-valued case was interesting and non-trivial. They also show how the Chow-Liu Recursive Grouping algorithm (as opposed to the Recursive Grouping algorithm) leads to improved dependence on the diameter of the underlying tree. Finally they give robustified algorithms based on the thresholded inner-product, whereby they can tolerate a total number of corruptions of about the squareroot of the number of clean samples. This paper has a nice mix of contributions and would make a solid addition to the conference.